# GENERALIZATION OF TWO-LAYER NEURAL NETWORKS: AN ASYMPTOTIC VIEWPOINT

**Jimmy Ba[1,2], Murat A. Erdogdu[1,2], Taiji Suzuki[3,4], Denny Wu[1,2,4], Tianzong Zhang[2,5]**
University of Toronto[1], Vector Institute[2], University of Tokyo[3], RIKEN AIP[4], Tsinghua University[5]
`{jba,erdogdu,dennywu}@cs.toronto.edu`,
`taiji@mist.i.u-tokyo.ac.jp, ztz16@mails.tsinghua.edu.cn`

## ABSTRACT

This paper investigates the generalization properties of two-layer neural networks in high-dimensions, i.e. when the number of samples $n$, features $d$, and neurons $h$ tend to infinity at the same rate. Specifically, we derive the exact population risk of the unregularized least squares regression problem with two-layer neural networks when either the first or the second layer is trained using a gradient flow under different initialization setups. When only the second layer coefficients are optimized, we recover the *double descent* phenomenon: a cusp in the population risk appears at $h \approx n$ and further overparameterization decreases the risk. In contrast, when the first layer weights are optimized, we highlight how different scales of initialization lead to different inductive bias, and show that the resulting risk is *independent* of overparameterization. Our theoretical and experimental results suggest that previously studied model setups that provably give rise to *double descent* might not translate to optimizing two-layer neural networks.

## 1 INTRODUCTION

In modern neural networks, the number of parameters can easily exceed the number of training samples, yet in many circumstances, there is little sign of overfitting even in the absence of explicit regularization (Zhang et al., 2016). This phenomenon is usually explained by the interplay between the model architecture and the optimization method. Existing works have analyzed the implicit regularization of gradient descent on simple models (Gunasekar et al., 2018; Ji and Telgarsky, 2018), and provided generalization guarantees (Arora et al., 2018; Bartlett et al., 2017; Dziugaite and Roy, 2017) that align with the empirical observations.

Recently, a series of works highlighted the implicit regularization of interpolators in the overparameterized regime (Belkin et al., 2018; Spigler et al., 2018; Geiger et al., 2018; Advani and Saxe, 2017). Specifically, a second decrease in the population risk is observed when the model is further overparameterized beyond the interpolation limit, i.e. when the model achieves zero training error. This phenomenon is known as *double descent*, and can be precisely quantified for certain linear models (Hastie et al., 2019; Mei and Montanari, 2019; Belkin et al., 2019; Bartlett et al., 2019; Xu and Hsu, 2019). Among the recent works, Hastie et al. (2019) and Mei and Montanari (2019) explicitly derived the population risk of linear regression and random features regression models in high dimensions using tools from random matrix theory.

However, there is still a gap between the practical benefit of overparameterization and the recently proved *double descent* phenomenon, which is typically established under models that exhibits the following structure: the trained model solves a linear inverse problem, and the "cusp" in the risk arises from the instability of the inverse at the interpolation threshold. Moreover, given a dataset or fixed $n, d$, the number of parameters in the linear regression model is also fixed, i.e. the level of overparameterization cannot be altered. It is therefore unclear if the trend persists in the optimization of more complex models, for instance in two-layer neural networks where overparameterization can be controlled simply by adding more neurons.

In this work, we analyze the generalization properties of two-layer neural networks in the unregularized least squares regression setting and examine the presence/absence of the double descent phenomenon. We consider the proportional asymptotic limit where the number of samples $n$, input

features $d$, and neurons $h$ tend to infinity at the same rate, under which overparameterization corresponds to increasing the limit of $h/n$ (network "width"). This regime is particularly interesting because even though $n \to \infty$, the empirical risk is not equivalent to the population risk. In addition, the joint scaling of $n, d, h$ is parallel to the practical choice of model architectures, where it is common to train a larger network when the number of samples and input features are larger. Following Hastie et al. (2019), we assume unit Gaussian input and noisy linear observations, and analytically derive the population risk of the solution of gradient flow on either the first or the second layer parameters when the flow is initialized close to zero.

Our findings can be summarized as follows (see Figure 1):

- When only the second layer is optimized, we derive the risk in its bias-variance decomposition and demonstrate the presence of the *double descent* phenomenon.

- When the first layer is optimized, we compare two solutions of gradient flow from different scales of initialization, which we term as *vanishing* and *non-vanishing* initialization, and show in both cases the population risk is independent to overparameterization.

- For the vanishing initialization, we show that the risk of the gradient flow solution is asymptotically close to that of a rank-1 model. For non-vanishing initialization, we show that the gradient flow solution is well-approximated by a kernel model and derive the risk.

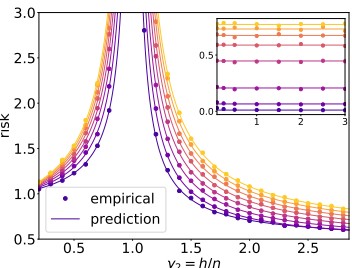

Figure 1: Illustration of the *double descent* risk curve in two-layer linear networks (SNR= 16). Brighter color indicates larger $\gamma_1 = d/n$. *Double descent* is observed when the second layer coefficients are optimized (main figure), but not when the first layer weights are optimized (subfigure).

## 1.1 RELATED WORKS

**Global Convergence of Two-layer Networks.** A plethora of recent works have explored the global convergence of shallow neural networks. Mei et al. (2018; 2019); Chizat and Bach (2018a); Rotskoff and Vanden-Eijnden (2018); Sirignano and Spiliopoulos (2018); Nitanda and Suzuki (2017) studied the mean-field limit where the number of neurons $h \to \infty$ and the second layer scaled by $1/h$, and established correspondence between the main-particle limit of gradient descent and Wasserstein gradient flow to demonsrate global convergence. On the other hand, Jacot et al. (2018); Du et al. (2018); Oymak and Soltanolkotabi (2019); Allen-Zhu et al. (2018b); Song and Yang (2019) considered a different scaling and showed that gradient descent on overparameterized models converges to global minimizer at a linear rate; key to these results is an observation that optimization via gradient descent is asymptotically equivalent to kernel regression with respect to the *neural tangent kernel*.

**Active vs. Lazy Training.** Following Chizat and Bach (2018b), we refer to the two aforementioned scalings as the *active* and *lazy* (kernel) regime. It has been observed that different regimes lead to contrasting inductive biases. Williams et al. (2019); Woodworth et al. (2019); Li et al. (2017) showed that for certain two-layer network or overparameterized linear model, the scale of initialization controls the implicit regularization of gradient descent (from sparse to smooth solution). In the student-teacher setup (Tian, 2017; Zhong et al., 2017), Ghorbani et al. (2019b;a) showed that kernel models in high dimensions perform no better than low-degree polynomials on the input or fully-trained two-layer network. Additionally, Suzuki (2018); Allen-Zhu and Li (2019); Yehudai and Shamir (2019); Wei et al. (2018) demonstrated that neural network outperforms linear estimators (including kernel method) in learning various target functions. The difference between fixed bases and adaptive bases mirrors the difference in optimizing the first or second layer in our setup.

**Generalization of Overparameterized Models.** It is often observed that overparameterization does not result in overfitting (Neyshabur et al., 2014). In the lazy regime, generalization guarantees can be derived from the distance traveled by the parameters (Neyshabur et al., 2018; Nagarajan and Kolter, 2019), which becomes small if the model is sufficiently overparameterized (Arora et al., 2019b; Li and Liang, 2018; Allen-Zhu et al., 2018a; Cao and Gu, 2019). Compared to these guarantees that usually require significant overparameterization, our result relies on stronger data assumptions, but consequently we obtain the exact population risk instead of a vacuous upper-bound. Beyond the kernel regime, Advani and Saxe (2017); Goldt et al. (2019) analyzed the generalization dynamics of overparameterized models in the student-teacher setup.

**Double Descent.** The term *double descent* refers to the phenomenon that the population risk of an empirical risk minimizer manifests a "cusp" at the interpolation threshold, and further overparameterization decreases the risk. First observed in Krogh and Hertz (1992), the phenomenon has been recently connected to the benefit of overparameterization (Belkin et al., 2018; Geiger et al., 2018; Spigler et al., 2018; Advani and Saxe, 2017), and can be precisely characterized for certain simple models (Hastie et al., 2019; Belkin et al., 2019; Bartlett et al., 2019; Xu and Hsu, 2019). Our work is inspired by Hastie et al. (2019) which uses random matrix theory to derive the asymptotic risk for linear and random feature models. Concurrent to our work, Mei and Montanari (2019) analyzed the random features model and derived its population risk for which double descent occurs both in bias and variance. This aligns with our results on optimizing the second layer in Section 4 although we do not derive the bias component explicitly. Compared to Hastie et al. (2019); Mei and Montanari (2019), the focus of this work is to highlight the different generalization property of models obtained from optimizing different layers of the network and from different initialization.

**Random Matrix Theory.** High-dimensional models, including kernel models and neural networks, can be analyzed by studying the properties of random matrices. El Karoui et al. (2010); Cheng and Singer (2013); Fan and Montanari (2019) studied the spectral properties of kernel matrix via decomposing the nonlinearity with Taylor series or Hermite polynomials, which in turn explains the generalization of high-dimensional kernel ridgeless interpolators (Liang and Rakhlin, 2018). In addition, similar tools have been used to study two-layer neural networks (Louart et al., 2018; Pennington and Worah, 2017) and related quantities such as the Fisher information matrix (Karakida et al., 2018; Pennington and Worah, 2018).

## 2 PRELIMINARIES: TWO-LAYER NEURAL NETWORK

Consider the following bias-free two-layer neural network $f : \mathbb{R}^d \to \mathbb{R}$ with $h$ hidden units

$$f(\boldsymbol{x}) = \sum_{i=1}^{h} a_i \phi(\langle \boldsymbol{x}, \boldsymbol{w}_i \rangle), \tag{1}$$

where $\boldsymbol{x} \in \mathbb{R}^d$ is the input, $\boldsymbol{w}_i \in \mathbb{R}^d$ is the weights corresponding to neuron $i$, $a_i \in \mathbb{R}$ is the $i$-th coefficient of the second layer, and $\phi : \mathbb{R} \to \mathbb{R}$ is a Lipschitz continuous activation function with bounded Gaussian moments, i.e. $\mathbb{E}[\phi(G)^k] < \infty, \forall k \in \mathbb{Z}_+$ for $G \sim \mathcal{N}(0, 1)$. For concise notation, we write $W = [\boldsymbol{w}_1, ... \boldsymbol{w}_h] \in \mathbb{R}^{d \times h}$ for the weight matrix, $\boldsymbol{a} = [a_1, ... a_h] \in \mathbb{R}^h$ for the coefficient vector, $X = [\boldsymbol{x}_1, ... \boldsymbol{x}_n] \in \mathbb{R}^{d \times n}$ for the data matrix, $\boldsymbol{y} \in \mathbb{R}^n$ for the corresponding vector of labels, and $\Phi = \phi(W^\top X) \in \mathbb{R}^{h \times n}$ for the feature matrix at the first layer. We omit arguments of $f$ when they are clear from the context.

We consider a student-teacher setup, in which data is generated by a teacher model $F : \mathbb{R}^d \to \mathbb{R}$ with additive noise, and the student model aims to minimize the squared loss:

$$(\boldsymbol{x}_i, \varepsilon_i) \overset{\text{i.i.d.}}{\sim} P_{\boldsymbol{x}} \times P_\varepsilon, \quad y_i = F(\boldsymbol{x}_i) + \varepsilon_i, \quad L(X; f) = \frac{1}{2n} \sum_{i=1}^{n} (y_i - f(\boldsymbol{x}_i))^2, \tag{2}$$

where $\mathbb{E}[\boldsymbol{x}_i] = 0, \text{Cov}(\boldsymbol{x}_i) = \Sigma, \mathbb{E}[\varepsilon_i] = 0, \text{Var}(\varepsilon_i) = \sigma^2$. We are interested in the population risk $R(f) = \mathbb{E}_{P_{\boldsymbol{x}}}[(F(\boldsymbol{x}) - f(\boldsymbol{x}))^2]$. Our analysis will be made under the proportional asymptotics:

$$n, d, h \to \infty; \quad d/n \to \gamma_1, \ h/n \to \gamma_2; \quad \gamma_1, \gamma_2 \in (0, \infty),$$

in which overparameterization corresponds to increasing $\gamma_2$. Thus the characteristics of *double descent* considered in this work are: 1) large population risk as $\gamma_2 \to 1$; 2) decrease in the risk for $\gamma_2 > 1$. While the empirical risk can be minimized in various ways, we analyze the solution of gradient flow, in which we update either the first layer $W$ or the second layer $\boldsymbol{a}$:

$$\mathrm{d}W(t) = -\nabla_W L(X; f)\,\mathrm{d}t \quad \text{or} \quad \mathrm{d}\boldsymbol{a}(t) = -\nabla_{\boldsymbol{a}} L(X; f)\,\mathrm{d}t, \tag{3}$$

from small initialization. The rest of the paper is organized as follows. In Section 3, we start with a simple example of two-layer linear network as warm-up. In Section 4, we consider optimizing the second layer coefficients (flow over $\boldsymbol{a}$) of a non-linear two-layer neural network under fixed Gaussian first layer, which is a random feature model. Section 5 considers optimizing the first layer weights (flow over $W$) of such network under fixed Rademacher second layer. We defer all proofs and details on experiments to appendix.

## 3   WARM-UP: LINEAR NETWORK

We begin with a simple linear model with $\phi(\boldsymbol{x}) = \boldsymbol{x}$, i.e. $\Phi = W^\top X$. We remark that although the model is linear, the solution obtained by gradient flow on the two-layer model can be different than that from directly solving the linear regression problem on input features.

**Training the Second Layer.**   Following Hastie et al. (2019), we fix the first layer parameters to be randomly drawn from a unit Gaussian and optimize the coefficients $\boldsymbol{a}$ by minimizing $\left\| \boldsymbol{a}^\top \Phi - \boldsymbol{y} \right\|_2^2$. The following lemma characterizes the solution of the gradient flow.

**Lemma 1** (Least squares solution). *Given data matrix X, response vector $\boldsymbol{y}$ and model $f(\boldsymbol{x}) = \langle \phi(\boldsymbol{x}^\top W), \boldsymbol{a} \rangle$ with fixed first layer coefficients $W$, gradient flow on the coefficients $\boldsymbol{a}$ starting from zero initialization converges to $\hat{\boldsymbol{a}} = \Phi^\dagger \boldsymbol{y}$, where $\dagger$ stands for the Moore-Penrose inverse.*

We make two assumptions on the data and the teacher model to simplify the computation.

**(A1) Gaussian Features:** $\boldsymbol{x}_i \sim \mathcal{N}(0, I_d)$; **(A2) Linear Teacher:** $F(\boldsymbol{x}) = \langle \boldsymbol{x}, \boldsymbol{\beta} \rangle$, $\|\boldsymbol{\beta}\| = r$.

Denote the linear student network as $f(\boldsymbol{x}) = \langle \boldsymbol{x}, \hat{\boldsymbol{\beta}} \rangle$, where $\hat{\boldsymbol{\beta}} = W \hat{\boldsymbol{a}}$ and $\hat{\boldsymbol{a}}$ is the least-square solution defined by Lemma 1. We write the population risk in its bias-variance decomposition.

$$R = \mathbb{E}_{\boldsymbol{x} \sim P_{\boldsymbol{x}}}[\|\hat{\boldsymbol{\beta}} - \boldsymbol{\beta}\|_\Sigma^2 | X, W] = \underbrace{\|\mathbb{E}[\hat{\boldsymbol{\beta}}|X, W] - \boldsymbol{\beta}\|_2^2}_{B = \text{bias}} + \underbrace{\text{tr}\left( \text{Cov}(\hat{\boldsymbol{\beta}}|X, W) \right)}_{V = \text{variance}}, \qquad (4)$$

where $\|\boldsymbol{x}\|_\Sigma^2 = \boldsymbol{x}^\top \Sigma \boldsymbol{x}$. We compute the bias and the variance separately to obtain the risk.

**Theorem 2.** *Given (A1)(A2) and let $\boldsymbol{w}_i \overset{\text{i.i.d.}}{\sim} \mathcal{N}(0, d^{-1} I_d)$, at $n, d, h \to \infty$ we have*

$$R_{(\gamma_1 < 1)} \to \begin{cases} \dfrac{\gamma_1 - \gamma_2}{\gamma_1 g_2} r^2 + \dfrac{\gamma_2}{g_2} \sigma^2, & \gamma_2 < \gamma_1, \\[2mm] \dfrac{\gamma_1}{g_1} \sigma^2, & \gamma_2 > \gamma_1, \end{cases} \qquad R_{(\gamma_1 > 1)} \to \begin{cases} \dfrac{\gamma_1 - \gamma_2}{\gamma_1 g_2} r^2 + \dfrac{\gamma_2}{g_2} \sigma^2, & \gamma_2 < 1, \\[2mm] \dfrac{\gamma_2 g_1}{\gamma_1 g_2} r^2 + \dfrac{g_1 + g_2}{g_1 g_2} \sigma^2, & \gamma_2 > 1. \end{cases}$$
$$(5)$$

*where $d/n \to \gamma_1$, $h/n \to \gamma_2$, $g_1 = |\gamma_1 - 1|$, and $g_2 = |\gamma_2 - 1|$.*

We observe that when $d > n$ (i.e. $\gamma_1 > 1$), we obtain the double descent risk curve, i.e., the population risk achieves its maximum at $\gamma_2 \to 1$ and further overparameterization ($\gamma_2 > 1$) reduces both the bias and the variance. Conversely when $n > d$ and $h > d$ (i.e. $\gamma_1 < \min(1, \gamma_2)$), the population risk becomes constant and equals to that of the minimum-norm solution $\hat{\boldsymbol{\beta}}_{\min} = X^\dagger \boldsymbol{y}$ on the input features.

**Training the First Layer.**   When the first layer of a linear network is optimized via gradient flow and the second layer is fixed, the following holds for zero-initialization of $W$.

**Proposition 3.** *Given $W(0) = 0$ and fixed $\boldsymbol{a}^{init} \neq 0$, at any time $t > 0$ of the gradient flow on $W$, $W(t)$ is rank-1. Further, $\hat{\boldsymbol{\beta}} = \widehat{W} \boldsymbol{a}$ converges to the least squares solution of $\boldsymbol{y} = X^\top \hat{\boldsymbol{\beta}}$, the population risk of which is given in (Hastie et al., 2019, Thm. 1 & 3)) as*

$$R_{(\gamma_1 < 1)} \to \frac{\gamma_1}{1 - \gamma_1} \sigma^2; \quad R_{(\gamma_1 > 1)} \to \frac{\gamma_1 - 1}{\gamma_1} r^2 + \frac{1}{\gamma_1 - 1} \sigma^2. \qquad (6)$$

In this case, overparameterization by increasing $\gamma_2$ does not influence the population risk. In addition, since the obtained two-layer linear model is equivalent to the minimum-norm solution on the input features $\hat{\boldsymbol{\beta}}_{\min}$, optimizing the first layer always results in smaller or equal population risk compared to optimizing the second layer.

In this simple scenario for two-layer linear networks, double descent is observed only when the second layer is optimized, which reduces the objective to least squares regression on the intermediate features. On the other hand, training the first layer from zero-initialization always yields the same solution that is independent to overparameterization. One natural question to ask is: *does this phenomenon generalize to nonlinear two-layer neural networks?* The following sections answer this question in the affirmative under certain conditions.

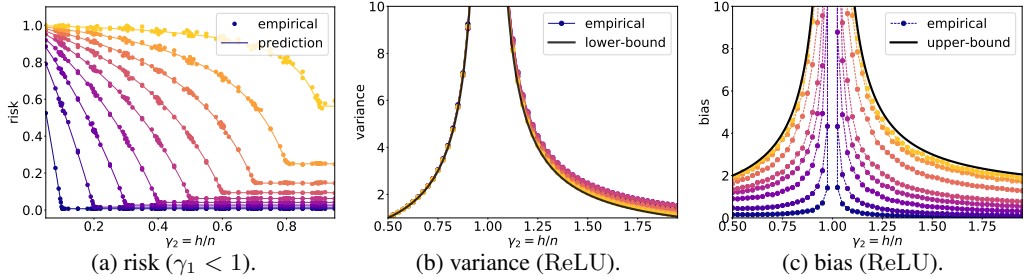

(a) risk ($\gamma_1 < 1$).     (b) variance (ReLU).     (c) bias (ReLU).

Figure 2: Population risk of two-layer neural networks with optimized second layer under (A1)(A2). Brighter color indicates larger $\gamma_1$. (a) risk of linear network with $r^2/\sigma^2 = 16$ and $\gamma_1 < 1$. ($\gamma_1 > 1$ is shown in Figure 1) (b) variance of network with ReLU activation. Black line corresponds to $\gamma_1 \to \infty$ predicted by Corollary 5. (c) bias of network with ReLU activation. Black line corresponds to $\gamma_1 \to \infty$ for linear network, which is empirically observed as an upper-bound. Note that as $\gamma_2 \to 1$ both bias and variance becomes unbounded.

## 4 NONLINEAR MODEL: OPTIMIZING THE SECOND LAYER

In this section, we analyze the case when the second layer $a$ is learned under fixed $W$ and a nonlinear activation function $\phi$ (a random feature model). We first observe that by Lemma 1, the gradient flow finds the solution $\hat{a} = \Phi^\dagger y$. We again consider the following bias-variance decomposition.

$$R = \mathbb{E}_{\boldsymbol{x} \sim P_{\boldsymbol{x}}} \left[ \left\| \phi(\boldsymbol{x}^\top W)\hat{a} - F(\boldsymbol{x}) \right\|_2^2 \big| X, W \right] \tag{7}$$

$$= \underbrace{\mathbb{E}_{\boldsymbol{x}} \left[ \left\| \mathbb{E}[\phi(\boldsymbol{x}^\top W)\hat{a} | X, W] - F(\boldsymbol{x}) \right\|_2^2 \right]}_{B = \text{bias}} + \underbrace{\mathbb{E}_{\boldsymbol{x}} \left[ \left\| \phi(\boldsymbol{x}^\top W)\hat{a} - \mathbb{E}[\phi(\boldsymbol{x}^\top W)\hat{a}] \right\|_2^2 \big| X, W \right]}_{V = \text{variance}}.$$

We highlight that the variance term does not depend on the target function. The following result characterizes the variance of the random feature model.

**Theorem 4.** *Given (A1) and $\boldsymbol{w}_i \overset{\text{i.i.d.}}{\sim} \mathcal{N}(0, d^{-1}I_d)$, when $n, d, h \to \infty$, we have*

$$V = \begin{cases} \sigma^2 \dfrac{\gamma_2}{1 - \gamma_2}, & \gamma_2 < 1, \\[3ex] \sigma^2 \lim_{\xi \to 0} - \left[ \gamma_2 \dfrac{\partial}{\partial x} m_1(\xi, c_1 x, c_2 x) \Big|_{x=0} + \dfrac{\partial}{\partial x} m_2(\xi, c_1 x, c_2 x) \Big|_{x=0} + \dfrac{\gamma_2 - 1}{\xi^2} \right] & \gamma_2 > 1. \end{cases}$$

*in which $m_1(\xi, \rho, \tau)$, $m_2(\xi, \rho, \tau)$ is the unique solution in $\{|m_1|, |m_2| < 1/\Im\xi\}$ of*

$$m_1^{-1} = -\xi - \rho - \gamma_1^{-1} \gamma_2 \tau^2 m_1 - c_1 m_2 + \frac{\tau^2 \gamma_1^{-1} \gamma_2 m_1^2 (c_2 m_2 - \tau) - 2\tau c_2 m_1 m_2 + c_2^2 m_1 m_2^2}{m_1(c_2 m_2 - \tau) - \gamma_1 \gamma_2^{-1}}, \tag{8}$$

$$m_2^{-1} = -\xi - r\gamma_2 m_1 + \frac{\gamma_2 c_2 m_1^2 (c_2 m_2 - \tau)}{m_1(c_2 m_2 - \tau) - \gamma_1 \gamma_2^{-1}}, \tag{9}$$

*where variables $\xi, \rho, \tau$ satisfies $\Im\xi > 0$ or $\xi < 0$, $\rho > \tau > 0$, and constants $c_1, c_2$ defined as,*

$$c_1 = \mathbb{E}[\phi(G)^2] - \mathbb{E}[\phi(G)]^2, \quad c_2 = \mathbb{E}[G\phi(G)]^2, \tag{10}$$

*for $G \sim \mathcal{N}(0,1)$ and $\Im\xi$ denoting the imaginary part of $\xi$.*

**Remark.** $c_1 \geq c_2$ and the equality holds iff $\phi$ is linear.

**Corollary 5.** *If we let $\gamma_1 \to \infty$, the variance is equal to the lowest value of the variance of the linear model $V_{(\gamma_1 \to \infty)} = \sigma^2 \min\{\gamma_2, 1\}/|1 - \gamma_2|$.*

The proof of Theorem 4 largely follows from Hastie et al. (2019) with techniques similar to Cheng and Singer (2013), but with modifications in otder to handle unnormalized and uncentered activation functions. The above theorem holds irrespective of the underlying teacher model, and is consistent with the double descent risk curve as it suggests that for all $\gamma_1$, variance of the random feature model

peaks at $h = n$ then drops as $\gamma_2$ further increases. Note that as $\gamma_1 \to \infty$, a linear and nonlinear network would have the same asymptotic variance.

Since double descent is observed in the variance term, we do not derive the bias for all $\gamma_1, \gamma_2$. Instead, we show that for linear teacher, the bias also becomes unbounded as $\gamma_2 \to 1$.

**Proposition 6.** *Given (A1)(A2) and $\boldsymbol{w}_i \overset{i.i.d.}{\sim} \mathcal{N}(0, I_d)$, then $B \to \infty$ as $\gamma_2 \to 1$. Furthermore, $B$ is finite when $\gamma_2 > 1$.*

Thus we have shown that a "cusp" in the population risk appears at $h = n$, which aligns with the double descent phenomenon. Empirically as $\gamma_1 \to \infty$ the nonlinear model also shares the same asymptotic bias with the linear model, as shown in Figure 2. We note that (Mei and Montanari, 2019, Thm. 1 & 3) analytically solved the risk of random feature model for a larger class of target functions than ours and confirmed that double descent appears in both the bias and the variance.

## 5 NONLINEAR MODEL: OPTIMIZING THE FIRST LAYER

Having observed the double descent phenomenon in optimizing the second layer, in the sequel we consider a two-layer neural network with fixed second layer coefficients initialized from a Rademacher distribution $a_i \sim \mathrm{Unif}\{-1/\sqrt{h}, 1\sqrt{h}\}$, and the first layer $W$ is optimized with the following update

$$\frac{\partial W(t)}{\partial t} = -\frac{\partial L(X; W)}{\partial W} = \frac{1}{n} \sum_{i=1}^{n} \left[ y_i - \boldsymbol{a}^\top \phi(W^\top \boldsymbol{x}_i) \right] \boldsymbol{x}_i [\phi'(\boldsymbol{x}_i^\top W) \circ \boldsymbol{a}], \tag{11}$$

which potentially has different stationary solutions with no explicit form, depending on the initialization. We denote the solution of this flow at time $t$ started from designated initialization by $W^{\mathrm{init}}(t)$, its stationary solution by $\widehat{W}$, and the corresponding network by $\hat{f}$.

***Remark.*** Although we let $n \to \infty$, this dynamics does not corresponds to the population gradient flow considered in Tian (2017). For instance when $\boldsymbol{x} \sim \mathcal{N}(0, I/d)$, the spectrum of the data covariance is Marčenko–Pastur, whereas the population covariance is identity.

As we cannot characterize the gradient flow solution from all possible initializations, we consider two specific scales of initialization:

$$\textbf{Vanishing: } \boldsymbol{w}_i^{\mathrm{Van}}(0) \sim \mathcal{N}(0, I_d/dh^{1+\epsilon}); \quad \textbf{Non-vanishing: } \boldsymbol{w}_i^{\mathrm{NV}}(0) \sim \mathcal{N}(0, I_d/d^{-\epsilon}).$$

Note that neither of the two initializations correspond to the "mean-field" regime (e.g. analyzed in Mei et al. (2018)) due to the $1/\sqrt{h}$ scaling of the second layer. In other words, as $h$ increases, we expect the distance traveled by each parameter to decrease under both initializations. The difference, however, is the "relative" amount the parameters traveled compared to their initialized magnitude, which leads to solutions with contrasting properties. As we will see, under (A1)(A2) and vanishing initialization we have $\|W(0) - \widehat{W}\|_F / \|W(0)\|_F \gg 1$, i.e. the contribution of initialization vanishes at the end of training, whereas for non-vanishing initialization the inequality is in the opposite direction, i.e. $\widehat{W}$ "barely moves" and resembles the initialization $W(0)$.

### 5.1 VANISHING INITIALIZATION

As $d, h \to \infty$, the vanishing initialization becomes arbitrarily close to zero-initialization. We thus expect the gradient flow under vanishing initialization to "resemble" that of starting from exactly zero if the flow converges sufficiently fast and the gradient being Lipschitz. The Lipschitz condition (Lemma 18) can be established under the following assumption on the activation.

   **(A3):** $\phi$ is smooth, Lipschitz and monotone with $\phi'(0) \neq 0$; $|\phi'(\pm x) - \phi'(\pm\infty)| = O(e^{-x})$.

The above assumption requires that the derivative of the nonlinearity $\phi$ saturates beyond a $O(1)$ region, which holds true for the commonly-used smooth activations such as sigmoid and SoftPlus. In addition, the choice of scaling ensures that the gradient flow converges sufficiently fast. We thus have the following characterization of the population risk:

**Theorem 7.** *Given (A1-3). Let $T = O(\log \log h)$ and $\hat{f}(\cdot) = f^{van}(\cdot, W(T))$, then as $n, d, h \to \infty$, the gradient flow reaches a $o(1)$ first-order stationary point at time $T$, i.e. $\|\partial W(T)/\partial t\|_F \in o(1)$, at which point the population risk is given as*

$$R(\hat{f}) \to \max\left\{0, \frac{\gamma_1 - 1}{\gamma_1}\right\} r^2 + \frac{\min\{\gamma_1, 1\}}{|1 - \gamma_1|} \sigma^2. \tag{12}$$

The expression above is the same as the risk of the least squares solution on input $\hat{\boldsymbol{\beta}} = X^\dagger \boldsymbol{y}$; therefore the risk is independent to overparameterization (increasing $\gamma_2$). The intuition is that when the weights are initialized sufficiently small and travel infinitesimally, then the activation can be linearized around 0 and thus the model is equivalent to a two-layer linear network. Note that this result does not apply to the non-smooth ReLU activation. Instead, in Appendix E we heuristically show that under the additional assumption that the data is symmetric, the risk of ReLU network is also independent to $\gamma_2$.

## 5.2 NON-VANISHING INITIALIZATION

When initialization is sufficiently large, the amount each parameter travels to minimize the empirical risk becomes asymptotically negligible compared to the magnitude of initialization. In this case we establish under (A1-3) that (11) is asymptotically equivalent to the kernel gradient flow on the *tangent kernel*: $k(\boldsymbol{x}, \boldsymbol{y}) = \langle \nabla_{W^{init}} f(\boldsymbol{x}), \nabla_{W^{init}} f(\boldsymbol{y}) \rangle$. The converged parameters under this linearized dynamics has the following closed-form:

$$\text{vec}(W^*) \approx \text{vec}(W^{init}) + \Delta; \quad \Delta = J^\dagger(\boldsymbol{y} - f^{init}(X)); \quad J_{[i,j]} = \nabla_{\text{vec}(W^{init})_j} f^{init}(\boldsymbol{x}_i), \tag{13}$$

where $J \in \mathbb{R}^{n \times (d \times h)}$ is the Jacobian matrix w.r.t. to the model parameters. We remark that in contrast to most NTK-type global convergence results that require the width of the model to grow faster than the number of data points (e.g. Du et al. (2018)), our result is not built upon the overparameterization on width, but instead an anti-concentration that relies on the scale of initialization. Consequently the above initialization is larger than the scale that is commonly used in practice.

One may naturally expect the *double descent* phenomenon to appear in this kernel solution, as $\Delta$ exhibits the form of a least squares solution which contains a pseudo-inverse. However, we show that this is not the case under the same assumptions in Section 4; in fact, the risk is also independent to $\gamma_2$.

An obstacle in computing the risk of the kernel model is the potentially non-zero $f^{init}(X)$. We thus adopt the "doubling-trick" from Chizat and Bach (2018b) to ensure $f^{init}(\cdot) = 0$, i.e. we assume the following symmetric property on the initialized weights:

**(A4) Symmetric Initialization:** $\forall i \in [1, h], \exists! j \in [1, h]$ s.t. $a_i \boldsymbol{w}_i^{init} = -a_j \boldsymbol{w}_j^{init}$.

**Theorem 8.** *Given (A1-4) and let $n, d, h \to \infty$, the stationary solution $\hat{f}$ has the following risk*

$$R(\hat{f}) \to \left( \frac{\gamma_1 - 1}{2\gamma_1} + \frac{\gamma_1(\gamma_1 + \gamma_1 m + m - 2) + 1}{2\gamma_1\sqrt{\gamma_1(\gamma_1 + m(\gamma_1(m+2)+2)-2)+1}} \right) r^2$$

$$+ \left( \frac{\gamma_1 + \gamma_1 m + 1}{4\sqrt{\gamma_1(\gamma_1 + m(\gamma_1(m+2)+2)-2)+1}} - \frac{1}{4} \right) \sigma^2, \tag{14}$$

*where $m = b_1^2/b_0^2$, $b_0^2 = \mathbb{E}[\phi'(G)]^2$, and $b_1^2 = \mathbb{E}[\phi'(G)^2] - b_0^2$, $G \sim \mathcal{N}(0, 1)$.*

Note that the population risk is again independent to $\gamma_2$, and thus *double descent* does not appear for this initialization. Roughly speaking, the reason that the risk does not become unbounded at some point is that in the asymptotic limit the pseudo-inverse $(JJ^\top)^\dagger$ is stable due to the nonlinearity and $dh \gg n$. We make two additional observations:

- the stability of the inverse at $\gamma_1 \to 1$ depends on the lowest eigenvalue of the tangent kernel matrix (smaller $\lambda_{\min}(K)$ entails larger variance), which is determined by the nonlinearity;
- While our result only holds for network with zero initial output, for non-symmetric (i.i.d.) initialization we also observe that the risk is independent to $\gamma_2$, but the bias is higher than that in symmetric initialization as shown in Figure 8. We comment that the non-zero $f^{init}(X)$ in (13) behaves as zero-mean Gaussian due to central limit theorem; therefore, in the kernel regime the function output at initialization is equivalent to additive noise to the labels $\boldsymbol{y}$, and we thus expect the magnitude of $f^{init}(X)$ to negatively influence the model generalization.

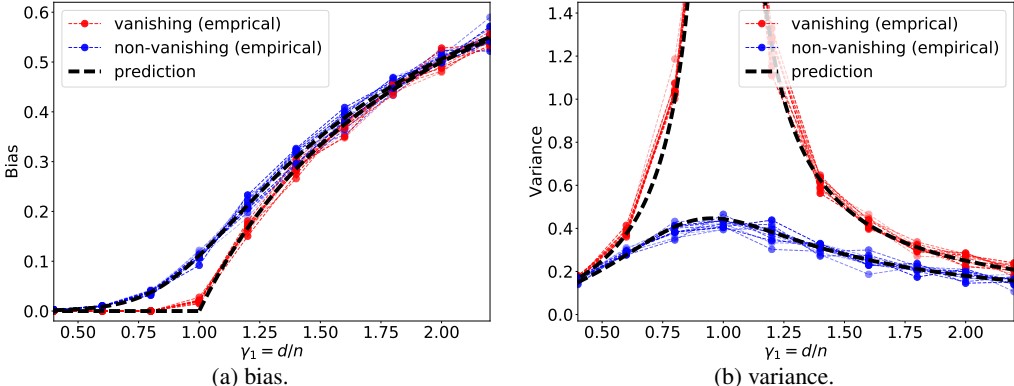

(a) bias.  (b) variance.

Figure 3: Bias and variance of two-layer sigmoid network with optimized first layer under (A1)(A2). Individual dotted lines correspond to different $\gamma_2$ (from 0.2 to 2) which is independent to the risk. The bias and variance for both initializations is well-aligned with Theorem 7 and Theorem 8.

### 5.3 COMPARING THE INITIALIZATIONS

Figure 3 shows the agreement between theoretical prediction and experimental results. Although in both cases the risk is independent to overparameterization ($\gamma_2$), the two initializations lead to models with contrasting properties, as demonstrated by the following comparison on the risk.

**Corollary 9.** *For any $\gamma_1 \in (0, \infty)$ and nonlinearity $\phi$, $B(\hat{f}^{Van}) \leq B(\hat{f}^{NV}) \leq 1$. On the other hand, for all $m > 0$, $V(\hat{f}^{NV}) = O(1)$, whereas $V(\hat{f}^{Van})$ can be arbitrarily large as $\gamma_1 \to 1$.*

**Remark.** $m \geq 0$ for all smooth activations $\phi$, and the equality holds if $\phi$ is linear.

Intuitively, small initialization enables the model "evolve" more during optimization and better align with the data and target function. This potentially results in a lower bias, at the expense of overfitting more to the noise (high variance). In contrast, with sufficiently large initialization the final model becomes close to the initialized model, and thus we may expect it to be less "aligned" to the target (high bias) but is more stable (lower variance).

In illustrate the different inductive bias of the two initializations, we plot the trajectory of neurons in Appendix A Figure 4. Observe that for vanishing initialization the neurons stay close to one another throughout the trajectory, which results in a low-rank weight matrix, as predicted by Theorem 7. In contrast, for non-vanishing initialization the neurons stay close to initialization (therefore full-rank), which validates the kernel approximation. Last but not least, although the derived risk is only for learning a linear target function, we empirically observe that when the teacher is also a two-layer network, the population risk follows the same trend, i.e. *double descent* occurs when only the second layer is optimized, as shown in Figure 7.

## 6 DISCUSSION AND FUTURE WORKS

We derived the exact population risk of high-dimensional two-layer neural networks in learning a linear target function over Gaussian data with additive label noise, and showed that optimizing the first or the second layer via gradient flow results in solutions with contrasting properties. Specifically, *double descent* is present when the second layer coefficients are optimized, but not when the first layer weights are optimized under certain initializations. Moreover, we highlight that the scale of initialization leads to different inductive bias in optimizing the first layer.

It should be noted that our analysis only applies to the unregularized objective: it has been shown that explicit regularization (such as $\ell_2$ penalty) stabilizes the singularity at $\gamma_2 \to 1$ (Mei and Montanari, 2019), and algorithmic regularization (Li et al., 2019a; Dong et al., 2019) also provides robustness against noisy observations. We further remark that our findings do not directly contradict the experimental *double descent* phenomenon, nor the practical benefit of overparameterization. In particular, the interpolation limit could occur at $\gamma_2 \to 0$ which is beyond the regime we consider (such as Figure 4 in (Belkin et al., 2018)). Thus what we conclude is that under the studied proportional

asymptotics, the mechanism that provably gives rise to *double descent* from previous works on least squares regression might not translate to neural networks trained with gradient descent.

To simplify the computation, we rely on a set of strong assumptions similar to those in Hastie et al. (2019), some of which we believe can be relaxed in future works, such as isotropic Gaussian input and linear target. Importantly, the two specific scales of initialization studied in Section 5 are by no means exhaustive, and thus one would expect that under a different initialization of the first layer, or the mean-field $1/h$ scaling of the second layer, the risk of the trained model can be very different. Changing the loss function may also alter the generalization behavior of the network. Another challenging problem is to extend the current analysis to beyond two layers. Last but not least, our result characterizes gradient flow which resembles gradient descent with small stepsize, and thus it would be interesting to study the effect of learning rate schedule, which has a known impact on generalization (Smith and Le, 2017; Li et al., 2019b).

### ACKNOWLEDGEMENT

We thank Xiuyuan Cheng, Xuechen Li, Yiping Lu, Atsushi Nitanda, Shengyang Sun and anonymous reviewers for helpful comments and feedback. JB and DW were partially funded by LG Electronics and NSERC. JB and MAE were supported by the CIFAR AI Chairs program. TS was partially supported by JSPS Kakenhi (26280009, 15H05707 and 18H03201), Japan Digital Design and JST-CREST.

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

## A  ADDITIONAL FIGURES AND PLOTS

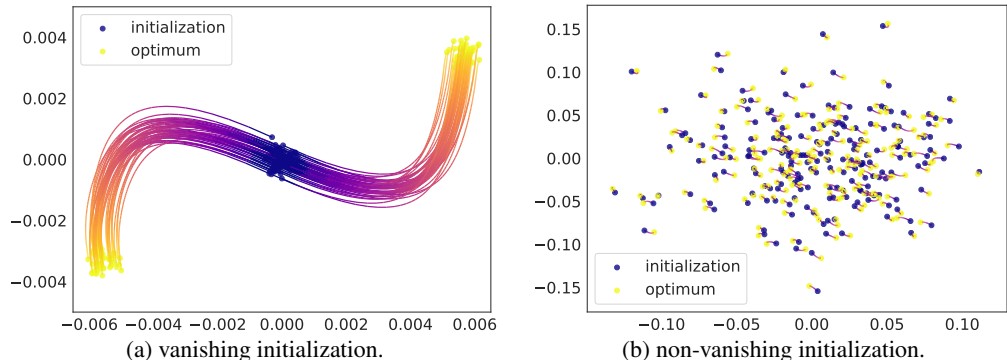

(a) vanishing initialization.  (b) non-vanishing initialization.

Figure 4: trajectory of neurons from initialization (dark blue) to optimum (orange) on the first two dimensions (two-layer SOFTPLUS student and linear teacher; SNR= $1/4$). For vanishing initialization the neurons stay close to one another throughout the trajectory, whereas for non-vanishing initialization the neurons stay close to initialization.

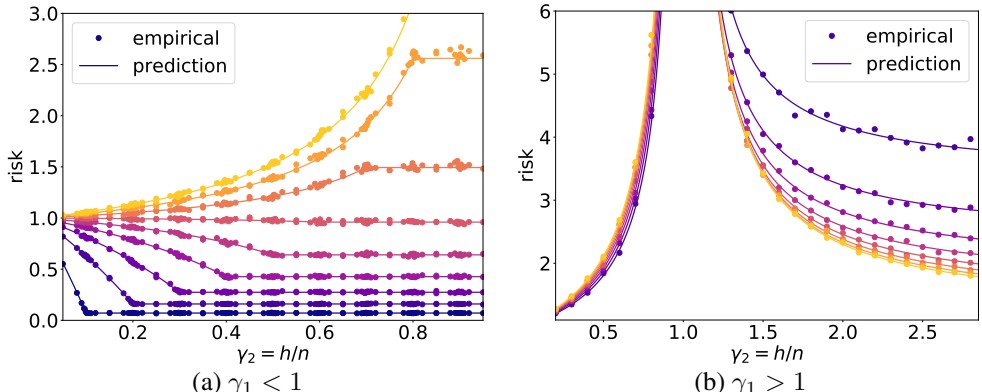

(a) $\gamma_1 < 1$  (b) $\gamma_1 > 1$

Figure 5: Population risk of two-layer linear network with fixed random 1st layer with SNR=$25/16$ under Gaussian input and linear teacher. Brighter color indicates larger $\gamma_1$.

SUMMARY OF THE PRESENCE / ABSENCE OF DOUBLE DESCENT

| Singularity in | 2nd Layer Trained (RF) | Vanishing Init. | Non-vanishing Init. |
|---|---|---|---|
| Bias | $\gamma_1$: No; $\gamma_2$: Yes | $\gamma_1$: No; $\gamma_2$: No | $\gamma_1$: No; $\gamma_2$: No |
| Variance | $\gamma_1$: No; $\gamma_2$: Yes | $\gamma_1$: Yes; $\gamma_2$: No | $\gamma_1$: No; $\gamma_2$: No |

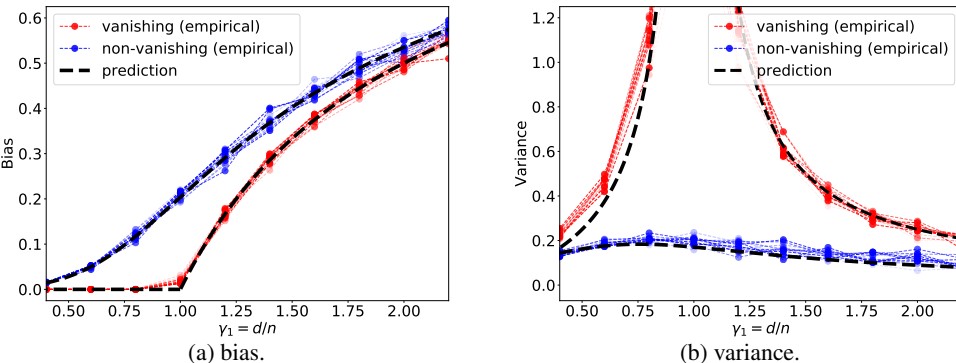

Figure 6: Bias and variance of two-layer SoftPlus network with optimized first layer under (A1)(A2). Individual dotted lines correspond to different $\gamma_2$ (from 0.2 to 2) which is independent to the risk. The bias and variance for both initializations is well-aligned with Theorem 7 and Theorem 8, respectively.

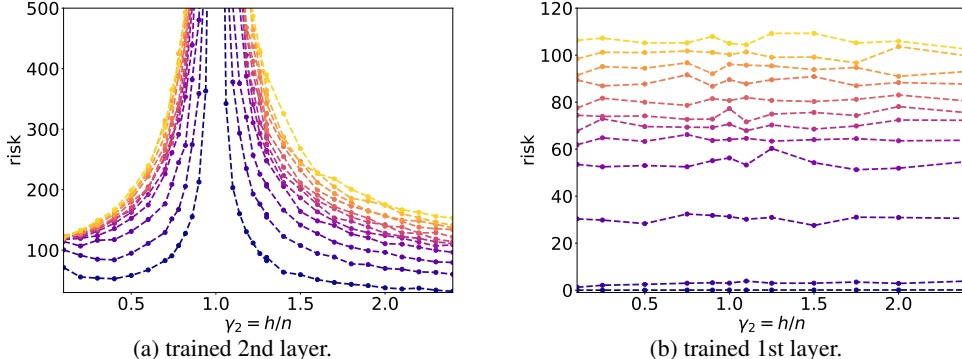

Figure 7: Population risk (scaled by $1/d$) of two-layer ReLU network trained to fit a two-layer ReLU teacher model with $h = d$ neurons. Brighter color corresponds to larger $\gamma_1$. Similar to the linear teacher case, *double descent* is observed when the second layer is optimized (a) but not when the first layer is optimized (b).

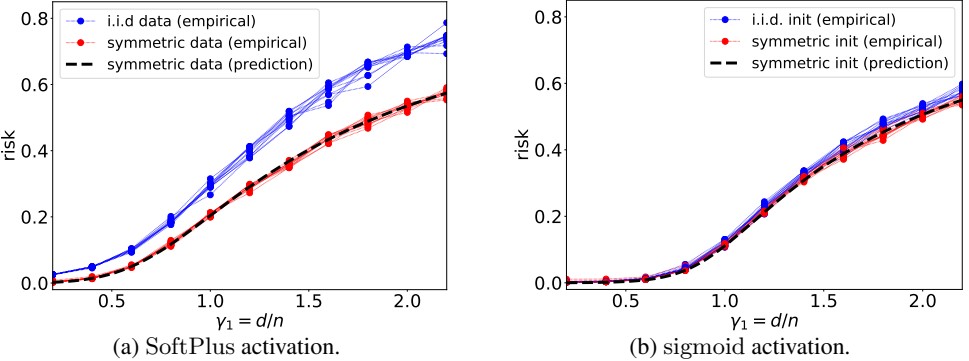

Figure 8: Bias of (a) SoftPlus and (b) sigmoid two-layer network with optimized first layer under (A1)(A2). Note that bias under i.i.d. initialization is also independent to overparameterization ($\gamma_2$), but is higher than the bias under symmetric initialization ("doubling trick") and not always upper-bounded by the null risk $r^2$.

# B  BACKGROUND

## B.1  ROTATIONAL INVARIANCE

The rotational invariance of Gaussian distribution is crucial in our analysis throughout this paper. A basic observation is that for a random Gaussian matrix $X$ and any fixed unitary matrix $U$, the distribution of $X$ and $UX$ are the same.

**Lemma 10** (Rotational Invariance). *Denote $A(X) \in \mathbb{R}^{d \times d}$ a matrix function of $X \in \mathbb{R}^{d \times n}$. If $A(X)$ satisfies that $A(UX) = UA(X)U^T$ for all unitary $U$, then*

$$\mathbb{E}_X[\boldsymbol{\beta}^T A(X)\boldsymbol{\beta}] = \frac{1}{d}\boldsymbol{\beta}^T\boldsymbol{\beta}\mathbb{E}_X[\operatorname{tr}(A(X))]. \tag{15}$$

*for any fixed nonzero $\boldsymbol{\beta} \in \mathbb{R}^d$ and random matrix $X$ with each entry i.i.d. $X_{ij} \sim \mathcal{N}(0, \sigma^2)$.*

**Proof.** Choose a set of Unitary matrices $\{U_i\}_{i=1}^d$ such that $U_i^\top \boldsymbol{\beta} = \|\boldsymbol{\beta}\|\, \boldsymbol{e}_i$, where $\boldsymbol{e}_i$ is the $i$-th canonical vector in $\mathbb{R}^d$. Since $U_i X \sim X$, we have $\mathbb{E}[A(X)] = \mathbb{E}[A(U_i X)] = U_i \mathbb{E}[A(X)]U_i^\top$ and hence

$$\mathbb{E}[\boldsymbol{\beta}^T A(X)\boldsymbol{\beta}] = \frac{1}{d}\sum_{i=1}^d \mathbb{E}[\boldsymbol{\beta}^T U_i A(X) U_i^\top \boldsymbol{\beta}] = \frac{1}{d}\sum_{i=1}^d \boldsymbol{e}_i^T \mathbb{E}[A(X)]\boldsymbol{e}_i = \frac{\boldsymbol{\beta}^T\boldsymbol{\beta}}{d}\mathbb{E}[\operatorname{tr}(A(X))]. \tag{16}$$

Note that the property also holds for matrix function $A(X)$ that satisfies $A(XU) = UA(X)U^T$, and can be extended to matrix function $A$ that takes multiple matrices as input. $\qquad\square$

For brevity we will refer to rotational invariance instead of equation (15).

## B.2 MARČENKO–PASTUR LAW

For a real symmetric random matrix $A \in \mathbb{R}^{p \times p}$, define its empirical spectral density as

$$\mu_A(d\lambda) = \frac{1}{p}\sum_{i=1}^p \delta_{\lambda_i(A)}(\lambda)d\lambda,$$

where $\delta_a(x) = \delta(x - a)$ is the Dirac delta function. Assume $A = W_p \sim W_p(I, n)$ is a *Wishart matrix*, i.e. $W_p = X^\top X/n$ and $X \in \mathbb{R}^{n \times p}$ is random Gaussian matrix with each column i.i.d. $X_i \sim \mathcal{N}(\mathbf{0}, I)$. Marčenko and Pastur (1967) showed that as $n, p \to \infty$ and $p/n = \gamma \in (0, \infty)$, the empirical spectral density $\mu_{W_p}(d\lambda)$ converges weakly to a limiting density $\mu_{\mathrm{MP}(\gamma)}(\lambda)$:

$$\mu_{\mathrm{MP}(\gamma)}(d\lambda) = [1 - \gamma^{-1}]_+\delta_0(\lambda)d\lambda + \frac{1}{2\pi\gamma\lambda}\sqrt{((1 + \sqrt{\gamma})^2 - \lambda)(\lambda - (1 - \sqrt{\gamma})^2)}d\lambda. \tag{17}$$

We say $\mu_{\mathrm{MP}(\gamma)}$ is the density of the *Marčenko–Pastur distribution* with support $S = [(1 - \sqrt{\gamma})^2, (1 + \sqrt{\gamma})^2]$ (for $0 < \gamma < 1$) or $S = \{0\} \cup [(1 - \sqrt{\gamma})^2, (1 + \sqrt{\gamma})^2]$ (for $\gamma \geq 1$). Note that this implies that the smallest non-zero eigenvalue of $A$ is bounded away from 0 a.s. for $\gamma \neq 1$.

The explicit form of Marčenko–Pastur distribution allows us to investigate the asymptotic properties of random matrices. Generally speaking, by Pormanteau theorem one can translate any bounded continuous function on the empirical spectral density to the one on Marčenko–Pastur distribution, i.e. for any bounded $f(\lambda) \in C(S)$, as $n, p \to \infty$ with $p/n = \gamma$, almost surely

$$\int_S f(\lambda) \cdot \mu_W(d\lambda) \to \int f_S(\lambda) \cdot \mu_{MP}(d\lambda). \tag{18}$$

One implication is the following trace concentration on the inverse Wishart matrix for $\gamma < 1$,

$$\operatorname{tr}\left(X^\top X\right) = \operatorname{tr}\left(\frac{1}{p}W_p^{-1}\right) = \frac{1}{p}\sum_{i=1}^p \frac{1}{\lambda_i(W_p)} = \int_S \frac{1}{\lambda}\mu_{W_p}(d\lambda) \to \int_S \frac{1}{\lambda}\mu_{\mathrm{MP}(\gamma)}(d\lambda) = \frac{1}{1-\gamma}. \tag{19}$$

We remark that instability of the trace of the invert Wishart matrix as $\gamma \to 1$ plays an important role in the double descent phenomenon.

## B.3 ORTHOGONAL POLYNOMIALS

Orthogonal polynomials are useful in the analysis of nonlinear random matrices. Suppose $\phi$ is a function in $L^2(\mathbb{R}, \mu_G)$, where $G \sim \mathcal{N}(0, 1)$, $\mu_G(dx) = (\sqrt{2\pi})^{-1}e^{-x^2/2}dx$ is the Gaussian measure. For $n \geq 0$, define the Hermite polynomials

$$H_n(x) = (-1)^n e^{-x^2/2}\frac{\partial^n}{\partial x^n}e^{-x^2/2}, \tag{20}$$

Note that orthogonality can be easily verified:

$$\mathbb{E}[H_j(G)H_k(G)] = \int_{\mathbb{R}} H_j(x)H_k(x)\mu_G(dx) = j! \cdot \delta_{jk}. \tag{21}$$

Since $\{H_i(x)\}_{i=0}^{\infty}$ forms a set of orthogonal basis in $L^2(\mathbb{R}, \mu_G)$, the function $\phi(x)$ can be expanded under the Hermite basis as

$$\phi(x) = \sum_{i=0}^{\infty} c_i H_i(x) = \sum_{i=0}^{\infty} \left( \frac{1}{k!} \int_{\mathbb{R}} \phi(a)H_i(a)\mu_G(da) \right) H_i(x). \tag{22}$$

Following Cheng and Singer (2013), we mainly focus on the first few terms of the Hermite expansion. One can check that $H_0(x) = 1$, $H_1(x) = x$, and $\phi(x)$ can be expanded as

$$\phi(x) = c_0 + c_1 x + \phi_{\perp}(x), \tag{23}$$

where $c_0 = \mathbb{E}[\phi(G)]$, $c_1 = \mathbb{E}[G\phi(G)]$, and terms in the RHS are orthogonal to one another in $L^2(\mathbb{R}, \mu_G)$. Taking square and expectation over both sides yields

$$\mathbb{E}[\phi(G)^2] = \mathbb{E}[\phi(G)]^2 + \mathbb{E}[G\phi(G)]^2 + \mathbb{E}[\phi_{\perp}(G)^2], \tag{24}$$

which indicates $\mathbb{E}[\phi(G)^2] - \mathbb{E}[\phi(G)]^2 - \mathbb{E}[G\phi(G)]^2 = \mathbb{E}[\phi_{\perp}(G)^2] \geq 0$. Note that the equality holds if and only if $\mathbb{E}[\phi_{\perp}(G)^2] = 0$, i.e. $\phi(x) = c_0 + c_1 x$ is linear. One application of this decomposition is to "linearize" a non-linear matrix in high dimensions, which will be useful for the following sections.

## C   PROOF OF MAIN RESULTS

### C.1   PROOF OF LEMMA 1

Given features $X \in \mathbb{R}^{d \times n}$, labels $\boldsymbol{y} \in \mathbb{R}^d$ and model parameters $\boldsymbol{\theta}$, the gradient flow of $\boldsymbol{\theta}$ on the squared loss $\left\| \boldsymbol{y} - X^{\top}\boldsymbol{\theta} \right\|_2^2$ can be written as

$$\frac{\partial \boldsymbol{\theta}(t)}{\partial t} = \frac{1}{n} X(y - X^{\top}\boldsymbol{\theta}(t)). \tag{25}$$

Thus with initialization $\boldsymbol{\theta}_0$, the solution of this ODE at time $t$ can be written in explicit form

$$\boldsymbol{\theta}(t) = e^{-\frac{t}{n}XX^{\top}}\boldsymbol{\theta}_0 + (XX^{\top})^{\dagger} \left( I - e^{-\frac{t}{n}XX^{\top}} \right) X\boldsymbol{y}. \tag{26}$$

Since $\boldsymbol{\theta}_0 = 0$, taking $t \to \infty$ yields the desired result. $\qquad\square$

### C.2   PROOF OF THEOREM 2

We compute the bias and variance for different cases of $\gamma_1, \gamma_2$. We first discuss the case where the random feature $\Phi_X$ is not full rank (Case I). Otherwise when $\Phi_X$ is full rank, we discuss whether it is full column rank (Case II) or full row rank (Case III).

**Case I: $W^{\top}X$ is not full rank,**   i.e. $\gamma_1 < 1, \gamma_2 > \gamma_1$. In this case $\operatorname{rank}(\Phi_X) = d < \min(n, h)$, and thus by taking the Moore-Penrose inverse we obtain

$$\hat{\boldsymbol{\beta}} = W(X^{\top}W)^{\dagger}\boldsymbol{y} = (XX^{\top})^{-1}X\boldsymbol{y}. \tag{27}$$

It is clear that the mean and variance is identical to the underparameterized regime in Hastie et al. (2019), i.e. when $n, d, h \to \infty$,

$$B \to 0; \quad V \to \frac{\gamma_1}{1 - \gamma_1}\sigma^2. \tag{28}$$

**Case II: $W^\top X$ has full column rank,** i.e. $\gamma_2 < 1, \gamma_1 > \gamma_2$. By Lemma 1, the solution of the second layer coefficients is

$$\hat{\boldsymbol{\beta}} = W(W^\top X X^\top W)^{-1} W^\top X \boldsymbol{y}. \tag{29}$$

Denote $W = U\Sigma V^\top$ the singular value decomposition. We perform the block decomposition:

$$\Sigma = \begin{bmatrix} \Sigma_0 \\ 0 \end{bmatrix}, \ X = \begin{bmatrix} X_0 \\ X_1 \end{bmatrix}, \tag{30}$$

where $\Sigma_0 \in \mathbb{R}^{h \times h}, X_0 \in \mathbb{R}^{h \times n}, X_1 \in \mathbb{R}^{(d-h) \times n}$, and notice that $X_0, X_1$ are independent. By a concentration of measure argument (e.g. Tao (2012); Hastie et al. (2019)) one can show that the quantity below tightly concentrates at its expectation. For the variance we have

$$
\begin{aligned}
V &= \operatorname{tr}\left( W \left(W^\top X X^\top W\right)^{-1} W^\top X^\top \sigma^2 X W \left(\left(W^\top X X^\top W\right)^{-1} W^\top\right) \right) \\
&= \sigma^2 \operatorname{tr}\left( W^\top W \left(W^\top X X^\top W\right)^{-1} \right) = \sigma^2 \operatorname{tr}\left( \Sigma^\top \Sigma \left(\Sigma^\top U^\top X X^\top U\Sigma\right)^{-1} \right) \\
&\sim \sigma^2 \operatorname{tr}\left( \Sigma^\top \Sigma \left(\Sigma^\top X X^\top \Sigma\right)^{-1} \right) = \sigma^2 \operatorname{tr}\left( (X_0 X_0^\top)^{-1} \right) \to \sigma^2 \frac{\gamma_2}{1 - \gamma_2},
\end{aligned}
\tag{31}
$$

where the last equality follows from Appendix B.2. Similarly for the bias term we have

$$
\begin{aligned}
B &= \left\| W(W^\top X X^\top W)^{-1} W^\top X X^\top \boldsymbol{\beta} - \boldsymbol{\beta} \right\|_2^2 \\
&= \boldsymbol{\beta}^\top \left( W(W^\top X X^\top W)^{-1} W^\top X X^\top - I_d \right)^\top \left( W(W^\top X X^\top W)^{-1} W^\top X X^\top - I_d \right) \boldsymbol{\beta} \\
&\overset{(i)}{=} \frac{r^2}{d} \operatorname{tr}\left( \left( W(W^\top X X^\top W)^{-1} W^\top X X^\top - I_d \right)^\top \left( W(W^\top X X^\top W)^{-1} W^\top X X^\top - I_d \right) \right) \\
&= \frac{r^2}{d} \operatorname{tr}\left( \left( U\Sigma V^\top (V\Sigma^\top U^\top X X^\top U\Sigma V^\top)^{-1} V\Sigma^\top U^\top X X^\top - I_d \right)^\top \left( \cdots \right) \right) \\
&= \frac{r^2}{d} \operatorname{tr}\left( \left( U\Sigma(\Sigma^\top X X^\top \Sigma)^{-1}\Sigma^\top X X^\top U^\top - UU^\top \right)^\top \left( \cdots \right) \right) \\
&= \frac{r^2}{d} \operatorname{tr}\left( \left( \Sigma(\Sigma^\top X X^\top \Sigma)^{-1}\Sigma^\top X X^\top - I_d \right)^\top \left( \cdots \right) \right),
\end{aligned}
\tag{32}
$$

where symmetric arguments are omitted as $(\cdots)$, and (i) follows from the rotational invariance argument introduced in Lemma 10 and that $\boldsymbol{\beta}^\top \boldsymbol{\beta} = r^2$. By the block decomposition (30),

$$\Sigma \left(\Sigma^\top X X^\top \Sigma\right)^{-1} \Sigma^\top X X^\top - I_d = \begin{bmatrix} 0 & (X_0 X_0^\top)^{-1} X_0 X_1^\top \\ 0 & -I_{d-h} \end{bmatrix}. \tag{33}$$

Therefore the bias term simplifies to

$$
\begin{aligned}
B &= \frac{r^2}{d} \operatorname{tr}\left( \left( \Sigma(\Sigma^\top X X^\top \Sigma)^{-1}\Sigma^\top X X^\top - I_d \right)^\top \left( \Sigma(\Sigma^\top X X^\top \Sigma)^{-1}\Sigma^\top X X^\top - I_d \right) \right) \\
&= \frac{r^2}{d} \left( \operatorname{tr}\left( (X_0 X_0^\top)^{-1} X_0 X_1^\top X_1 X_0^\top (X_0 X_0^\top)^{-1} \right) + (d-h) \right) \\
&\to \frac{r^2}{d} \left( \frac{(d-h)h}{n-h-1} + d-h \right) \to \frac{\gamma_1 - \gamma_2}{\gamma_1(1-\gamma_2)} r^2.
\end{aligned}
\tag{34}
$$

Thus we have obtained that as $n, d, h \to \infty$

$$B \to \frac{\gamma_1 - \gamma_2}{\gamma_1(1-\gamma_2)} r^2. \tag{35}$$

**Case III: $W^\top X$ has full row rank,** i.e. $\gamma_1 > 1, \gamma_2 > 1$. Similarly, the least squares solution is

$$\hat{\boldsymbol{\beta}} = WW^\top X(X^\top WW^\top X)^{-1} \boldsymbol{y}, \tag{36}$$

Simplifying the variance:

$$
\begin{aligned}
V &= \mathrm{tr}\left(WW^\top X \left(X^\top WW^\top X\right)^{-1} \sigma^2 \left(X^\top WW^\top X\right)^{-1} X^\top WW^\top\right) \\
&= \sigma^2 \mathrm{tr}\left(WW^\top U\Sigma V^\top \left(V\Sigma^\top U^\top WW^\top U\Sigma V^\top\right)^{-2} V\Sigma^\top U^\top WW^\top\right) \\
&\sim \sigma^2 \mathrm{tr}\left(WW^\top \Sigma \left(\Sigma^\top WW^\top \Sigma\right)^{-2} \Sigma^\top WW^\top\right).
\end{aligned} \tag{37}
$$

where we applied the SVD of $X = U\Sigma V^\top$ and the rotational invariance argument. Using a similar block decomposition on $W$:

$$
\Sigma = \begin{bmatrix} \Sigma_0 \\ 0 \end{bmatrix}, \ W = \begin{bmatrix} W_0 \\ W_1 \end{bmatrix}, \tag{38}
$$

where $\Sigma_0 \in \mathbb{R}^{n\times n}, W_0 \in \mathbb{R}^{n\times h}, W_1 \in \mathbb{R}^{(d-n)\times h}$, and $W_0, W_1$ independent. We thus simplify $V$ as

$$
\begin{aligned}
V &= \sigma^2 \mathrm{tr}\left(WW^\top \Sigma (\Sigma^\top WW^\top \Sigma)^{-2}\Sigma^\top WW^\top\right) \\
&= \sigma^2 \mathrm{tr}\left(\begin{bmatrix} W_0 W_0^\top \Sigma_0 (\Sigma_0^\top W_0 W_0^\top \Sigma_0)^{-2}\Sigma_0 W_0 W_0^\top & \cdots \\ \cdots & W_1 W_0^\top \Sigma_0 (\Sigma_0^\top W_0 W_0^\top \Sigma_0)^{-2}\Sigma_0 W_0 W_1^\top \end{bmatrix}\right) \\
&= \sigma^2 \left(\mathrm{tr}\left(\Sigma_0^{-T}\Sigma_0^{-1}\right) + \mathrm{tr}\left(W_1 W_0^\top \left(W_0 W_0^\top\right)^{-1}\Sigma_0^{-T}\Sigma_0^{-1}\left(W_0 W_0^\top\right)^{-1} W_0 W_1^\top\right)\right) \\
&= \sigma^2 \mathrm{tr}\left(\left(X^\top X\right)^{-1}\right) + \sigma^2 \mathrm{tr}\left(W_1^\top W_1 W_0^\top \left(W_0 W_0^\top\right)^{-1}\left(\Sigma_0^\top \Sigma_0\right)^{-1}\left(W_0 W_0^\top\right)^{-1} W_0\right). \tag{39}
\end{aligned}
$$

Hence we obtain the following expression on the variance

$$
V \to \sigma^2 \frac{1}{\gamma_1 - 1} + \sigma^2 (d-n)\mathbb{E}_{W,X} V \mathrm{tr}\left(\left(W_0 W_0^\top\right)^{-1}\left(\Sigma_0^\top \Sigma_0\right)^{-1}\right) \to \sigma^2 \left(\frac{1}{\gamma_1 - 1} + \frac{1}{\gamma_2 - 1}\right). \tag{40}
$$

We omit the derivation of bias, which follows a similar derivation:

$$
B \to \frac{\gamma_2(\gamma_1 - 1)}{\gamma_1(\gamma_2 - 1)} r^2. \tag{41}
$$

Combining Case I, II, III yields theorem 2. $\qquad\square$

### C.3 PROOF OF PROPOSITION 3

Given the squared loss, one can derive the dynamics of $W$ with fixed second layer $\boldsymbol{a}$ w.r.t the loss:

$$
\frac{\partial W(t)}{\partial t} = -\frac{1}{n} X(\boldsymbol{y} - X^\top W(t)\boldsymbol{a})\boldsymbol{a}^\top. \tag{42}
$$

Note that the update of $W$ can be written as a linear combination of $\boldsymbol{a}$. Since $W(0) = 0$, we can write $W(t) = \hat{\boldsymbol{w}}(t)\boldsymbol{a}^\top$ for some $\hat{\boldsymbol{w}}$. The corresponding flow on $\hat{\boldsymbol{w}}$ is

$$
\frac{\partial \hat{\boldsymbol{w}}(t)}{\partial t} = -\frac{1}{n} X(\boldsymbol{y} - X^\top \hat{\boldsymbol{w}}(t)\|\boldsymbol{a}\|_2^2), \tag{43}
$$

which gives the following solution

$$
\hat{\boldsymbol{w}}^* = \frac{1}{\|\boldsymbol{a}\|_2^2} X^\dagger \boldsymbol{y} \ \Rightarrow \ \hat{\boldsymbol{\beta}} = W^* \boldsymbol{a} = X^\dagger \boldsymbol{y}. \tag{44}
$$

Thus gradient flow on the first layer leads to the minimum-norm solution on the input features. $\quad\square$

## C.4 Proof of Theorem 4

Following the bias-variance decomposition (7), the variance term can be written as ($\sigma^2$ omitted)

$$
\begin{aligned}
V &= \mathbb{E}_{\boldsymbol{x},\varepsilon}\left[\|\boldsymbol{a}^\top\phi(W^\top\boldsymbol{x}) - \mathbb{E}_\varepsilon\boldsymbol{a}^\top\phi(W^\top\boldsymbol{x})\|_2^2\right]\\
&= \mathbb{E}_{\boldsymbol{x}}\left[\left\|\left[\phi(W^\top X)\right]^\dagger\phi(W^\top\boldsymbol{x})\right\|_2^2\right]\\
&= \mathrm{tr}\left(\left[\phi(X^\top W)\right]^\dagger\left[\phi(W^\top X)\right]^\dagger\mathbb{E}_{\boldsymbol{x}}\left[\phi(W^\top\boldsymbol{x})\phi(W^\top\boldsymbol{x})^\top\right]\right)\\
&= \mathrm{tr}\left(\left[\phi(X^\top W)\right]^\dagger\left[\phi(W^\top X)\right]^\dagger K_W\right),
\end{aligned}
\tag{45}
$$

where we define the expected non-linear Gram matrix $K_W \in \mathbb{R}^{h\times h}$ as

$$
K_W = \mathbb{E}_{\boldsymbol{x}}\left[\phi(W^\top\boldsymbol{x})\phi(W^\top\boldsymbol{x})^\top\right].
\tag{46}
$$

and for each entry we have $(K_W)_{[i,j]} = \mathbb{E}_{\boldsymbol{x}}\left[\phi(\boldsymbol{w}_i^\top\boldsymbol{x})\phi(\boldsymbol{w}_j^\top\boldsymbol{x})\right]$.

Random matrix in the form of covariance matrix of nonlinear features has been studied in many works Hastie et al. (2019); Mei and Montanari (2019); Liao and Couillet (2018); Louart et al. (2018); Pennington and Worah (2017). We note that our setup for the variance term is very similar to that for nonlinear features in Hastie et al. (2019) with modifications mentioned below.

In contrast to the linear network in Section C.2, the Gram matrix of a nonlinear activation is almost surely full-rank, as specified in the following lemma from Pennington and Worah (2017):

**Lemma 11.** *Suppose $\phi$ is not linear. Then the smallest singular value of $\Phi = \phi(W^\top X) \in \mathbb{R}^{h\times n}$ is of order $O(\sqrt{n})$. To be precise, consider the empirical spectral density $\mu_{\Phi\Phi^\top/n}(d\lambda)$, when $n,d,h\to\infty$ with $d/n\to\gamma_1$ and $h/n\to\gamma_2$, $\mu_{\Phi\Phi^\top/n}(d\lambda)$ converges weakly to*

$$
\mu_{\Phi\Phi^\top/n}(d\lambda) \to [1-\gamma_2^{-1}]_+\delta_0(\lambda)d\lambda + \mu^+(d\lambda),
$$

*where $\mu^+(d\lambda)$ has non-negative support $[\rho,\infty)$ with $\rho > 0$.*

We therefore consider two scenarios: $\Phi$ is full column rank (Case I) or full row rank (Case II).

**Case 1.** $h < n$. In this case (45) simplifies into

$$
V = \mathrm{tr}\left(\left(\phi(W^\top X)\phi(X^\top W)\right)^{-1}K_W\right) = \lim_{\xi\to0^-}\mathrm{tr}\left(\left(\Phi\Phi^\top - \xi I\right)^{-1}K_W\right) = \lim_{\xi\to0^-}V_\xi,
$$

where the continuity and boundness of $V_\xi$ at $\xi = 0^-$ is guaranteed by Lemma 11 when $\gamma_2 = h/n \neq 1$. A ridge $\xi$ is added make use of (Louart et al., 2018, Theorem 1), which derived the asymptotic equivalent of the resolvent $\left(\Phi\Phi^\top - \xi I\right)^{-1}$. It follows that as $n,d,h\to\infty$,

$$
\begin{aligned}
&\mathrm{tr}\left(h^{-1}\left(\Phi\Phi^\top - \xi I\right)^{-1}\left(\frac{n}{h}\frac{K_W}{1+h^{-1}\mathrm{tr}\left(h\left(\Phi\Phi^\top - \xi I\right)^{-1}K_W\right)}\right) - h^{-1}I\right)\\
&\leq \frac{1}{h}\left\|\left(\Phi\Phi^\top - \xi I\right)^{-1} - \left(\frac{n}{h}\frac{K_W}{1+h^{-1}\mathrm{tr}\left(h\left(\Phi\Phi^\top - \xi I\right)^{-1}K_W\right)}\right)^{-1}\right\|_F\\
&\quad\cdot\left\|\frac{n}{h}\frac{K_W}{1+h^{-1}\mathrm{tr}\left(h\left(\Phi\Phi^\top - \xi I\right)^{-1}K_W\right)}\right\|_2\\
&\leq \frac{1}{h}O(n^{-1/2+\epsilon})O(n^{1/2}) \to 0.
\end{aligned}
\tag{47}
$$

where we have used the inequality $\mathrm{tr}(AB) \leq \|A\|_F\|B\|_2$, and equivalently by taking $\xi\to0$ we get

$$
\lim_{\xi\to0}\lim_{n,d,h\to\infty}\frac{n}{h}\frac{\mathrm{tr}\left(\left(\Phi\Phi^\top - \xi I\right)^{-1}K_W\right)}{1+\mathrm{tr}\left(\left(\Phi\Phi^\top - \xi I\right)^{-1}K_W\right)} = 1.
\tag{48}
$$

Therefore $\lim_{\xi \to 0} \lim_{n,d,h \to \infty} n/h \cdot \text{tr}\left(\left(\Phi\Phi^\top - \xi I\right)^{-1} K_W\right) = \gamma_2/(1-\gamma_2)$. Note that $\partial V_\xi/\partial \xi = \text{tr}\left(\xi(\Phi\Phi^\top - \xi I)^{-2} K_W\right)$ is bounded around the neighbourhood of $\xi = 0$, hence following the same argument as (Hastie et al., 2019, Theorem 4), we exchange the limit of $\xi \to 0$ and $n,d,h \to 0$:

$$V \to \frac{\gamma_2}{1-\gamma_2}. \tag{49}$$

**Case 2.** $h > n$. Techniques used in the current proof are largely borrowed from Hastie et al. (2019); Cheng and Singer (2013), and we include the full proof for completeness. It should be noted that compared to Hastie et al. (2019) we handle the non-zero expectation of the nonlinearity under Gaussian distribution, i.e. the off-diagonal entries of the kernel matrix is no longer zero-centered. For simplicity we mainly adhere to the notations in Hastie et al. (2019).

We briefly summarizes the procedure for deriving $V$. Instead of calculating the variance directly, we analyze a modified quantity $V_\xi$ and then take $\xi \to 0$, which can be connected to the trace of the resolvent of matrix $\tilde{A}$ defined in (56); this translates the calculation of $V_\xi$ into the calculation of the Stieltjes transform of $\tilde{A}$ (57), (61), (62).

## C.5   Deriving the variance $V$ for $h > n$

**Step 1. An equivalent expression.**   For notational simplicity we omit the magnitude $\sigma$:

$$V = \text{tr}\left(\phi(W^\top X)\left(\phi(X^\top W)\phi(W^\top X)\right)^{-2}\phi(X^\top W)K_W\right). \tag{50}$$

and due to the same continuity argument as in Case 1 we have

$$V = \lim_{\xi \to 0} \frac{1}{n}\left[\text{tr}\left(S(S^\top S - \xi I_n)^{-2}S^\top K_W\right)\right] = \lim_{\xi \to 0} V_\xi.$$

where $S = \phi(W^\top X)/\sqrt{n} = \Phi/\sqrt{n}$, $\xi \in \mathbb{C}$ and $\Im\xi > 0$ or $\xi < 0$.

We decompose the normalized feature matrix $S = \phi(W^\top X)/\sqrt{n}$ as $S = U\Sigma V^\top$, where $\Sigma = \text{diag}_{h \times n}(\phi_1, \cdots, \phi_n) \in \mathbb{R}^{h \times n}$ is a tall diagonal matrix, and $U = [\boldsymbol{u}_1, \cdots, \boldsymbol{u}_h] \in \mathbb{R}^{h \times h}$ is the set of orthogonal eigenvectors of $SS^\top = \phi(W^\top X)\phi(X^\top W)/n$, and $V \in \mathbb{R}^{n \times n}$ is the set of orthogonal eigenvectors of $S^\top S = \phi(X^\top W)\phi(W^\top X)/n$. Now the variance can be written as

$$V_\xi = \frac{1}{n}\text{tr}\left(U\Sigma(\Sigma^\top \Sigma + \xi I_n)^{-2}\Sigma U^\top K_W\right). \tag{51}$$

By the same argument as in Lemma 13 of Hastie et al. (2019) one can show that when $n,d,h \to \infty$

$$V_\xi = \frac{1}{n}\left[\text{tr}\left(U\Sigma(\Sigma^\top \Sigma + \xi I_n)^{-2}\Sigma U^\top K_W\right)\right] \to \frac{1}{n}\left[\text{tr}\left(U\Sigma(\Sigma^\top \Sigma + \xi I_n)^{-2}\Sigma U^\top \tilde{K}_W\right)\right], \tag{52}$$

in which $\tilde{K}_W$ is the approxmation of $K_W$ defined in Lemma 16. Writing the trace explicitly (denote eigenvalues $\lambda_i = \phi_i^2$, and $\phi_{n+1} = \cdots = \phi_h = 0$), we have

$$V_\xi \to \frac{1}{n}\text{tr}\left(U\Sigma(\Sigma^\top \Sigma + \xi I_n)^{-2}\Sigma U^\top \tilde{K}_W\right) = \gamma_2 \frac{1}{h}\sum_{i=1}^{h} \frac{\lambda_i}{(\lambda_i + \xi)^2}\boldsymbol{u}_i^\top \tilde{K}_W \boldsymbol{u}_i. \tag{53}$$

Since the positive support of spectrum $\lambda$ is lower bounded and the density at 0 is $1 - \gamma_2^{-1}$, we have

$$V_\xi \to \gamma_2 \frac{1}{h}\lim_{h,d,n \to \infty}\sum_{i=1}^{h} \frac{\lambda_i}{(\lambda_i + \xi)^2}\boldsymbol{u}_i^\top \tilde{K}_W \boldsymbol{u}_i = \gamma_2 \int \frac{\lambda}{(\lambda + \xi)^2}\mu_\infty(d\lambda) = \gamma_2 \int_{\lambda > \rho} \frac{\lambda}{(\lambda + \xi)^2}\mu_\infty^+(d\lambda), \tag{54}$$

where we define $\mu_n(x) = \frac{1}{h}\sum_{i=1}^{h} \delta_{\lambda_i}(x)\boldsymbol{u}_i^\top \tilde{K}_W \boldsymbol{u}_i$ and its positive part $\mu_n^+(x)$. Hence we have

$$V = \lim_{\xi \to 0} V_\xi = \lim_{\xi \to 0} \gamma_2 \int_{\lambda > \rho} \frac{\lambda}{(\lambda + \xi)^2}\mu_\infty^+(d\lambda) = \gamma_2 \int_{\lambda > \rho} \frac{1}{\lambda}\mu_\infty^+(d\lambda). \tag{55}$$

We define the following matrix $\tilde{A}_n(\rho, \varsigma, \tau) \in \mathbb{R}^{N \times N}$ where $N = n + h$:

$$\tilde{A}_n(\rho, \varsigma, \tau) = \begin{bmatrix} \rho I_h + \varsigma \mathbf{1}_h \mathbf{1}_h^\top + \tau Q & S \\ S^\top & 0_n \end{bmatrix}. \tag{56}$$

And denote the Stieltjes transform of $\tilde{A}_n$ as

$$\tilde{m}_n(\xi, \rho, \varsigma, \tau) = \frac{1}{n} \text{tr} \left( (\tilde{A}_n(\rho, \varsigma, \tau) - \xi I_N)^{-1} \right). \tag{57}$$

Then following the definition of $\tilde{K}_W$ one can show that

$$\tilde{m}_n(\xi, rx, sx, tx) = \frac{1}{n} \text{tr} \left( \begin{bmatrix} \tilde{K}_W x - \xi I_h & S \\ S^\top & -I_n \end{bmatrix}^{-1} \right), \tag{58}$$

and taking matrix derivative gives

$$-\frac{\partial}{\partial x} \tilde{m}_n(\xi, rx, sx, tx) \Big|_{x=0} = \frac{1}{n} \text{tr} \left( \begin{bmatrix} -\xi I_h & S \\ S^\top & -I_n \end{bmatrix}^{-1} \begin{bmatrix} \tilde{K}_W & 0 \\ 0 & 0 \end{bmatrix} \begin{bmatrix} -\xi I_h & S \\ S^\top & -I_n \end{bmatrix}^{-1} \right)$$

$$= \frac{1}{n} \text{tr} \left( \begin{bmatrix} \xi^2 I_h + SS^\top & 0 \\ 0 & I_n + S^\top S \end{bmatrix}^{-1} \begin{bmatrix} \tilde{K}_W & 0 \\ 0 & 0 \end{bmatrix} \right)$$

$$= \frac{1}{n} \text{tr} \left( U(\Sigma\Sigma^\top + \xi^2 I_h)^{-1} U^\top \tilde{K}_W \right) = \frac{1}{n} \sum_{i=1}^{h} \frac{1}{\lambda + \xi^2} \mathbf{u}_i^\top \tilde{K}_W \mathbf{u}_i. \tag{59}$$

Denote the limit $\tilde{m}(\xi, \rho, \varsigma, \tau) = \lim_{n,h,d \to \infty} \tilde{m}_n(\xi, \rho, \varsigma, \tau)$, and the derivative is given as

$$-\frac{\partial}{\partial x} \tilde{m}(\xi, rx, sx, tx) \Big|_{x=0} = \lim_{h,d,n \to \infty} \frac{1}{n} \sum_{i=1}^{h} \frac{1}{\lambda + \xi^2} \mathbf{u}_i^\top \tilde{K}_W \mathbf{u}_i$$

$$= \gamma_2 \int_{\lambda \geq 0} \frac{1}{\lambda + \xi^2} \mu_\infty(d\lambda) = \frac{\gamma_2 - 1}{\xi^2} + \gamma_2 \int_{\lambda > \rho} \frac{1}{\lambda + \xi^2} \mu_\infty^+(d\lambda). \tag{60}$$

For simplicity we define the following function on $\xi$:

$$q(\xi) = -\frac{\partial}{\partial x} \tilde{m}(\xi, rx, sx, tx) \Big|_{x=0}, \tag{61}$$

also denote $q_+(\xi) = q(\xi) - \frac{\gamma_2 - 1}{\xi^2} = \gamma_2 \int_{\lambda > \rho} \frac{1}{\lambda + \xi^2} \mu_\infty^+(d\lambda)$. Comparing with (55) yields:

$$V = \lim_{\xi \to 0} q_+(\xi). \tag{62}$$

**Step 2. Calculating $q(\xi)$ and $m_n(\xi, \rho, \varsigma, \tau)$**   This subsection aims to calculate $\tilde{m}(\xi, \rho, \varsigma, \tau)$ and $q(\xi) = -\tilde{m}_x'(\xi, rx, sx, tx)|_{x=0}$, from which the variance can be computed from (57)(61)(62).

We define $A_n$ by subtracting the off-diagonal entries of the upper-left block of $\tilde{A}_n$:

$$A_n(\rho, \tau) = \begin{bmatrix} \rho I_h + \tau Q & S \\ S^\top & 0_n \end{bmatrix}, \tag{63}$$

where $S = \tilde{S} - a_0 I_{p \times n}$, i.e. $S_{ik} = \phi(\mathbf{w}_i^\top \mathbf{x}_k) - a_0 = \varphi(\mathbf{w}_i^\top \mathbf{x}_k)$, $a_0 = \mathbb{E}[\phi(x)]$. The Stieltjes transform of $A_n$ given by $m_n(\xi, \rho, \tau) = \frac{1}{n} \text{tr} \left( (A_n(\rho, \tau) - \xi I_N)^{-1} \right)$. The following Lemma shows that $\tilde{m}_n$ and $m_n$ have the same limit:

**Lemma 12.** *when $n \to \infty$ and for $\Im \xi > 0$ or $\xi < 0$, we have $m_n(\xi, \rho, \tau) \to \tilde{m}_n(\xi, \rho, \varsigma, \tau)$.*

**Proof.** By definition of $\bar{A}_n$ and $A_n$,

$$\tilde{A}_n(\rho, \varsigma, \tau) - A_n(\rho, \tau) = \begin{bmatrix} \mathbf{1}_h & 0_h \\ 0_n & \mathbf{1}_n \end{bmatrix} \begin{bmatrix} \xi & a_0 \\ a_0 & 0 \end{bmatrix} \begin{bmatrix} \mathbf{1}_h & 0_h \\ 0_n & \mathbf{1}_n \end{bmatrix}^\top \tag{64}$$

which is a rank-2 matrix. By theorem A.43 from Bai and Silverstein (2010), which characterizes the effect of finite-rank perturbation on the e.s.d. of random matrices:

$$\sup_x |F^{\tilde{A}_n}(x) - F^{A_n}(x)| \leq O\left(n^{-1}\right), \tag{65}$$

where $F^M$ is the empirical spectral distribution of $M \in \mathbb{R}^{n \times n}$. The claim follows from the Stieltjes continuity theorem (e.g. Section 2.4 in Tao (2012)). $\qquad\square$

To calculate the Stieltjes transform $m_n$, we take advantage of the block structure of $A_n$:

$$m_{1,n}(\xi, \rho, \tau) = \frac{1}{p}\text{tr}\left((A_n(\rho, \tau) - \xi I_N)^{-1}_{[1..p, 1..p]}\right), \tag{66}$$

$$m_{2,n}(\xi, \rho, \tau) = \frac{1}{n}\text{tr}\left((A_n(\rho, \tau) - \xi I_N)^{-1}_{[p+1..p+n, p+1..p+n]}\right). \tag{67}$$

One can observe that $m_n(\xi, \rho, \tau) = \gamma_2 m_{1,n}(\xi, \rho, \tau) + m_{2,n}(\xi, \rho, \tau)$.

In the following equations we omit the subscript $n$, as well as dependency on $\rho, \varsigma, \tau$. Following Hastie et al. (2019), we rewrite the matrix $A_n = A = \begin{bmatrix} A_* & \boldsymbol{a} \\ \boldsymbol{a}^\top & 0 \end{bmatrix}$, where $A^*$ is a $(N-1) \times (N-1)$ matrix with last column and row of $A$ removed and $\boldsymbol{s}$ the activation vector:

$$A_* = \begin{bmatrix} \rho I_h + \tau Q & S_* \\ S_*^\top & 0_{n-1} \end{bmatrix}; \quad \boldsymbol{a}^\top = [\phi(W^\top \boldsymbol{x}_n)^\top \quad \boldsymbol{0}_{n-1}^\top] = [\boldsymbol{s}^\top \quad \boldsymbol{0}_{n-1}^\top]. \tag{68}$$

Hence by the block matrix inverse formula

$$(A - \xi I_N)^{-1} = \begin{bmatrix} * & * \\ * & [-\xi - \boldsymbol{a}^\top(A_* - \xi I_{N-1})^{-1}\boldsymbol{a}]^{-1} \end{bmatrix}. \tag{69}$$

Plugging this back in the Stieltjes transform we have

$$m_{2,n}(\xi, \rho, \tau) = \frac{1}{n}\text{tr}\left((A_n(\rho, \varsigma, \tau) - \xi I_N)^{-1}_{[h+1, N]}\right) = \mathbb{E}_{\boldsymbol{a}}\left[(A_n(\rho, \tau) - \xi I_N)^{-1}\right]_{NN}$$

$$= \mathbb{E}_{\boldsymbol{a}}\left[\left(-\xi - \boldsymbol{a}^\top(A^* - \xi I_{N-1})^{-1}\boldsymbol{a}\right)^{-1}\right]. \tag{70}$$

To obtain an asymptotic description of $m_{2,n}$, we perform the orthonormal decomposition on the nonlinearity $\varphi$ introduced in Cheng and Singer (2013): $\varphi(x) = a_1 x + \varphi_\perp(x)$, where $a_1 = \mathbb{E}_{x \sim \mathcal{N}(0,1)}[x\varphi(x)]$. For each $\boldsymbol{w}_i(1 \leq i \leq h)$, we perform the following orthonormal decomposition (along the direction of $\boldsymbol{x}_n$ and the direction of $\tilde{\boldsymbol{w}}_i$ perpendicular to $\boldsymbol{x}_n$):

$$\boldsymbol{w}_i = \underbrace{\boldsymbol{w}_i^\top \boldsymbol{x}_n}_{\eta_i} \frac{\boldsymbol{x}_n}{\|\boldsymbol{x}_n\|} + \tilde{\boldsymbol{w}}_i = \eta_i \frac{\boldsymbol{x}_n}{\|\boldsymbol{x}_n\|} + \tilde{\boldsymbol{w}}_i. \tag{71}$$

We can thus simplify the activation vector $\boldsymbol{a}$ as

$$\boldsymbol{a}^\top = \begin{bmatrix} \frac{1}{\sqrt{n}}\varphi(\|\boldsymbol{x}_n\|\eta_1) & \cdots & \frac{1}{\sqrt{n}}\varphi(\|\boldsymbol{x}_n\|\eta_h) & \underbrace{0 \quad \cdots \quad 0}_{n-1} \end{bmatrix}$$

$$= \underbrace{\begin{bmatrix} \frac{1}{\sqrt{n}}a_1\|\boldsymbol{x}_n\|\eta_1 & \cdots & \frac{1}{\sqrt{n}}a_1\|\boldsymbol{x}_n\|\eta_h & \underbrace{0 \quad \cdots \quad 0}_{n-1} \end{bmatrix}}_{\boldsymbol{a}_1^\top = [\boldsymbol{s}_1^\top, \boldsymbol{0}^\top]}$$

$$+ \underbrace{\begin{bmatrix} \frac{1}{\sqrt{n}}\varphi_\perp(\|\boldsymbol{x}_n\|\eta_1) & \cdots & \frac{1}{\sqrt{n}}\varphi_\perp(\|\boldsymbol{x}_n\|\eta_h) & \underbrace{0 \quad \cdots \quad 0}_{n-1} \end{bmatrix}}_{\boldsymbol{a}_2^\top = [\boldsymbol{s}_2^\top, \boldsymbol{0}^\top]}. \tag{72}$$

and for $1 \le i \ne j \le h$, $1 \le k \le n-1$,

$$Q_{ij} = \left( \eta_i \frac{\boldsymbol{x}_n}{\|\boldsymbol{x}_n\|} + \tilde{\boldsymbol{w}}_i \right)^\top \left( \eta_j \frac{\boldsymbol{x}_n}{\|\boldsymbol{x}_n\|} + \tilde{\boldsymbol{w}}_j \right) = \eta_i \eta_j + \underbrace{\tilde{\boldsymbol{w}}_i^\top \tilde{\boldsymbol{w}}_j}_{\tilde{Q}_{ij}}. \tag{73}$$

Similarly we decompose the activation function in $S$,

$$S_{ik} = \frac{1}{\sqrt{n}} \varphi \left( \eta_i \frac{\boldsymbol{x}_n^\top \boldsymbol{x}_k}{\|\boldsymbol{x}_n\|} + \tilde{\boldsymbol{w}}_i^\top \boldsymbol{x}_k \right) = \frac{1}{\sqrt{n}} a_1 \eta_i \frac{\boldsymbol{x}_n^\top \boldsymbol{x}_k}{\|\boldsymbol{x}_n\|} + \frac{1}{\sqrt{n}} a_1 \tilde{\boldsymbol{w}}_i^\top \boldsymbol{x}_k + \frac{1}{\sqrt{n}} \varphi_\perp \left( \eta_i \frac{\boldsymbol{x}_n^\top \boldsymbol{x}_k}{\|\boldsymbol{x}_n\|} + \tilde{\boldsymbol{w}}_i^\top \boldsymbol{x}_k \right)$$

$$= \underbrace{\frac{1}{\sqrt{n}} \varphi(\tilde{\boldsymbol{w}}_i^\top \boldsymbol{x}_k)}_{\tilde{S}_{ik}} + \underbrace{\frac{1}{\sqrt{n}} a_1 \eta_i \frac{\boldsymbol{x}_n^\top \boldsymbol{x}_k}{\|\boldsymbol{x}_n\|}}_{a_1 \eta_i u_k} + \underbrace{\frac{1}{\sqrt{n}} \left[ \varphi_\perp \left( \eta_i \frac{\boldsymbol{x}_n^\top \boldsymbol{x}_k}{\|\boldsymbol{x}_n\|} + \tilde{\boldsymbol{w}}_i^\top \boldsymbol{x}_k \right) - \varphi_\perp(\tilde{\boldsymbol{w}}_i^\top \boldsymbol{x}_k) \right]}_{E_{ik}}. \tag{74}$$

We thus have an equivalent expression of matrix $A_*$

$$A_* = \begin{bmatrix} \rho I_h + \tau \tilde{Q} & \tilde{S}_* \\ \tilde{S}_*^\top & 0_{n-1} \end{bmatrix} + \begin{bmatrix} t \boldsymbol{\eta} \boldsymbol{\eta}^\top & a_1 \boldsymbol{\eta} \boldsymbol{u}^\top \\ a_1 \boldsymbol{u} \boldsymbol{\eta}^\top & 0_{n-1} \end{bmatrix} + \begin{bmatrix} E_0 & E_1 \\ E_1^\top & 0_{n-1} \end{bmatrix}$$

$$= \underbrace{\begin{bmatrix} \rho I_h + \tau \tilde{Q} & \tilde{S}_* \\ \tilde{S}_*^\top & 0_{n-1} \end{bmatrix}}_{\tilde{A}_*} + \underbrace{\begin{bmatrix} \boldsymbol{\eta} & \boldsymbol{0}_h \\ \boldsymbol{0}_{n-1} & \boldsymbol{u} \end{bmatrix}}_{U} \underbrace{\begin{bmatrix} \tau & a_1 \\ a_1 & 0 \end{bmatrix}}_{C} \underbrace{\begin{bmatrix} \boldsymbol{\eta} & \boldsymbol{0}_h \\ \boldsymbol{0}_{n-1} & \boldsymbol{u} \end{bmatrix}^\top}_{U^\top} + \underbrace{\begin{bmatrix} E_0 & E_1 \\ E_1^\top & 0_{n-1} \end{bmatrix}}_{E}$$

$$= \tilde{A}_* + UCU^\top + E. \tag{75}$$

By argument similar to (Hastie et al., 2019, B.1.2), $E$ diminishes to 0 as $n \to \infty$ with respect to the Frobenius norm, therefore by the Woodbury's identity and the expression of $m_{2,n}$ in (70)

$$m_{2,n}(\xi, \rho, \tau) = \mathbb{E}_{\boldsymbol{a}} \left[ \left( -\xi - \boldsymbol{a}^\top (A_* - \xi I_{N-1})^{-1} \boldsymbol{a} \right)^{-1} \right]$$

$$\to \mathbb{E}_{\boldsymbol{a}} \left[ \left( -\xi - \boldsymbol{a}^\top (\tilde{A}_* - \xi I_{N-1} + UCU^\top)^{-1} \boldsymbol{a} \right)^{-1} \right]$$

$$= \mathbb{E}_{\boldsymbol{a}} \left[ \left( -\xi - \underbrace{\boldsymbol{a}^\top (\tilde{A}_* - \xi I_{N-1})^{-1} \boldsymbol{a}}_{u} + \right. \right.$$

$$\left. \left. \underbrace{\boldsymbol{a}^\top (\tilde{A}_* - \xi I_{N-1})^{-1} U}_{\boldsymbol{v}^\top} \underbrace{\left( C^{-1} + U^\top (\tilde{A}_* - \xi I_{N-1})^{-1} U \right)^{-1}}_{S} \underbrace{U(\tilde{A}_* - \xi I_{N-1})^{-1} \boldsymbol{a}}_{\boldsymbol{v}} \right)^{-1} \right]. \tag{76}$$

We bound each term $u, \boldsymbol{v}, S$ to compute $m_{2,n}$. For $u$

$$\mathbb{E}_{\boldsymbol{a}} u = \mathbb{E}_{\boldsymbol{a}} \left[ \boldsymbol{a}^\top (\tilde{A}_* - \xi I_{N-1})^{-1} \boldsymbol{a} \right] = \text{tr} \left( \mathbb{E}_{\boldsymbol{s}}[\boldsymbol{s}\boldsymbol{s}^\top](\tilde{A}_* - \xi I_{N-1})^{-1}_{[1..h,1..h]} \right) = b \gamma_2 m_{1,n}(\xi, \rho, \tau), \tag{77}$$

where $b = \mathbb{E}_{x \sim \mathcal{N}(0,1)}[\varphi(x)^2] = \mathbb{E}_{x \sim \mathcal{N}(0,1)}[(\phi(x) - \mathbb{E}\phi(x))^2] = r$. From a standard concentration of measure argument we have that as $n, h, d \to \infty$

$$u \to \mathbb{E}_{\boldsymbol{a}} u = r \gamma_2 m_{1,n}(\xi, \rho, \tau). \tag{78}$$

And for $\boldsymbol{v}$ (note that $U$ is dependent on $\boldsymbol{a}$)

$$\mathbb{E}_{\boldsymbol{a}} \boldsymbol{v}^\top = \mathbb{E}_{\boldsymbol{a}} \left[ \boldsymbol{a}^\top (\tilde{A}_* - \xi I_{N-1})^{-1} U \right] = \mathbb{E}_{\boldsymbol{a}} \left[ [\boldsymbol{s}^\top, 0_{n-1}^\top](\tilde{A}_* - \xi I_{N-1})^{-1} \begin{bmatrix} \boldsymbol{\eta} & \boldsymbol{0}_h \\ \boldsymbol{0}_{n-1} & \boldsymbol{u} \end{bmatrix} \right]$$

$$= \left[ \mathbb{E}_{\boldsymbol{s}}[\boldsymbol{s}^\top (\tilde{A}_* - \xi I_{N-1})^{-1}_{[1..h,1..h]} \boldsymbol{\eta}] \quad 0 \right] = \left[ \underbrace{\text{tr} \left( (\tilde{A}_* - \xi I_{N-1})^{-1}_{[1..h,1..h]} \mathbb{E}_{\boldsymbol{s}}[\boldsymbol{\eta} \boldsymbol{s}^\top] \right)}_{v} \quad 0 \right]. \tag{79}$$

where $v = \text{tr} \left( (\tilde{A}_* - \xi I_{N-1})^{-1}_{[1..h,1..h]} \mathbb{E}_{\boldsymbol{s}}[\boldsymbol{\eta} \boldsymbol{s}^\top] \right) = a_1 \sqrt{\gamma_2^2/\gamma_1} m_{1,n}(\xi, \rho, \tau)$. We thus have

$$\boldsymbol{v} \to \mathbb{E}_{\boldsymbol{a}} \boldsymbol{v} = \left[ a_1 \sqrt{\gamma_2^2/\gamma_1} m_{1,n}(\xi, \rho, \tau) \quad 0 \right]. \tag{80}$$

as $n, d, p \to \infty$. And finally for $S$,

$$
\begin{aligned}
\mathbb{E}_{\boldsymbol{a}} S^{-1} &= \mathbb{E}_{\boldsymbol{a}} \left[ C^{-1} + U^\top (\tilde{A}_* - \xi I_{N-1})^{-1} U \right] \\
&= \begin{bmatrix} \tau & a_1 \\ a_1 & 0 \end{bmatrix}^{-1} + \mathbb{E}_{\boldsymbol{a}} \left[ \begin{bmatrix} \boldsymbol{\eta} & \mathbf{0}_h \\ \mathbf{0}_{n-1} & \boldsymbol{u} \end{bmatrix}^\top (\tilde{A}_* - \xi I_{N-1})^{-1} \begin{bmatrix} \boldsymbol{\eta} & \mathbf{0}_h \\ \mathbf{0}_{n-1} & \boldsymbol{u} \end{bmatrix} \right] \\
&= \begin{bmatrix} 0 & 1/a_1 \\ 1/a_1 & -\tau/a_1^2 \end{bmatrix} + \mathbb{E}_{\boldsymbol{a}} \begin{bmatrix} \boldsymbol{\eta}^\top (\tilde{A}_* - \xi I_{N-1})^{-1}_{[1..h]} \boldsymbol{\eta} & 0 \\ 0 & \boldsymbol{u}^\top (\tilde{A}_* - \xi I_{N-1})^{-1}_{[h+1..h+n-1]} \boldsymbol{u} \end{bmatrix} \\
&= \begin{bmatrix} 0 & 1/a_1 \\ 1/a_1 & -\tau/a_1^2 \end{bmatrix} + \begin{bmatrix} \operatorname{tr}\left( (\tilde{A}_* - \xi I_{N-1})^{-1}_{[1..h]} \mathbb{E}_{\boldsymbol{a}}[\boldsymbol{\eta}\boldsymbol{\eta}^\top] \right) & 0 \\ 0 & \operatorname{tr}\left( (\tilde{A}_* - \xi I_{N-1})^{-1}_{[h+1..h+n-1]} \mathbb{E}_{\boldsymbol{a}}[\boldsymbol{u}\boldsymbol{u}^\top] \right) \end{bmatrix} \\
&= \begin{bmatrix} 0 & 1/a_1 \\ 1/a_1 & -\tau/a_1^2 \end{bmatrix} + \begin{bmatrix} \gamma_2/\gamma_1 m_{1,n}(\xi, \rho, \tau) & 0 \\ 0 & m_{2,n}(\xi, \rho, \tau) \end{bmatrix} \\
&= \begin{bmatrix} \gamma_1^{-1} \gamma_2 m_{1,n}(\xi, \rho, \tau) & 1/a_1 \\ 1/a_1 & m_{2,n}(\xi, \rho, \tau) - \tau/a^2 \end{bmatrix}.
\end{aligned}
\tag{81}
$$

And hence as $n, d, h \to \infty$,

$$
S^{-1} \to \mathbb{E}_{\boldsymbol{a}} S^{-1} = \begin{bmatrix} \gamma_1^{-1} \gamma_2 m_{1,n}(\xi, \rho, \tau) & 1/a_1 \\ 1/a_1 & m_{2,n}(\xi, \rho, \tau) - \tau/a^2 \end{bmatrix}.
\tag{82}
$$

Therefore by combining (78), (80), (82), we arrive at the following expression on $m_{2,n}$,

$$
\begin{aligned}
m_{2,n}(\xi, \rho, \tau) &\to \mathbb{E}_{\boldsymbol{a}} \left[ \left( -\xi - u + \boldsymbol{v}^\top S \boldsymbol{v} \right)^{-1} \right] \to \left( -\xi - u + \boldsymbol{v}^\top S \boldsymbol{v} \right)^{-1} \\
&\to \left( -\xi - r\gamma_2 m_{1,n} + \left( a_1 \sqrt{\gamma_2^2/\gamma_1} m_{1,n} \right)^2 \begin{bmatrix} \gamma_1^{-1} \gamma_2 m_{1,n}(\xi, \rho, \tau) & 1/a_1 \\ 1/a_1 & m_{2,n}(\xi, \rho, \tau) - \tau/a^2 \end{bmatrix}^{-1}_{[1,1]} \right)^{-1} \\
&= \left( -\xi - r\gamma_2 m_{1,n} + \frac{\gamma_2 a_1^2 m_{1,n}^2 (a_1^2 m_{2,n} - \tau)}{m_{1,n}(a_1^2 m_{2,n} - \tau) - \gamma_1 \gamma_2^{-1}} \right)^{-1}.
\end{aligned}
\tag{83}
$$

Similarly we can calculate $m_{1,n}(\xi, \rho, \tau)$ as

$$
m_{1,n}(\xi, \rho, \tau) \to
$$

$$
\left( -\xi - \rho - \gamma_1^{-1} \gamma_2 \tau^2 m_{1,n} - r m_{2,n} + \frac{\tau^2 \gamma_1^{-1} \gamma_2 m_{1,n}^2 (a_1^2 m_2 - \tau) - 2\tau a_1^2 m_{1,n} m_{2,n} + a_1^4 m_{1,n} m_{2,n}^2}{m_{1,n}(a_1^2 m_{2,n} - \tau) - \gamma_1 \gamma_2^{-1}} \right)^{-1}.
\tag{84}
$$

Uniqueness in (84)(83) follows from (Hastie et al., 2019, Sec B.1) and is omitted. $\qquad\square$

## C.6 Proof of Corollary 5

In this section we take the limit $\gamma_1 \to \infty$. In this case (8), (9) simplify to

$$
m_2 = (-\xi - r\gamma_2 m_1)^{-1},
\tag{85}
$$

$$
m_1 = (-\xi - \rho - \gamma_2 \tau^2 m_1 - r m_2)^{-1}.
\tag{86}
$$

Recall that $m(\xi, \rho, \tau) = \gamma_2 m_1(\xi, \rho, \tau) + m_2(\xi, \rho, \tau)$. By taking the derivative we have

$$
\begin{aligned}
-q(\xi) &= \frac{\partial}{\partial x} m(\xi, rx, tx)\Big|_{x=0} = r\frac{\partial}{\partial \rho} m(\xi, \rho, 0)\Big|_{\rho=0} + t\frac{\partial}{\partial \tau} m(\xi, 0, \tau)\Big|_{\tau=0} \\
&= r\gamma_2 \frac{\partial}{\partial \rho} m_1(\xi, \rho, 0)\Big|_{\rho=0} + r\frac{\partial}{\partial \rho} m_2(\xi, \rho, 0)\Big|_{\rho=0} + t\gamma_2 \frac{\partial}{\partial \tau} m_1(\xi, 0, \tau)\Big|_{\tau=0} + t\frac{\partial}{\partial \tau} m_2(\xi, 0, \tau)\Big|_{\tau=0}.
\end{aligned}
\tag{87}
$$

Observe that (85), (86) constitutes a set of implicit functions. Thus differentiating the two functions with respect to $\tau, \rho$ and then substitute by $\rho = \tau = 0$ gives

$$q(\xi) = \frac{\left(r(\gamma_2 - 1) - \xi^2\right)\left(\sqrt{\left(r(\gamma_2 - 1) + \xi^2\right)^2 - 4r\gamma_2\xi^2} + r(\gamma_2 - 1) + \xi^2\right)}{2\xi^2\sqrt{\left(r(\gamma_2 - 1) + \xi^2\right)^2 - 4r\gamma_2\xi^2}}. \tag{88}$$

Hence by (62) we obtain the asymptotic variance:

$$V_{(\gamma_1 \to \infty)} = \lim_{\xi \to 0} q_+(\xi) = \lim_{\xi \to 0} \left(q(\xi) - \frac{\gamma_2 - 1}{\xi^2}\right) = \frac{1}{\gamma_2 - 1}. \tag{89}$$

Combining the case where $\gamma_2 < 1$ in Theorem 4 completes the proof. $\square$

***Remark.*** For $\phi(x) = \text{ReLU}(x)$, $c_1 = 1/2 - 1/(2\pi)$, $c_2 = 1/4$. For $\phi(x) = \text{SoftPlus}(x) = \log(1 + e^x)$, numerical integration yields $c_1 \approx 0.2715$, $c_2 = 1/4$.

### C.7 PROOF OF COROLLARY 6

#### C.7.1 UNBOUNDED BIAS WHEN $h = n$

From the bias-variance decomposition (7), the bias $B$ is written as $B = \frac{r^2}{d}\text{tr}\left(Q_1 + Q_2 + I_d\right)$, where

$$Q_1 = X[\phi(W^\top X)]^\dagger K_W[\phi(X^\top W)]^\dagger X^\top; \quad Q_2 = X[\phi(W^\top X)]^\dagger W^\top. \tag{90}$$

When $h = n$, due to the nonlinearity of $\phi$, we have $\phi(W^\top X)$ is full rank a.s., and hence $[\phi(X^\top W)]^\dagger = [\phi(X^\top W)]^{-1}$. We have the following bound for $Q_1$

$$\frac{1}{d}\text{tr}\left(Q_1\right) = \frac{1}{d}\text{tr}\left(X[\phi(W^\top X)]^\dagger K_W[\phi(X^\top W)]^\dagger X^\top\right) = \frac{1}{d}\text{tr}\left(K_W[\phi(X^\top W)]^{-1}X^\top X[\phi(W^\top X)]^{-1}\right)$$

$$\geq \frac{1}{d}\lambda_{\min}(K_W)\text{tr}\left([\phi(X^\top W)]^{-1}X^\top X[\phi(W^\top X)]^{-1}\right) = \frac{\lambda_{\min}(K_W)}{n}\text{tr}\left((SS^\top)^{-1} \cdot \frac{1}{d}X^\top X\right).$$

Since $W$ and $X/\sqrt{d}$ are $\mathbb{R}^{d \times n} = \mathbb{R}^{d \times p}$ follows the same distribution where each entry i.i.d. $\mathcal{N}(0, 1/d)$,

$$\frac{1}{d\lambda_{\min}(K_W)}\text{tr}\left(Q_1\right) \geq \frac{1}{n}\text{tr}\left((SS^\top)^{-1} \cdot \frac{1}{d}X^\top X\right) \sim \frac{1}{n}\text{tr}\left((S^\top S)^{-1} \cdot W^\top W\right)$$

$$= \frac{1}{n}\text{tr}\left((S^\top S)^{-1} \cdot (I + Q)\right) = \lim_{\xi \to 0} -\frac{\partial}{\partial x}\tilde{m}_n(\xi, x, 0, x) \to \infty, \tag{91}$$

where in Section C.5 we have showed (91) is unbounded when $n \to \infty$. Moreover, by (154) and Weyl's theorem we have $\lambda_{\min}(K_W) = O(1)$, and thus $d^{-1}\text{tr}\left(Q_1\right)$ is unbounded. For $d^{-1}\text{tr}\left(Q_2\right)$,

$$\frac{1}{d}\text{tr}\left(Q_2\right) = \frac{1}{d}\text{tr}\left(W^\top X[\phi(W^\top X)]^{-1}\right) \leq \frac{1}{d}\lambda_{\max}(\phi(W^\top X)^{-1})\text{tr}\left(W^\top X\right) = O(1). \tag{92}$$

To sum up, for $n \to \infty$ and $\gamma_2 \to 1$ we have $B \to \infty$. $\square$

#### C.7.2 BOUNDED BIAS WHEN $\gamma_2 > 1$

Since $h > n$, the two terms in the expression of the bias can be written as

$$Q_1 = X\left(\phi(X^\top W)\phi(W^\top X)\right)^{-1}\phi(X^\top W)K_W\phi(W^\top X)\left(\phi(X^\top W)\phi(W^\top X)\right)^{-1}X^\top, \tag{93}$$

$$Q_2 = X\left(\phi(X^\top W)\phi(W^\top X)\right)^{-1}\phi(X^\top W)W^\top. \tag{94}$$

Therefore we have

$$\frac{2}{d}\text{tr}\left(Q_1\right) = 2\text{tr}\left(\frac{X^\top X}{d}\left(\phi(X^\top W)\phi(W^\top X)\right)^{-1}\phi(X^\top W)K_W\phi(W^\top X)\left(\phi(X^\top W)\phi(W^\top X)\right)^{-1}\right)$$

$$\leq 2\lambda_{\max}\left(\frac{X^\top X}{d}\right)\mathrm{tr}\left(\left(\phi(X^\top W)\phi(W^\top X)\right)^{-1}\phi(X^\top W)K_W\phi(W^\top X)\left(\phi(X^\top W)\phi(W^\top X)\right)^{-1}\right)$$

$$= O(1)\cdot\mathrm{tr}\left(\phi(W^\top X)\left(\phi(X^\top W)\phi(W^\top X)\right)^{-2}\phi(X^\top W)K_W\right)$$

$$\leq O(1)\cdot\lambda_{\max}\left(\phi(W^\top X)\left(\phi(X^\top W)\phi(W^\top X)\right)^{-2}\phi(X^\top W)\right)\cdot\mathrm{tr}\left(K_W\right)$$

$$= O(1)\cdot\sigma_{\min}^{-2}(\phi(X^\top W))\mathrm{tr}\left(K_W\right) = O(1)\cdot O(n^{-1})\cdot O(n) = O(1), \tag{95}$$

in which we used Lemma 11 and $\mathrm{tr}\left(AB\right)\leq\lambda_{\max}(A)\mathrm{tr}\left(B\right)$ for positive semi-definite $A, B$. Similarly

$$\frac{2}{d}\mathrm{tr}\left(Q_2\right) = 2\mathrm{tr}\left(\left(\phi(X^\top W)\phi(W^\top X)\right)^{-1}\phi(X^\top W)\cdot\frac{1}{d}W^\top X\right)$$

$$\leq\mathrm{tr}\left(\left(\left(\phi(X^\top W)\phi(W^\top X)\right)^{-1}\phi(X^\top W)\right)(\ldots)^\top\right) + \mathrm{tr}\left(d^{-2}X^\top WW^\top X\right)$$

$$\leq n\cdot\sigma_{\min}\left(\phi(X^\top W)\right)^{-2} + \frac{1}{d}\lambda_{\max}\left(\frac{1}{d}XX^\top\right)\mathrm{tr}\left(WW^\top\right) = O(1), \tag{96}$$

where we applied $\mathrm{tr}\left(AB\right)\leq\left(\mathrm{tr}\left(A^\top A\right)+\mathrm{tr}\left(B^\top B\right)\right)/2$. We therefore conclude that $B$ is bounded when $h > n$ and $h, n \to \infty$. $\qquad\square$

***Remark***. Concurrent to this work, Mei and Montanari (2019) provides a complete characterization of the bias term and confirms our observations above.

## C.8    PROOF OF THEOREM 7

For simplicity we assume $n, d, h$ to be even and let $d_0 = d/2$, $n_0 = n/2$ and $h_0 = h/2$. Since the second layer is fixed $a_i \sim \mathrm{Unif}\{-1/\sqrt{h}, 1/\sqrt{h}\}$, we let $a_i = 1/\sqrt{h}$ and $a_{i+h_0} = -1/\sqrt{h}$ for all $1 \leq i \leq h_0$. We therefore write $\boldsymbol{a}^\top = h^{-1/2}[\mathbf{1}_{h_0}, -\mathbf{1}_{h_0}]^\top$ and $W = [W_+, W_-]$:

$$f(\boldsymbol{x}; W_-, W_+) = \boldsymbol{a}^\top\phi(W^\top\boldsymbol{x}) = \frac{1}{\sqrt{h}}\mathbf{1}^\top\phi(W_+^\top\boldsymbol{x}) - \frac{1}{\sqrt{h}}\mathbf{1}^\top\phi(W_-^\top\boldsymbol{x}). \tag{97}$$

The empirical risk can thus be written as

$$L(X; W_+, W_-) = \frac{1}{n}\sum_{\boldsymbol{x}\in X}L(\boldsymbol{x}; W_+, W_-) = \frac{1}{n}\sum_{\boldsymbol{x}\in X}\left[y - \frac{1}{\sqrt{h}}\mathbf{1}^\top\phi(W_+^\top\boldsymbol{x}) + \frac{1}{\sqrt{h}}\mathbf{1}^\top\phi(W_-^\top\boldsymbol{x})\right]^2. \tag{98}$$

### C.8.1    DEFINING GRADIENT FLOWS

In this section we define three gradient flows and show that the three flows are similar in some sense. ***GF-Original*** is the original gradient flow (11), i.e.

$$\frac{\partial W_+^O}{\partial t} = \frac{1}{2n_0}\sum_{i=1}^{2n_0}\left[\frac{1}{\sqrt{h}}\left(y_i - \frac{1}{\sqrt{h}}\mathbf{1}^\top\phi(W_+^{O\top}\boldsymbol{x}_i) + \frac{1}{\sqrt{h}}\mathbf{1}^\top\phi(W_-^{O\top}\boldsymbol{x}_i)\right)\boldsymbol{x}_i\phi'(\boldsymbol{x}_i^\top W_+^O)\right] \tag{99}$$

$$= \frac{1}{2n_0}\left[X(\boldsymbol{y}-\boldsymbol{y}^O(t))\frac{1}{\sqrt{h}}\mathbf{1}^\top\circ\phi'(XW_+^O)\right], \tag{100}$$

starting from the vanishing initialization $\boldsymbol{w}_i^O(0) \sim \mathcal{N}(\mathbf{0}, I/dh^{1+\epsilon})$. Note that the gradient for the negative part $W_-^O$ can be similarly defined.

We now define the flow under the same objective but from exact zero initialization $\boldsymbol{w}_i^D(0) = \mathbf{0}$ termed ***GF-Double***. Due to zero initialization, a basic observation is that the solution $[W_+^D, W_-^D]$ is

at most rank-2, and more precisely, the parameters in the flow takes the form of $W_{\pm}^D(t) = \boldsymbol{w}_{\pm}^D(t)\boldsymbol{1}^\top$ where $\boldsymbol{w}_{\pm}^D(t)$ admits the following dynamics:

$$\frac{\partial \boldsymbol{w}_+^D}{\partial t} = \boldsymbol{g}_+^D(\boldsymbol{w}_+^D) = \frac{1}{2n_0} \sum_{i=1}^{2n_0} \left[ \frac{1}{\sqrt{h}} \left( y_i - \sqrt{h}\phi(\boldsymbol{w}_+^{D\top}\boldsymbol{x}_i) + \sqrt{h}\phi(\boldsymbol{w}_-^{D\top}\boldsymbol{x}_i) \right) \phi'(\boldsymbol{w}_+^{D\top}\boldsymbol{x}_i)\boldsymbol{x}_i \right]. \tag{101}$$

Lastly, we define the **GF-Single** with solution denoted as $\boldsymbol{w}_{\pm} = \boldsymbol{w}_{\pm}^S(t)$ :

$$\frac{\partial \boldsymbol{w}_+^S}{\partial t} = \boldsymbol{g}_+^S(\boldsymbol{w}_+^S) = \frac{1}{2n_0} \sum_{i=1}^{2n_0} \left[ \frac{1}{\sqrt{h}} \left( y_i - \sqrt{h}\phi'(0)\boldsymbol{w}_+^{S\top}\boldsymbol{x}_i + \sqrt{h}\phi'(0)\boldsymbol{w}_-^{1\top}\boldsymbol{x}_i \right) \phi'(0)\boldsymbol{x}_i \right]. \tag{102}$$

from zero initialization $\boldsymbol{w}_{\pm}^D(0) = \boldsymbol{0}$. This can be seen as replacing the nonlinearity $\phi$ with its first-order Taylor expansion at the origin.

### C.8.2 FROM GF-DOUBLE TO GF-SINGLE

**Step 1. Solution of GF-single.** Among the three flows defined above, only GF-single an explicit form at any time $t$. Specifically, the solution can be written as:

$$\boldsymbol{w}_+^S(t) = -\boldsymbol{w}_-^S(t) = \frac{1}{2\sqrt{h}} \left( I - e^{-\frac{\phi'(0)^2}{n_0}XX^\top t} \right) (XX^\top)^{-1}X\boldsymbol{y}, \tag{103}$$

when $d < n$, or otherwise

$$\boldsymbol{w}_+^S(t) = -\boldsymbol{w}_-^S(t) = \frac{1}{2\sqrt{h}} X \left( I - e^{-\frac{\phi'(0)^2}{n_0}X^\top X t} \right) (X^\top X)^{-1}\boldsymbol{y}. \tag{104}$$

when $d > n$. For simplicity we elaborate the proof only for $d < n$. We provide a condition under which the difference between the trajectories can be controlled, and we first that this condition holds for the linearized flow GF-single for all $t$ in Lemma 17.

**Condition A:** $\|\boldsymbol{w}(t)\|_2 = O\left(\frac{1}{\sqrt{d}}\right)$ ; $\|X^\top\boldsymbol{w}(t)\|_\infty = O\left(\frac{\text{poly}\log d}{\sqrt{d}}\right)$.

**Step 2. Bounding the Difference in Gradient Flow Trajectory.** Due to low rank property of GF-single and GF-double, in this subsection we slightly abuse the notation and define

$$f(\boldsymbol{x};\boldsymbol{w}_{\pm}) = f(\boldsymbol{x};\boldsymbol{w}_+\boldsymbol{1}^\top, \boldsymbol{w}_-\boldsymbol{1}^\top) = \sqrt{h}\phi(\boldsymbol{w}_+^\top\boldsymbol{x}) - \sqrt{h}\phi(\boldsymbol{w}_-^\top\boldsymbol{x}). \tag{105}$$

We now show that the difference between the two trajectories defined above is asymptotically vanishing for $\boldsymbol{w}_+$ ($\boldsymbol{w}_-$ follows the same argument). Compare the two trajectories up to time $T$

$$\left\|\boldsymbol{w}_+^D(T) - \boldsymbol{w}_+^S(T)\right\|_2 = \left\|\int_0^T \boldsymbol{g}_+^D(\boldsymbol{w}_+^D(t)) - \boldsymbol{g}_+^S(\boldsymbol{w}_+^S(t))dt\right\|_2$$

$$\leq \left\|\int_0^T \boldsymbol{g}_+^S(\boldsymbol{w}_+^S(t)) - \boldsymbol{g}_+^S(\boldsymbol{w}_+^D(t))\,dt\right\|_2 + \left\|\int_0^T \boldsymbol{g}_+^D(\boldsymbol{w}_+^D(t)) - \boldsymbol{g}_+^S(\boldsymbol{w}_+^D(t))\,dt\right\|_2$$

$$= \left\|\int_0^T \frac{1}{2n_0} \sum_{i=1}^{2n_0} \left[ \left( -\phi'(0)(\boldsymbol{w}_+^D(t) - \boldsymbol{w}_+^S(t))^\top\boldsymbol{x}_i + \phi'(0)(\boldsymbol{w}_-^D(t) - \boldsymbol{w}_-^S(t))^\top\boldsymbol{x}_i \right) \phi'(0)\boldsymbol{x}_i \right]\,dt\right\|_2 + E_+$$

$$= O(1) \int_0^T \left( \left\|\boldsymbol{w}_+^D(t) - \boldsymbol{w}_+^S(t)\right\|_2 + \left\|\boldsymbol{w}_-^D(t) - \boldsymbol{w}_-^S(t)\right\|_2 \right)\,dt + E_+, \tag{106}$$

where we have defined the error term as

$$E_+ = \left\|\int_0^T \boldsymbol{g}_+^D(\boldsymbol{w}_+^D(s)) - \boldsymbol{g}_+^S(\boldsymbol{w}_+^D(s))\,dt\right\|_2. \tag{107}$$

To bound the error term, we note that at $t = 0$ Condition A holds for GF-double. Assume that for some $0 \leq t \leq T$, Condition A also holds for GF-double, we have

$$
\begin{aligned}
E_+ &= \left\| \int_0^T \boldsymbol{g}_+^D(\boldsymbol{w}_+^D(t)) - \boldsymbol{g}_+^S(\boldsymbol{w}_+^D(t)) \, \mathrm{d}t \right\|_2 \\
&\leq \left\| \int_0^T \frac{1}{n} \sum_{i=1}^n \Big[ \frac{1}{\sqrt{h}} \Big( y_i - \sqrt{h}\phi(\boldsymbol{w}_+^{D\top}\boldsymbol{x}_i) + \sqrt{h}\phi(\boldsymbol{w}_-^{D\top}\boldsymbol{x}_i) \Big) \Big( \phi'(\boldsymbol{w}_+^{D\top}\boldsymbol{x}_i) - \phi'(0) \Big) \boldsymbol{x}_i \Big] \, \mathrm{d}t \right\|_2 \\
&\quad + \left\| \int_0^T \frac{1}{n} \sum_{i=1}^n \Big[ \Big( -\phi(\boldsymbol{w}_+^{D\top}\boldsymbol{x}_i) + \phi'(0)\boldsymbol{w}_+^{D\top}\boldsymbol{x}_i + \phi(0) + \phi(\boldsymbol{w}_-^{D\top}\boldsymbol{x}_i) - \phi'(0)\boldsymbol{w}_-^{D\top} - \phi(0) \Big) \phi'(0)\boldsymbol{x}_i \Big] \, \mathrm{d}t \right\|_2 \\
&\overset{(i)}{\leq} \left\| \int_0^T \frac{1}{n\sqrt{h}} X(\boldsymbol{y} - \boldsymbol{y}^D(t)) \circ (\phi'(X^\top \boldsymbol{w}_+^D) - \phi'(0)\boldsymbol{1}) \, \mathrm{d}t \right\|_2 \\
&\quad + \left\| \int_0^T \frac{1}{n} \sum_{i=1}^n \Big[ \Big( O(\boldsymbol{w}_+^{D\top}\boldsymbol{x}_i)^2 - O(\boldsymbol{w}_-^{D\top}\boldsymbol{x}_i)^2 \Big) \boldsymbol{x}_i \Big] \, \mathrm{d}t \right\|_2 \\
&\overset{(i)}{\leq} \frac{1}{n\sqrt{h}} \int_0^T \|X\|_2 \left\|\boldsymbol{y} - \boldsymbol{y}^D(t)\right\|_\infty \left\| O(\phi'(X^\top \boldsymbol{w}_+^D)) \right\|_2 \, \mathrm{d}t + O(n^{-1}) \int_0^T \|X\|_2 \sqrt{n} \left\| X^\top \boldsymbol{w}_\pm^D \right\|_\infty^2 \, \mathrm{d}t \\
&\overset{(ii)}{\leq} \frac{1}{n\sqrt{h}} \int_0^T O(\sqrt{d}) O(\log^c d) O(1) \, \mathrm{d}t + \frac{1}{\sqrt{n}} \int_0^T O(\sqrt{d}) O\left( \frac{\log^c d}{d} \right) \, \mathrm{d}t = O\left( \frac{\log^c d}{d} \right) T.
\end{aligned}
\tag{108}
$$

where (i) follows from the error of Taylor expansion on $\phi$ and (ii) from Condition A. Therefore, by Equation (106) and Gronwall's inequality,

$$
\left\| \boldsymbol{w}_+^D(T) - \boldsymbol{w}_+^S(T) \right\|_2 \leq C_1 \cdot \frac{\log^c h}{h} e^{C_2 T} = O\left( \frac{\operatorname{poly}\log h}{h} \right) \to 0.
\tag{109}
$$

for $T \in O(\log\log h)$. This shows that up to time T, the difference between the trajectories of GF-single and GF-double vanishes under the the assumption that Condition A holds for $\boldsymbol{w}^D$ up to T. Importantly, note that $\boldsymbol{w}^S(0) = \boldsymbol{w}^D(0) = 0$, and that Condition A holds for $\boldsymbol{w}^S$ for all $T > 0$. Therefore, by a standard contradiction argument (e.g. (Du et al., 2018, Lemma 3.4)), one can show that this closeness between the two trajectories implies that Condition A also holds for $\boldsymbol{w}^D$ up to time T. With Equation (108) we bound the difference in population risk between the two flows.

**Step 3. Bounding the Difference in Risk.** Now we consider the difference of the population risk $R^S$ and $R^D$ of two models with parameters $\boldsymbol{w}_\pm^S$ and $\boldsymbol{w}_\pm^D$.

$$
\begin{aligned}
|2(R^S - R^D)| &= \left| \mathbb{E}_{\boldsymbol{x}} \left( \boldsymbol{x}^\top \boldsymbol{\beta} - f(\boldsymbol{x}; \boldsymbol{w}_\pm^S) \right)^2 - \mathbb{E}_{\boldsymbol{x}} \left( \boldsymbol{x}^\top \boldsymbol{\beta} - f(\boldsymbol{x}; \boldsymbol{w}_\pm^D) \right)^2 \right| \\
&\overset{(i)}{\leq} \sqrt{ \mathbb{E}_{\boldsymbol{x}}[f(\boldsymbol{x}; \boldsymbol{w}_\pm^S) - f(\boldsymbol{x}; \boldsymbol{w}_\pm^D)]^2 \mathbb{E}_{\boldsymbol{x}}[\boldsymbol{\beta}^\top \boldsymbol{x} - f(\boldsymbol{x}; \boldsymbol{w}_\pm^S) + \boldsymbol{\beta}^\top \boldsymbol{x} - f(\boldsymbol{x}; \boldsymbol{w}_\pm^D)]^2 } \\
&\leq \sqrt{ h\mathbb{E}_{\boldsymbol{x}}[|\phi(\boldsymbol{w}_+^{S\top}\boldsymbol{x}) - \phi(\boldsymbol{w}_+^{D\top}\boldsymbol{x})| + |\phi(\boldsymbol{w}_-^{S\top}\boldsymbol{x}) - \phi(\boldsymbol{w}_-^{D\top}\boldsymbol{x})|]^2 } \\
&\quad \cdot \sqrt{ \mathbb{E}_{\boldsymbol{x}}[|\boldsymbol{\beta}^\top \boldsymbol{x} - f(\boldsymbol{x}; \boldsymbol{w}_\pm^S)| + |\boldsymbol{\beta}^\top \boldsymbol{x} - f(\boldsymbol{x}; \boldsymbol{w}_\pm^D)|]^2 } \\
&\overset{(ii)}{\leq} 2\sqrt{ h\mathbb{E}_{\boldsymbol{x}}[|\phi(\boldsymbol{w}_+^{S\top}\boldsymbol{x}) - \phi(\boldsymbol{w}_+^{D\top}\boldsymbol{x})|^2 + |\phi(\boldsymbol{w}_-^{S\top}\boldsymbol{x}) - \phi(\boldsymbol{w}_-^{D\top}\boldsymbol{x})|^2] } \\
&\quad \cdot \sqrt{ \mathbb{E}_{\boldsymbol{x}}[|\boldsymbol{\beta}^\top \boldsymbol{x} - f(\boldsymbol{x}; \boldsymbol{w}_\pm^S)|^2 + |\boldsymbol{\beta}^\top \boldsymbol{x} - f(\boldsymbol{x}; \boldsymbol{w}_\pm^D)|^2] } \\
&\overset{(iii)}{\leq} 2\sqrt{ h\operatorname{Lip}(\phi)\mathbb{E}_{\boldsymbol{x}}[|\boldsymbol{w}_+^{S\top}\boldsymbol{x} - \boldsymbol{w}_+^{D\top}\boldsymbol{x}|^2 + |\boldsymbol{w}_-^{S\top}\boldsymbol{x} - \boldsymbol{w}_-^{D\top}\boldsymbol{x}|^2] } \cdot \sqrt{R^S + R^D} \\
&\leq O(\sqrt{h})\sqrt{ \left\|\boldsymbol{w}_+^S - \boldsymbol{w}_+^D\right\|_2^2 + \left\|\boldsymbol{w}_-^S - \boldsymbol{w}_-^D\right\|_2^2 } \leq O\left( \frac{\operatorname{poly}\log h}{\sqrt{h}} \right),
\end{aligned}
\tag{110}
$$

where (i) is due to Cauchy-Schwarz inequality on norm, (ii) from Jenson's inequality on squares and Young's inequality, and (iii) from the Lipschitz assumption on the activation, and the observation

that both $R^S$ and $R^D$ are finite for $\gamma_1 \neq 1$ due to the justified Condition A above. Therefore the difference between $R^S$ and $R^D$ vanishes for $T = O(\log \log h)$.

**Step 4. Bounding the Difference from Stationarity** We compute the difference in risk between the model at some finite time $t$ and the stationary point i.e. $t = \infty$,

$$
|R^S(t) - R^S(\infty)| \leq C\sqrt{h} \cdot \left\| \boldsymbol{w}_+^S(t) - \boldsymbol{w}_+^S(\infty) \right\| = C\sqrt{h} \left\| \frac{1}{2\sqrt{h}} e^{-\frac{\phi'(0)^2}{n_0} XX^\top t} (XX^\top)^{-1} X\boldsymbol{y} \right\|_2
$$

$$
= C \left\| e^{-\frac{\phi'(0)^2}{n_0} XX^\top t} (XX^\top)^{-1} XX^\top \boldsymbol{\beta} \right\|_2 \leq C \exp\left( -\phi'(0)^2 \left\| \frac{1}{n} XX^\top \right\|_2 t \right) \|\boldsymbol{\beta}\|_2 = C_3 e^{-C_4 t}.
$$
(111)

for constants $C_3, C_4 > 0$. Combining (110) and (111) yields for $T = \log \log h$,

$$
|R^D(T) - R^S(\infty)| \leq |R^S(T) - R^D(T)| + |R^S(T) - R^S(\infty)|
$$

$$
= O\left( \frac{\text{poly} \log h}{\sqrt{h}} \right) + O\left( \frac{1}{\text{poly} \log h} \right) \to 0.
$$
(112)

The result in (112) for case $d > n$ follows a similar proof and is omitted.

### C.8.3 FROM GF-ORIGINAL TO GF-DOUBLE

In this section we compare GF-original with GF-double. Note that the two flows differ only at initialization: vanishing initialization $\boldsymbol{w}_i^O(0) \sim \mathcal{N}(\boldsymbol{0}, I/dh^{1+\epsilon})$ v.s. zero initialization $\boldsymbol{w}_i^D(0) = \boldsymbol{0}$. Note that at initialization $\|\boldsymbol{y} - f(X)\|_2 = O(\sqrt{n})$, and gradient flow decreases the empirical risk; therefore the condition for Lemma 18 is satisfied, and by the Lipschitz condition on the empirical gradient we have

$$
\left\| W^O(T) - W^D(T) \right\|_F^2
$$

$$
= \left\| W^O(0) - W^D(0) \right\|_F^2 + \int_0^T \frac{\partial \left\| W^O(t) - W^D(t) \right\|_F^2}{\partial t}\, dt
$$

$$
= \left\| W^O(0) \right\|_F^2 + \int_0^T \left\| W^O(t) - W^D(t)) \right\|_F \left\| \frac{\partial L(X; W^O(t))}{\partial W} - \frac{\partial L(X; W^D(t))}{\partial W} \right\|_F dt
$$

$$
\leq \left\| W^O(0) \right\|_F^2 + \int_0^T \left\| W^O(t) - W^D(t) \right\|_F^2 + \left\| \frac{\partial L(X; W^O(t))}{\partial W} - \frac{\partial L(X; W^D(t))}{\partial W} \right\|_F^2 dt
$$

$$
\leq \left\| W^O(0) \right\|_F^2 + (1 + L_f) \int_0^T \left\| W^O(t) - W^D(t) \right\|_F^2 dt,
$$
(113)

And hence by Gronwall's lemma one obtains:

$$
\left\| W^O(T) - W^D(T) \right\|_F \leq \left\| W^O(0) \right\|_F e^{(1+L_f)T/2} = O(d^{-(1+\epsilon)/2}) e^{CT}.
$$
(114)

Therefore the difference in the function output can be bounded as

$$
\left\| f(\boldsymbol{x}, W^D(T)) - f(\boldsymbol{x}, W^O(T)) \right\|_2 = \left\| \phi(\boldsymbol{x}^\top W^D(T))\boldsymbol{a} - \phi(\boldsymbol{x}^\top W^O(T))\boldsymbol{a} \right\|_2
$$

$$
\leq \left\| \phi(\boldsymbol{x}^\top W^D(T)) - \phi(\boldsymbol{x}^\top W^O(T)) \right\|_2 \|\boldsymbol{a}\|_2 \leq L_\phi \|\boldsymbol{x}\|_2 \left\| W^D(T) - W^O(T) \right\|_F \|\boldsymbol{a}\|_2
$$

$$
= O(\sqrt{d}) \cdot \left\| W^D(T) - W^O(T) \right\|_F = O(d^{-\epsilon/2}) e^{CT}.
$$
(115)

Taking $T = \log \log h$, together with the same argument in Step 1 yields

$$
|R^O(T) - R^D(T)| = O\left( \frac{\text{poly} \log h}{d^{\epsilon/2}} \right) \to 0.
$$
(116)

### C.8.4 PUTTING THINGS TOGETHER

By (112) and (116), we know that for $T = \log \log h$,

$$
|R^O(T) - R^S(\infty)| \leq |R^O(T) - R^D(T)| + |R^D(T) - R^S(\infty)| \to 0.
$$
(117)

Finally the proof is completed by observing that $R^S(\infty)$ is the risk of the minimum-norm solution on the input discussed in Section 3. In addition, note that at $T = \log\log h$, from (109)(114) one obtains that $\left\|W^O(t) - W^S(t)\right\|_F \to 0$, and therefore by Lemma 18 we have $\left\|\partial L(X; W^O(t))/\partial W\right\|_F \to 0$, i.e. the flow on the original objective reaches a $o(1)$ first-order stationary point. $\qquad\square$

## C.9 Proof of Theorem 8

Denote $\boldsymbol{\omega} = \mathrm{vec}(W) = \mathrm{vec}([W_+, W_-])$, and $\boldsymbol{\omega}_0 = \mathrm{vec}(W^{\mathrm{init}})$. Define

$$K(t) = \frac{\partial \boldsymbol{f}(X; \boldsymbol{\omega}(t))}{\partial \boldsymbol{\omega}(t)}^\top \frac{\partial \boldsymbol{f}(X; \boldsymbol{\omega}(t))}{\partial \boldsymbol{\omega}(t)}, \tag{118}$$

which is the kernel matrix of the *neural tangent kernel* Jacot et al. (2018); Du et al. (2018). In the following sections we show that under the non-vanishing initialization, the trained two-layer network is well-approximated by the regression model on the NTK, for which we derive the population risk.

### C.9.1 The Kernel Linearization

Write $\boldsymbol{y}_{NN}(t) = f_{NN}(X, t) \in \mathbb{R}^n$ and its evolution:

$$\mathrm{d}\boldsymbol{y}_{NN}(t) = \frac{1}{n} K(t)(\boldsymbol{y} - \boldsymbol{y}_{NN}(t)) \, \mathrm{d}t, \tag{119}$$

and the corresponding linearized flow:

$$\mathrm{d}\boldsymbol{y}_{NTK}(t) = \frac{1}{n} K(0)(\boldsymbol{y} - \boldsymbol{y}_{NTK}(t)) \, \mathrm{d}t, \tag{120}$$

Previous works (e.g. Du et al. (2018); Oymak and Soltanolkotabi (2019)) have proved (non-asymptotically) that the two trajectories (119) and (120) are close if the model is overparameterized, i.e. $h = \mathrm{poly}(n)$, under no assumptions on the teacher model. In our asymptotic setup (together with assumptions (A1-3)), we argue that similar conclusion holds without significant overparameterization.

We first show the global convergence of the training of two-layer neural network $f_{NN}$. We employ an argument similar to (Du et al., 2018, Theo. 3.2) by first identifying the condition under which training converges at linear rate:

**Condition B:** $\lambda_{\min}(K(t)) = O(d)$.

From Corollary 15 we know that at initialization the lowest eigenvalue of the NTK matrix satisfies $\lambda_{\min}(K(0)) = O(d)$, and thus the kernel regression on the NTK enjoys linear convergence. By Lemma 19, we know that for $\|W(t) - W(0)\|_2 = O(d^{1-\epsilon})$, the order of $\lambda_{\min}(K(t)) = O(d)$ remains unchanged. Therefore, if we assume that up to time $T$ the weights satisfy $\|\boldsymbol{w}_i(t) - \boldsymbol{w}_i(0)\|_2 = O(d^{-1/2})$, then Condition B is satisfied for $\boldsymbol{y}_{NN}$ and from (Chizat and Bach, 2018b, Lemma B1) we have the following linear convergence

$$\|\boldsymbol{y}_{NN}(t) - \boldsymbol{y}\|_2 \le C_1 \|\boldsymbol{y}_{NN}(0) - \boldsymbol{y}\|_2 \, e^{-C_2 t}. \tag{121}$$

Since $\|\boldsymbol{y}_{NN}(0) - \boldsymbol{y}\|_2 = O(\sqrt{n})$ at initialization, setting $T = O(\log d)$ ensures that the training loss $\frac{1}{n}\|\boldsymbol{y}_{NN}(T) - \boldsymbol{y}\|_2^2 \to 0$ as $n \to \infty$. Consequently it is easy to check that $\left\|\frac{\partial L(X; W(T))}{\partial W(T)}\right\|_2 \to 0$ and thus at time T the gradient flow reaches an o(1) first order stationary point.

We now verify that each weight vector $\boldsymbol{w}_i$ travels at most $O(d^{-1/2})$ from initialization. The norm of gradient for $\boldsymbol{w}_i$ can be bounded as:

$$\left\|\frac{\partial L(X; \boldsymbol{w}_i(t))}{\partial \boldsymbol{w}_i(t)}\right\|_2 = \left\|\frac{1}{n} X \left[\frac{1}{\sqrt{h}}(\boldsymbol{y} - \boldsymbol{y}_{NN}(t)) \circ \phi'(X^\top \boldsymbol{w}_i(t))\right]\right\|_2$$

$$\overset{(i)}{\le} O(1)\frac{1}{d^{1.5}} \|X\|_2 \|\boldsymbol{y} - \boldsymbol{y}_{NN}(t)\|_2 \le O(1) d^{-1} \|\boldsymbol{y} - \boldsymbol{y}_{NN}(0)\|_2 \, e^{-t}, \tag{122}$$

where (i) follows from the boundedness of $\phi'$. Integrating the gradient yields $\|\boldsymbol{w}_i(t) - \boldsymbol{w}_i(0)\|_2 = O(d^{-1/2})$, i.e. $\|W(t) - W(0)\|_2 = O(1)$. Thus the distance traveled by $W$ indeed satisfies the norm

assumption above, and following the same argument as (Du et al., 2018, Lemma 3.4) we conclude that Condition B holds true for the gradient flow of $f_{NN}$ for $t > 0$.

Next we show that for any input $\hat{\boldsymbol{x}}$ with $\|\hat{\boldsymbol{x}}\|_2 = O(\sqrt{d})$, the prediction of the neural network is uniformly close to the prediction of the prediction of the kernel model, a result similar to (Arora et al., 2019a, Lemma F.1). Following the notation of Arora et al. (2019a), we write the time derivative of the prediction as

$$\frac{\mathrm{d}}{\mathrm{d}t} f_{NN}(\hat{\boldsymbol{x}}, t) = \frac{1}{n} \boldsymbol{u}_{NN}(\hat{\boldsymbol{x}}, t)^\top (\boldsymbol{y} - \boldsymbol{y}_{NN}(t)); \; \frac{\mathrm{d}}{\mathrm{d}t} f_{NTK}(\hat{\boldsymbol{x}}, t) = \frac{1}{n} \boldsymbol{u}_{NTK}(\hat{\boldsymbol{x}}, t)^\top (\boldsymbol{y} - \boldsymbol{y}_{NTK}(t)),$$

where $\boldsymbol{u}_{NN}(\boldsymbol{x}, t) = \frac{\partial \boldsymbol{f}(X; \boldsymbol{\omega}(t))}{\partial \boldsymbol{\omega}(t)}^\top \frac{\partial f(\boldsymbol{x}; \boldsymbol{\omega}(t))}{\partial \boldsymbol{\omega}(t)} \in \mathbb{R}^n$ and $\boldsymbol{u}_{NTK}(\boldsymbol{x}, t)$ similarly defined on the initialized weights $\omega(0)$. We bound the difference between the predictions on $\hat{\boldsymbol{x}}$ up to terminal time T as

$$|f_{NN}(\hat{\boldsymbol{x}}, t) - f_{NTK}(\hat{\boldsymbol{x}}, t)| = |\int_0^T \left[ \frac{\mathrm{d}}{\mathrm{d}t} f_{NN}(\hat{\boldsymbol{x}}, t) - \frac{\mathrm{d}}{\mathrm{d}t} f_{NTK}(\hat{\boldsymbol{x}}, t) \right] \mathrm{d}t|$$

$$= \frac{1}{n} |\int_0^T \left[ \boldsymbol{u}_{NN}(\hat{\boldsymbol{x}}, t)^\top (\boldsymbol{y} - \boldsymbol{y}_{NN}(t)) - \boldsymbol{u}_{NTK}(\hat{\boldsymbol{x}}, t)^\top (\boldsymbol{y} - \boldsymbol{y}_{NTK}(t)) \right] \mathrm{d}t|$$

$$\leq \frac{1}{n} |\int_0^T \boldsymbol{u}_{NTK}(\hat{\boldsymbol{x}}, t)^\top (\boldsymbol{y}_{NN}(t) - \boldsymbol{y}_{NTK}(t)) \, \mathrm{d}t|$$

$$+ \frac{1}{n} |\int_0^T (\boldsymbol{u}_{NN}(\hat{\boldsymbol{x}}, t) - \boldsymbol{u}_{NTK}(\hat{\boldsymbol{x}}, t))^\top (\boldsymbol{y} - \boldsymbol{y}_{NN}(t)) \, \mathrm{d}t|$$

$$\leq \frac{1}{n} \|\boldsymbol{u}_{NTK}(\hat{\boldsymbol{x}}, t)\|_2 \int_0^T \|\boldsymbol{y}_{NN}(t) - \boldsymbol{y}_{NTK}(t)\|_2 \, \mathrm{d}t$$

$$+ \frac{1}{n} \max_{t < T} \|\boldsymbol{u}_{NN}(\hat{\boldsymbol{x}}, t) - \boldsymbol{u}_{NTK}(\hat{\boldsymbol{x}}, t)\|_2 \int_0^T \|\boldsymbol{y} - \boldsymbol{y}_{NN}(t)\|_2 \, \mathrm{d}t. \tag{123}$$

For the first term have

$$\|\boldsymbol{y}_{NN}(T) - \boldsymbol{y}_{NTK}(T)\|_2 \leq \frac{1}{n} \int_0^T \|K(t)(\boldsymbol{y} - \boldsymbol{y}_{NN}(t)) - K(0)(\boldsymbol{y} - \boldsymbol{y}_{NTK}(t))\|_2 \, \mathrm{d}t$$

$$\leq \frac{1}{n} \max_{0 < t < T} \|K(t) - K(0)\|_2 \int_0^T \|\boldsymbol{y} - \boldsymbol{y}_{NN}(t)\|_2 \, \mathrm{d}t + \frac{1}{n} \|K(0)\|_2 \int_0^T \|\boldsymbol{y}_{NN}(t) - \boldsymbol{y}_{NTK}(t)\|_2 \, \mathrm{d}t$$

$$\overset{(i)}{\leq} \frac{1}{n} O(d^{1/2 - \epsilon'}) O(\sqrt{d}) + O(1) \int_0^T \|\boldsymbol{y}_{NN}(t) - \boldsymbol{y}_{NTK}(t)\|_2 \, \mathrm{d}t \overset{(ii)}{\leq} O(d^{-\epsilon'}), \tag{124}$$

where (i) is due to Corollary 15, Lemma 19 for some $\epsilon' > 0$ and the linear convergence of $\boldsymbol{y}_{NN}$, and (ii) is due to Gronwall's inequality. Note that the log factor in $T$ is omitted. Similarly, for the second term we have

$$\frac{1}{n} \max_{t < T} \|\boldsymbol{u}_{NN}(\hat{\boldsymbol{x}}, t) - \boldsymbol{u}_{NTK}(\hat{\boldsymbol{x}}, t)\|_2 \int_0^T \|\boldsymbol{y} - \boldsymbol{y}_{NN}(t)\|_2 \, \mathrm{d}t \overset{(i)}{\leq} \frac{1}{n} O(d^{1/2 - \epsilon'}) O(\sqrt{d}) = O(d^{-\epsilon'}), \tag{125}$$

where we used Lemma 19 and the linear convergence of $\boldsymbol{y}_{NN}$ in (i). Combining the two cases yields

$$|f_{NN}(\hat{\boldsymbol{x}}, t) - f_{NTK}(\hat{\boldsymbol{x}}, t)| \leq \frac{1}{n} \|\boldsymbol{u}_{NTK}(\hat{\boldsymbol{x}}, t)\|_2 O(d^{-\epsilon'}) + O(d^{-\epsilon'}) \overset{(i)}{=} O(d^{-\epsilon'}), \tag{126}$$

where we utilized Corollary 14 in (i). Thus we know that the difference between the population risk of $f_{NN}$ and $f_{NTK}$ is also asymptotically vanishing (note that the derivation above is independent of the target function as long as $\|\boldsymbol{y}\|_2 = O(\sqrt{n})$). Therefore, in the following subsection we compute the risk of the linearized (kernel) model $f_{NTK}$.

### C.9.2 COMPUTING THE KERNEL RISK

Given input $X \in \mathbb{R}^{d \times n}$ and label $\boldsymbol{y} = \boldsymbol{\beta}^\top X + \boldsymbol{\varepsilon}$, gradient flow on the tangent kernel solves the following equation of the parameters $\boldsymbol{\omega}$:

$$\boldsymbol{y} = \boldsymbol{f}(X; \boldsymbol{\omega}) = \frac{\partial \boldsymbol{f}(X; \boldsymbol{\omega}_0)}{\partial \boldsymbol{\omega}}^\top (\boldsymbol{\omega} - \boldsymbol{\omega}_0), \tag{127}$$

where $\partial \boldsymbol{f}(X; \boldsymbol{\omega}_0)/\partial \boldsymbol{\omega}$ is a $dh \times n$ matrix with column $\partial f(\boldsymbol{x}_i; \boldsymbol{\omega}_0)/\partial \boldsymbol{\omega}$. Note that for $n \to \infty$ and $\gamma_1, \gamma_2 \in (0, \infty)$, $dh > n$ trivially holds, and by Corollary 15 we know that solution is given by

$$\boldsymbol{\omega}_1 = \boldsymbol{\omega}_0 + \frac{\partial \boldsymbol{f}(X; \boldsymbol{\omega}_0)}{\partial \boldsymbol{\omega}} \left( \frac{\partial \boldsymbol{f}(X; \boldsymbol{\omega}_0)}{\partial \boldsymbol{\omega}}^\top \frac{\partial \boldsymbol{f}(X; \boldsymbol{\omega}_0)}{\partial \boldsymbol{\omega}} \right)^{-1} (X^\top \boldsymbol{\beta} + \boldsymbol{\varepsilon}). \tag{128}$$

And the population risk can be written as (note that there is a factor of 2 due to the "doubling trick" at initialization to ensure $f^{\text{init}}(\cdot) = 0$):

$$2R = \mathbb{E}_{\boldsymbol{x}, \boldsymbol{\varepsilon}} \left[ \left( \boldsymbol{x}^\top \boldsymbol{\beta} - f(x; \boldsymbol{\omega}_1) \right)^2 \right]$$

$$= \mathbb{E}_{\boldsymbol{x}, \boldsymbol{\varepsilon}} \left[ \left( \boldsymbol{x}^\top \boldsymbol{\beta} - \frac{\partial f(\boldsymbol{x}; \boldsymbol{\omega}_0)}{\partial \boldsymbol{\omega}}^\top (\boldsymbol{\omega}_1 - \boldsymbol{\omega}_0) \right)^2 \right]$$

$$= \mathbb{E}_{\boldsymbol{x}, \boldsymbol{\varepsilon}} \left[ \left( \boldsymbol{x}^\top \boldsymbol{\beta} - \frac{\partial f(\boldsymbol{x}; \boldsymbol{\omega}_0)}{\partial \boldsymbol{\omega}}^\top \frac{\partial \boldsymbol{f}(X; \boldsymbol{\omega}_0)}{\partial \boldsymbol{\omega}} \left( \frac{\partial \boldsymbol{f}(X; \boldsymbol{\omega}_0)}{\partial \boldsymbol{\omega}}^\top \frac{\partial \boldsymbol{f}(X; \boldsymbol{\omega}_0)}{\partial \boldsymbol{\omega}} \right)^{-1} (X^\top \boldsymbol{\beta} + \boldsymbol{\varepsilon}) \right)^2 \right]$$

$$= \underbrace{\mathbb{E}_{\boldsymbol{x}} \left[ \left( \boldsymbol{x}^\top \boldsymbol{\beta} - \hat{\boldsymbol{u}}^\top \hat{K}_X^{-1} X^\top \boldsymbol{\beta} \right)^2 \right]}_{2B} + \underbrace{\mathbb{E}_{\boldsymbol{x}} \left[ \hat{\boldsymbol{u}} \hat{K}_X^{-1} \hat{K}_X^{-1} \hat{\boldsymbol{u}}^\top \right] \sigma^2}_{2V}, \tag{129}$$

where a bias-variance decomposition is made here, and for simplicity we define

$$\hat{\boldsymbol{u}} = \frac{\partial \boldsymbol{f}(X; \boldsymbol{\omega}_0)}{\partial \boldsymbol{\omega}}^\top \frac{\partial f(\boldsymbol{x}; \boldsymbol{\omega}_0)}{\partial \boldsymbol{\omega}}, \quad \hat{K}_X = \frac{\partial \boldsymbol{f}(X; \boldsymbol{\omega}_0)}{\partial \boldsymbol{\omega}}^\top \frac{\partial \boldsymbol{f}(X; \boldsymbol{\omega}_0)}{\partial \boldsymbol{\omega}}. \tag{130}$$

### C.9.3 APPROXIMATING THE KERNEL MATRIX

In this section we drop the negligible $\epsilon$ in the initialization. Following Cheng and Singer (2013) we utilize the orthonormal decomposition of $\phi'(x)$ in $L^2(\mathbb{R}, \mu_G)$. Denote $b_0 = \mathbb{E}[\phi'(G)]$, and $b_1^2 = \mathbb{E}[\phi'(G)^2] - b_0^2$. We have the orthogonal decomposition of $\phi'$

$$\phi'(x) = b_0 + \phi'_\perp(x), \tag{131}$$

where $\mathbb{E}[\phi'_\perp(G)] = 0$. We develop the following lemmas to approximate the kernel matrix.

**Lemma 13** (Approximation of $(\hat{K}_X)_{ij}$). *There exist constants $c, c' > 0$ such that for $i \neq j$ with probability $1 - e^{-cn\varepsilon^2}$ we have*

$$\left| \frac{1}{d} \frac{\partial f(\boldsymbol{x}_i; \boldsymbol{\omega}_0)}{\partial \boldsymbol{\omega}}^\top \frac{\partial f(\boldsymbol{x}_j; \boldsymbol{\omega}_0)}{\partial \boldsymbol{\omega}} - \frac{1}{d} b_0^2 \boldsymbol{x}_i^\top \boldsymbol{x}_j \right| < \varepsilon^2, \tag{132}$$

*and with probability $1 - e^{-c'n\varepsilon^2}$,*

$$\left| \frac{1}{d} \frac{\partial f(\boldsymbol{x}_i; \boldsymbol{\omega}_0)}{\partial \boldsymbol{\omega}}^\top \frac{\partial f(\boldsymbol{x}_i; \boldsymbol{\omega}_0)}{\partial \boldsymbol{\omega}} - (b_0^2 + b_1^2) \right| < \varepsilon. \tag{133}$$

**Proof.** When $i \neq j$ (i.e. Equation (132)), we have

$$\frac{1}{d}[\hat{K}_X]_{ij} = \frac{1}{d} \frac{\partial f(\boldsymbol{x}_i; \boldsymbol{\omega}_0)}{\partial \boldsymbol{\omega}}^\top \frac{\partial f(\boldsymbol{x}_j; \boldsymbol{\omega}_0)}{\partial \boldsymbol{\omega}}$$

$$= \frac{1}{d} \sum_{k=1}^h \frac{\partial f(\boldsymbol{x}_i; \boldsymbol{\omega}_0)}{\partial \boldsymbol{w}_k}^\top \frac{\partial f(\boldsymbol{x}_j; \boldsymbol{\omega}_0)}{\partial \boldsymbol{w}_k} = \frac{1}{dh} \sum_{k=1}^h \boldsymbol{x}_i^\top \boldsymbol{x}_j \phi'(\boldsymbol{w}_k^\top \boldsymbol{x}_i) \phi'(\boldsymbol{w}_k^\top \boldsymbol{x}_j)$$

$$\to \frac{1}{d} \boldsymbol{x}_i^\top \boldsymbol{x}_j \mathbb{E}_{\boldsymbol{w}} \left[ \phi'(\boldsymbol{w}^\top \boldsymbol{x}_i) \phi'(\boldsymbol{w}^\top \boldsymbol{x}_j) \right] = \frac{1}{d} H(\boldsymbol{x}_i, \boldsymbol{x}_j). \tag{134}$$

The matrix $H(\boldsymbol{x}_i, \boldsymbol{x}_j) = \boldsymbol{x}_i^\top \boldsymbol{x}_j \mathbb{E}_{\boldsymbol{w}}\left[\phi'(\boldsymbol{w}^\top \boldsymbol{x}_i)\phi'(\boldsymbol{w}^\top \boldsymbol{x}_j)\right]$ can be seen as the expected tangent kernel of nonlinear activation function studied in Du et al. (2018); Arora et al. (2019b). Moreover, due to the assumed boundedness of $\phi'(x)$ (A3), by Hoeffding's inequality we have

$$\Pr\left|\frac{1}{d}\frac{\partial f(\boldsymbol{x}_i; \boldsymbol{\omega}_0)}{\partial \boldsymbol{\omega}}^\top \frac{\partial f(\boldsymbol{x}_j; \boldsymbol{\omega}_0)}{\partial \boldsymbol{\omega}} - \frac{1}{d}H(\boldsymbol{x}_i, \boldsymbol{x}_j)\right| < \frac{1}{d}\boldsymbol{x}_i^\top \boldsymbol{x}_j \varepsilon > 1 - e^{-c_1 h \varepsilon^2}. \tag{135}$$

In addition, by the concentration of $\boldsymbol{x}_i^\top \boldsymbol{x}_j$ and $\|\boldsymbol{x}_i\|_2^2$, i.e. $\Pr \boldsymbol{x}_i^\top \boldsymbol{x}_j/d > \varepsilon < 1 - e^{-c_2 d \varepsilon^2}$ and $\Pr |\boldsymbol{x}_i^\top \boldsymbol{x}_i/d - 1| < \varepsilon > 1 - e^{-c_3 d \varepsilon^2}$, the orthonormal decomposition $\phi'(x) = b_0 x + \phi'_\perp(x)$ leads to the following linear approximation of the matrix $H$

$$\frac{1}{d}H(\boldsymbol{x}_i, \boldsymbol{x}_j) = b_0^2 \frac{1}{d}\boldsymbol{x}_i^\top \boldsymbol{x}_j + O\big((\boldsymbol{x}_i^\top \boldsymbol{x}_j/d)^2\big). \tag{136}$$

and by taking $\varepsilon = \boldsymbol{x}_i^\top \boldsymbol{x}_j/d$ under the joint event we can show that

$$\left|\frac{1}{d}\frac{\partial f(\boldsymbol{x}_i; \boldsymbol{\omega}_0)}{\partial \boldsymbol{\omega}}^\top \frac{\partial f(\boldsymbol{x}_j; \boldsymbol{\omega}_0)}{\partial \boldsymbol{\omega}} - \frac{1}{d}b_0^2 \boldsymbol{x}_i^\top \boldsymbol{x}_j\right| < \varepsilon^2 \tag{137}$$

with probability $1 - e^{-cd\varepsilon^2}$. The same argument follows for the case where $i = j$. $\qquad\square$

**Corollary 14** (Approximation of $\hat{\boldsymbol{u}}$). *For large enough $l > 0$, with probability $1 - de^{-c\log^l d}$*

$$\frac{1}{d}\|\hat{\boldsymbol{u}} - \tilde{\boldsymbol{u}}\|_2 = \left\|\frac{1}{d}\frac{\partial f(\boldsymbol{x}; \boldsymbol{\omega}_0)}{\partial \boldsymbol{\omega}}^\top \frac{\partial \boldsymbol{f}(X; \boldsymbol{\omega}_0)}{\partial \boldsymbol{\omega}} - \frac{1}{d}\tilde{\boldsymbol{u}}\right\|_2 < \frac{\log^l d}{d}, \tag{138}$$

*where $\tilde{\boldsymbol{u}} = b_0^2 \boldsymbol{x}^\top X$.*

**Proof.** Taking $\varepsilon = \log^l d/d$ together with Lemma 13 yields the desired result. $\qquad\square$

**Corollary 15** (Approximation of $\hat{K}_X$). *With probability $1 - de^{-c\log^l d}$*

$$\frac{1}{d}\left\|\hat{K}_X - \tilde{K}_X\right\|_F = \left\|\frac{1}{d}\frac{\partial \boldsymbol{f}(X; \boldsymbol{\omega}_0)}{\partial \boldsymbol{\omega}}^\top \frac{\partial \boldsymbol{f}(X; \boldsymbol{\omega}_0)}{\partial \boldsymbol{\omega}} - \frac{1}{d}\tilde{K}_X\right\|_F < \log^l d, \tag{139}$$

*where $\tilde{K}_X = b_0^2 X^\top X + b_1^2 dI$.*

**Proof.** Also by directly applying Lemma 13. $\qquad\square$

***Remark.*** For initialization larger or equal to $\boldsymbol{w}_i(0) \sim N(0, I_d/d)$, the above approximation does not depend on the scale of initialization.

***Remark.*** For $\phi(x) = \text{SoftPlus}(x)$, $b_0^2 = 1/4$, $b_1^2 = 0.043379$. For $\phi(x) = \text{sigmoid}(x) = (1 + e^{-x})^{-1}$, $b_0^2 = 0.042692$, $b_1^2 = 0.002144$. Note that $b_1 \geq 0$ for all smooth activations $\phi$, and the equality holds (i.e. $b_1 = 0$) if and only if $\phi$ is linear. We comment that smaller $b_1$ entails larger variance as $\gamma_1 \to 1$, and vice versa, as shown in the following section.

### C.9.4 THE BIAS TERM

With these approximation above we proceed to calculating (129)

$$2B = \mathbb{E}_{\boldsymbol{x}}\left[\left(\boldsymbol{x}^\top \boldsymbol{\beta} - \hat{\boldsymbol{u}}^\top \hat{K}_X^{-1} X^\top \boldsymbol{\beta}\right)^2\right]. \tag{140}$$

We first bound the error in substituting $\hat{\boldsymbol{u}}$ with $\tilde{\boldsymbol{u}}$:

$$\left\|\hat{\boldsymbol{u}}^\top \hat{K}_X^{-1} X^\top \boldsymbol{\beta} - \tilde{\boldsymbol{u}}^\top \hat{K}_X^{-1} X^\top \boldsymbol{\beta}\right\|_2 \leq \|\hat{\boldsymbol{u}} - \tilde{\boldsymbol{u}}\|_2 \left\|\hat{K}_X^{-1}\right\|_2 \|X\|_2 \|\boldsymbol{\beta}\|_2$$

$$\overset{(i)}{=} O\left(\log^l d \cdot d^{-1} \cdot \sqrt{d} \cdot 1\right) = O\left(\frac{\log^l d}{\sqrt{d}}\right), \tag{141}$$

where (i) is due to the fact that $\left\|\hat{K}_X^{-1}\right\|_2 = \lambda_{\min}^{-1}(\hat{K}_X) = O(1/d)$ and $\|X\|_2 = O(\sqrt{d})$. Therefore we have as $n, d, h \to \infty$

$$2B = \mathbb{E}_{\boldsymbol{x}}\left[\left(\boldsymbol{x}^\top\boldsymbol{\beta} - \hat{\boldsymbol{u}}^\top\hat{K}_X^{-1}X^\top\boldsymbol{\beta}\right)^2\right] \to \mathbb{E}_{\boldsymbol{x}}\left[\left(\boldsymbol{x}^\top\boldsymbol{\beta} - \tilde{\boldsymbol{u}}^\top\hat{K}_X^{-1}X^\top\boldsymbol{\beta}\right)^2\right]$$

$$= \mathbb{E}_{\boldsymbol{x}}\left[\left(\boldsymbol{x}^\top\boldsymbol{\beta} - b_0^2\boldsymbol{x}^\top X\hat{K}_X^{-1}X^\top\boldsymbol{\beta}\right)^2\right]. \quad (142)$$

By taking expectation over $\boldsymbol{x}$ and the rotational invariance argument similar to Hastie et al. (2019),

$$\mathbb{E}_{\boldsymbol{x}}\left[\left(\boldsymbol{x}^\top\boldsymbol{\beta} - b_0^2\boldsymbol{x}^\top X\hat{K}_X^{-1}X^\top\boldsymbol{\beta}\right)^2\right] = \mathbb{E}_{\boldsymbol{x}}\left[\boldsymbol{\beta}^\top\left(I - b_0^2 X\hat{K}_X^{-1}X^\top\right)^2\boldsymbol{\beta}\right]$$

$$= \frac{\boldsymbol{\beta}^\top\boldsymbol{\beta}}{d}\mathrm{tr}\left(\left(I - b_0^2 X\hat{K}_X^{-1}X^\top\right)\left(I - b_0^2 X\hat{K}_X^{-1}X^\top\right)\right). \quad (143)$$

In addition, we bound the error in substituting $\hat{K}_X$ by $\tilde{K}_X$ defined in (139):

$$\left|\frac{1}{d}\mathrm{tr}\left(X\hat{K}_X^{-1}X^\top - X\tilde{K}_X^{-1}X^\top\right)\right| = \left|\frac{1}{d}\mathrm{tr}\left(X^\top X\hat{K}_X^{-1}(\hat{K}_X - \tilde{K}_X)\tilde{K}_X^{-1}\right)\right|$$

$$< \frac{1}{d}\left\|X^\top X\right\|_2\left\|\hat{K}_X^{-1}\right\|_2\left\|\hat{K}_X - \tilde{K}_X\right\|_F\left\|\tilde{K}_X^{-1}\right\|_2$$

$$= O\left(d^{-1}\cdot d \cdot d^{-1}\cdot d\log^l d \cdot d^{-1}\right) = O\left(\frac{\log^l d}{d}\right), \quad (144)$$

and similarly,

$$\left|\frac{1}{d}\mathrm{tr}\left(X\hat{K}_X^{-1}X^\top X\hat{K}_X^{-1}X^\top - X\tilde{K}_X^{-1}X^\top X\tilde{K}_X^{-1}X^\top\right)\right|$$

$$= \left|\frac{1}{d}\mathrm{tr}\left(X^\top X(\hat{K}_X^{-1} - \tilde{K}_X^{-1})X^\top X(\hat{K}_X^{-1} + \tilde{K}_X^{-1})\right)\right|$$

$$< \frac{1}{d}\left\|X^\top X\right\|_2\left\|\hat{K}_X^{-1}\right\|_2\left\|\hat{K}_X - \tilde{K}_X\right\|_F\left\|\tilde{K}_X^{-1}\right\|_2\left\|X^\top X\right\|_2\left\|\hat{K}_X^{-1} + \tilde{K}_X^{-1}\right\|_2$$

$$= O\left(d^{-1}\cdot d \cdot d^{-1}\cdot d\log^l d \cdot d^{-1}\cdot d \cdot d^{-1}\right) = O\left(\frac{\log^l d}{d}\right). \quad (145)$$

Combining these two formulas in (143) yields

$$2B \to \mathbb{E}_{\boldsymbol{x}}\left[\left(\boldsymbol{x}^\top\boldsymbol{\beta} - b_0^2\boldsymbol{x}^\top X\hat{K}_X^{-1}X^\top\boldsymbol{\beta}\right)^2\right]$$

$$\to \frac{\boldsymbol{\beta}^\top\boldsymbol{\beta}}{d}\mathrm{tr}\left(\left(I - b_0^2 X\tilde{K}_X^{-1}X^\top\right)\left(I - b_0^2 X\tilde{K}_X^{-1}X^\top\right)\right)$$

$$= \frac{\boldsymbol{\beta}^\top\boldsymbol{\beta}}{d}\mathrm{tr}\left(\left(I - b_0^2 X(b_0^2 X^\top X + b_1^2 dI)^{-1}X^\top\right)^2\right). \quad (146)$$

Utilizing the Marčenko–Pastur law from Section B.2 we obtain

$$B = \boldsymbol{\beta}^\top\boldsymbol{\beta}\left(\frac{\gamma_1 - 1}{2\gamma_1} + \frac{\gamma_1(\gamma_1 + \gamma_1 m + m - 2) + 1}{2\gamma_1\sqrt{\gamma_1(\gamma_1 + m(\gamma_1(m+2)+2)-2)+1}}\right), \quad (147)$$

where $m = b_0^{-2}b_1^2$.

### C.9.5 THE VARIANCE TERM

Similarly, for the variance we utilize the approximation

$$2V = \mathbb{E}_{\boldsymbol{x}}\left[\hat{\boldsymbol{u}}\hat{K}_X^{-1}\hat{K}_X^{-1}\hat{\boldsymbol{u}}^\top\right]\sigma^2 \quad (148)$$

Specifically, we bound the approximation error

$$\left| \hat{u} \hat{K}_X^{-1} \hat{K}_X^{-1} \hat{u}^\top - \tilde{u} \hat{K}_X^{-1} \hat{K}_X^{-1} \tilde{u}^\top \right| \le \| \hat{u} - \tilde{u} \|_2 \left\| \hat{K}_X^{-1} \right\|_2 \| \hat{u} + \tilde{u} \|_2 \left\| \hat{K}_X^{-1} \right\|_2$$

$$= O \left( \log^l d \cdot \frac{1}{d} \cdot d \cdot \frac{1}{d} \right) = O \left( \frac{\log^l d}{d} \right), \qquad (149)$$

and similarly

$$\left| \mathbb{E}_{\boldsymbol{x}} \left[ \tilde{u} \hat{K}_X^{-1} \hat{K}_X^{-1} \tilde{u}^\top - \tilde{u} \tilde{K}_X^{-1} \tilde{K}_X^{-1} \tilde{u}^\top \right] \right|$$

$$= \mathrm{tr} \left( \left( \hat{K}_X^{-1} - \tilde{K}_X^{-1} \right) \left( \hat{K}_X^{-1} + \tilde{K}_X^{-1} \right) \mathbb{E}_{\boldsymbol{x}} \tilde{u} \tilde{u}^\top \right)$$

$$= \mathrm{tr} \left( \hat{K}_X^{-1} \left( \hat{K}_X - \tilde{K}_X \right) \tilde{K}_X^{-1} \left( \hat{K}_X^{-1} + \tilde{K}_X^{-1} \right) X^T X \right)$$

$$\le \left\| \hat{K}_X^{-1} \right\|_2 \left\| \hat{K}_X - \tilde{K}_X \right\|_F \left\| \tilde{K}_X^{-1} \right\|_2 \left\| \hat{K}_X^{-1} + \tilde{K}_X^{-1} \right\|_2 \| X^T X \|_2$$

$$= O(d^{-1} \cdot d \log^l d \cdot d^{-1} \cdot d^{-1} \cdot d) = O \left( \frac{\log^l d}{d} \right), \qquad (150)$$

By combining the two approximations above we know that as $n, d, p \to \infty$

$$\left| 2V - \mathbb{E}_{\boldsymbol{x}} \left[ \tilde{u} \tilde{K}_X^{-1} \tilde{K}_X^{-1} \tilde{u}^\top \right] \sigma^2 \right| = O \left( \frac{\log^l d}{d} \right) \to 0. \qquad (151)$$

Therefore the variance is given as

$$2V \to \sigma^2 \mathbb{E}_{\boldsymbol{x}} \left[ \tilde{u} \tilde{K}_X^{-1} \tilde{K}_X^{-1} \tilde{u}^\top \right]$$

$$= \sigma^2 \mathbb{E}_{\boldsymbol{x}} \left[ b_0^4 \boldsymbol{x}^T X \left( b_0^2 X^\top X + b_1^2 d I \right)^{-2} X^T \boldsymbol{x} \right]$$

$$= \sigma^2 \frac{1}{d} \mathrm{tr} \left( \frac{1}{d} X^T X \cdot \left( \frac{1}{d} X^T X + b_0^{-2} b_1^2 I \right)^{-2} \right)$$

$$= \sigma^2 \left( -\frac{1}{2} + \frac{\gamma_1 + \gamma_1 m + 1}{2 \sqrt{\gamma_1 (\gamma_1 + m(\gamma_1(m+2)+2) - 2) + 1}} \right), \qquad (152)$$

where $m$ is defined in the derivation of the bias term.

### C.9.6 Putting Things Together

Recall the population risk is the sum of the bias and variance

$$R \to r^2 \left( \frac{\gamma_1 - 1}{2\gamma_1} + \frac{\gamma_1(\gamma_1 + \gamma_1 m + m - 2) + 1}{2\gamma_1 \sqrt{\gamma_1(\gamma_1 + m(\gamma_1(m+2)+2) - 2) + 1}} \right)$$

$$+ \sigma^2 \left( -\frac{1}{4} + \frac{\gamma_1 + \gamma_1 m + 1}{4 \sqrt{\gamma_1(\gamma_1 + m(\gamma_1(m+2)+2) - 2) + 1}} \right). \qquad (153)$$

Observe that the population risk is independent of $\gamma_2$, i.e. *double descent* does not occur when the network is overparameterized via changing the width. In addition, the bias is monotonically increasing and upper-bounded by the null risk $r^2$ and lower-bounded by the bias of the least squares solution on the input features $\hat{\boldsymbol{\beta}} = X^\dagger \boldsymbol{y}$, whereas the variance remains bounded for all $\gamma_1 \in (0, \infty)$ as long as $m > 0$, i.e. $\phi$ is nonlinear.

$\square$

# D USEFUL LEMMAS

**Lemma 16.** *Given $K_W$ from* (46)*, define*

$$\tilde{K}_W = rI_h + s\mathbf{1}_h\mathbf{1}_h^\top + tQ, \tag{154}$$

*where $Q \in \mathbb{R}^{h \times h}$ with $Q_{i \neq j} = \boldsymbol{w}_i^\top \boldsymbol{w}_j$ and $Q_{i,i} = 0$, and $r = \mathbb{E}[\phi(G)^2] - \mathbb{E}[\phi(G)]^2$, $s = \mathbb{E}[\phi(G)]^2$, $t = \mathbb{E}[G\phi(G)]^2$. Then as $n, d, h \to \infty$, $\left\| K_W - \tilde{K}_W \right\|_F \leq \log^c d$ a.s..*

**Proof.** Consider the event where $\mathcal{A}_\epsilon = \left\{ |\, \|\boldsymbol{w}_i\|_2 - 1| < \epsilon, |\boldsymbol{w}_i^\top \boldsymbol{w}_j| < \epsilon \right\}$. Under event $\mathcal{A}_\epsilon$, for the diagonal term of the kernel matrix we have

$$\left| [K_W]_{ii} - [\tilde{K}_W]_{ii} \right| = \left| \mathbb{E}_{\boldsymbol{x}}\left[ \phi(\boldsymbol{w}_i^\top \boldsymbol{x})\phi(\boldsymbol{w}_i^\top \boldsymbol{x}) \right] - \mathbb{E}[\phi(G)^2] \right|$$
$$= \left| \mathbb{E}[\phi(\|\boldsymbol{w}_i\|_2 \, G)^2] - \mathbb{E}[\phi(G)^2] \right| = O(\epsilon). \tag{155}$$

And for off-diagonal term, by the decomposition introduced in Section B.3 we have

$$[K_W]_{ij} = \mathbb{E}_{\boldsymbol{x}}\left[ \phi(\boldsymbol{w}_i^\top \boldsymbol{x})\phi(\boldsymbol{w}_j^\top \boldsymbol{x}) \right] = \|\boldsymbol{w}_i\|_2 \|\boldsymbol{w}_j\|_2 + \mathbb{E}_{\boldsymbol{x}}[\phi_\perp(\boldsymbol{w}_i^\top x)\phi_\perp(\boldsymbol{w}_j^\top x)], \tag{156}$$

hence $|[K_W]_{ij} - [\tilde{K}_W]_{ij}| < (\boldsymbol{w}_i^\top \boldsymbol{w}_j)^2$. Notice that $\mathcal{A}_\epsilon$ holds a.s. for $\epsilon = \log^c d/\sqrt{d}$ and large enough $c > 0$; we therefore have $\left\| K_W - \tilde{K}_W \right\|_F \leq \log^c d$. $\qquad \square$

**Lemma 17.** *Let $\hat{\boldsymbol{\beta}}(t)$ be the solution to the gradient flow at time $t$ defined in* (103)(104)*. Then as $n, d \to \infty$ and $\gamma_1 \neq 1$ the following holds.:*

$$\|\hat{\boldsymbol{\beta}}(t)\|_2 = O_P(1); \quad \|X^\top\hat{\boldsymbol{\beta}}(t)\|_\infty = O_P(\text{poly} \log d).$$

**Proof.** We consider $d < n$ for simplicity, and result for the other case follows in similar fashion.

In this case $\hat{\boldsymbol{\beta}}(t) = \left( I - \exp(-\frac{t}{n}XX^\top) \right)(XX^\top)^{-1}X\boldsymbol{y}$. From (Hastie et al., 2019, Corollary 1) we know that $\|\hat{\boldsymbol{\beta}}(\infty)\|_2 = \left\| (XX^\top)^{-1}X\boldsymbol{y} \right\|_2 = O(1)$ for $\gamma_1 < 1$. Note that $\left\| I - \exp(-\frac{t}{n}XX^\top) \right\|_2 = O(1)$ for $t \geq 0$; it follows that $\|\hat{\boldsymbol{\beta}}(t)\|_2 = O(1)$.

For the second part, we utilize the SVD $X = U\Sigma V^\top$, where $\Sigma = \left[ \hat{\Sigma}; 0 \right]$, $\hat{\Sigma} \in \mathbb{R}^{d \times d}$ and $\hat{\Sigma}_{ii} = \lambda_i$. We have $\left( I - \exp(-\frac{t}{n}XX^\top) \right) = U\bar{\Sigma}U^\top$ where $\bar{\Sigma}_{i,i} = 1 - \exp(-t\lambda_i^2/n)$ if $i \leq d$ and 0 otherwise.

$$\|X^\top\hat{\boldsymbol{\beta}}(t)\|_\infty = \left\| X^\top \left( I - \exp(-\frac{t}{n}XX^\top) \right)(XX^\top)^{-1}X\boldsymbol{y} \right\|_\infty$$
$$\leq \left\| V\Sigma^\top U^\top U\bar{\Sigma}U^\top U\hat{\Sigma}^{-2}U^\top U\Sigma V^\top \right\|_\infty \|\boldsymbol{y}\|_\infty$$
$$\leq \left\| V\Sigma^\top \bar{\Sigma}\hat{\Sigma}^{-2}\Sigma V^\top \right\|_\infty \left( \|X^\top\boldsymbol{\beta}\|_\infty + \|\varepsilon\|_\infty \right)$$
$$\leq \|V\|_\infty \left\| \Sigma^\top \bar{\Sigma}\hat{\Sigma}^{-2}\Sigma \right\|_\infty \|V^\top\|_\infty \left( \|X^\top\boldsymbol{\beta}\|_\infty + \|\varepsilon\|_\infty \right) \overset{(i)}{\leq} O_P(\text{poly} \log d), \tag{157}$$

where (i) follows from the concentration of the Gaussian maxima, and the fact that the law of $V$ is the Haar measure on $SO(n)$, and thus for any unit vector $\boldsymbol{z}$ independent to $V$, $V\boldsymbol{z}$ is uniform on sphere and $\|V\boldsymbol{z}\|_\infty = O(\log d/\sqrt{d})$. We wherefore have

$$\|V\|_\infty = \sup_{\boldsymbol{z}} \frac{\|V\boldsymbol{z}\|_\infty}{\|\boldsymbol{z}\|_\infty} = O\left( \frac{\log d}{\sqrt{d}} \right) \frac{\|\boldsymbol{z}\|_2}{\|\boldsymbol{z}\|_\infty} = O(\log d), \tag{158}$$

Note that this result also implies that $\|\boldsymbol{y} - X^\top\boldsymbol{\beta}(t)\|_\infty = O_P(\text{poly} \log d)$.

$\qquad \square$

**Lemma 18.** *For weight matrices $W,W'$ satisfying $\|\boldsymbol{y} - f(X)\|_2 = O(\sqrt{n})$, where $f(X) = \phi(XW)\boldsymbol{a}$ with fixed $a_i \sim \text{Unif}\{-1/\sqrt{h}, 1/\sqrt{h}\}$, given (A1)-(A3), the gradient of the empirical risk defined in* (11) *is Lipschitz w.r.t. $W$ in the Frobenius norm, i.e.*

$$\left\| \frac{\partial L(X;W)}{\partial W} - \frac{\partial L(X;W')}{\partial W} \right\|_F \leq L \|W - W'\|_F. \tag{159}$$

**Proof.** Denote $\boldsymbol{y}_1 = \phi(X^\top W_1)\boldsymbol{a}$ and $\boldsymbol{y}_2 = \phi(X^\top W_2)\boldsymbol{a}$ for $W_1, W_2$ satisfying the assumption above (which can be seen as a condition on the magnitude of training loss), we have

$$\left\| \frac{\partial L(W_1)}{\partial W_1} - \frac{\partial L(W_2)}{\partial W_2} \right\|_F$$

$$= \left\| \frac{1}{n} X \left[ (\boldsymbol{y} - \boldsymbol{y}_1)\boldsymbol{a}^\top \circ \phi'(X^\top W_1) \right] - \frac{1}{n} X \left[ (\boldsymbol{y} - \boldsymbol{y}_2)\boldsymbol{a}^\top \circ \phi'(X^\top W_2) \right] \right\|_F$$

$$\leq \frac{1}{n} \|X\|_2 \left\| (\boldsymbol{y} - \boldsymbol{y}_1)\boldsymbol{a}^\top \circ \phi'(X^\top W_1) - (\boldsymbol{y} - \boldsymbol{y}_2)\boldsymbol{a}^\top \circ \phi'(X^\top W_2) \right\|_F$$

$$\leq O\left( \frac{1}{\sqrt{d}} \right) \left\| (\boldsymbol{y}_2 - \boldsymbol{y}_1)\boldsymbol{a}^\top \circ \phi'(X^\top W_1) \right\|_F$$

$$+ O\left( \frac{1}{\sqrt{d}} \right) \left\| (\boldsymbol{y} - \boldsymbol{y}_2)\boldsymbol{a}^\top \circ (\phi'(X^\top W_1) - \phi'(X^\top W_2)) \right\|_F. \tag{160}$$

We upper bound the two terms separately:

$$\left\| (\boldsymbol{y}_2 - \boldsymbol{y}_1)\boldsymbol{a}^\top \circ \phi'(X^\top W_1) \right\|_2 \overset{(i)}{\leq} \max\{\phi'(X^\top W_1)_{ij}\} \|\boldsymbol{y}_2 - \boldsymbol{y}_1\|_2 \|\boldsymbol{a}\|_2$$

$$\overset{(ii)}{\leq} O(1) \left\| \phi'(X^\top W_1)\boldsymbol{a} - \phi'(X^\top W_2)\boldsymbol{a} \right\|_F$$

$$\overset{(iii)}{\leq} O(1) \|X\|_2 \|W_1 - W_2\|_F = O(\sqrt{d}) \|W_1 - W_2\|_F, \tag{161}$$

where we applied the inequality $\|A \circ B\|_F \leq \max\{|A_{ij}|\} \|B\|_F$ in (i), boundedness of $\phi'$ in (ii) and Lipschitzity of $\phi$ in (iii). Similarly, for the second term

$$\left\| (\boldsymbol{y} - \boldsymbol{y}_2)\boldsymbol{a}^\top \circ (\phi'(X^\top W_1) - \phi'(X^\top W_2)) \right\|_F$$

$$\leq \max\{|a_i|\} \|\boldsymbol{y} - \boldsymbol{y}_2\|_2 \left\| \phi'(X^\top W_1) - \phi'(X^\top W_2) \right\|_F$$

$$\overset{(i)}{\leq} O(1) \|X\|_2 \|W_1 - W_2\|_F = O(\sqrt{d}) \|W_1 - W_2\|_F, \tag{162}$$

where we used the assumption on the training loss and the Lipscthizity of $\phi'$ in (i). Combining the two terms yields the desired result.

$\square$

**Lemma 19.** *Under assumptions (A1-3) and the non-vanishing initialization, given that $\|\boldsymbol{w}_i(t) - \boldsymbol{w}_i(0)\|_2 = O(d^{-1/2})$ for all $i$, then we have $\|K(t) - K(0)\|_2 = O(d^{1/2-\epsilon'})$ for some positive $\epsilon' \in \Theta(1)$.*

**Proof.** Recall the definition of the NTK:

$$K_{ij}(t) = \frac{\partial f(\boldsymbol{x}_i; \boldsymbol{\omega}(t))}{\partial \boldsymbol{\omega}(t)}^\top \frac{\partial f(\boldsymbol{x}_j; \boldsymbol{\omega}(t))}{\partial \boldsymbol{\omega}(t)} = \boldsymbol{x}_i^\top \boldsymbol{x}_j \frac{1}{h} \sum_{k=1}^h \phi'(\boldsymbol{w}_k(t)^\top \boldsymbol{x}_i)\phi'(\boldsymbol{w}_k(t)^\top \boldsymbol{x}_j), \tag{163}$$

or equivalently the matrix form

$$K(t) = X^\top X \circ \frac{1}{h}[\phi'(X^\top W(t))\phi'(W(t)^\top X)]. \tag{164}$$

At initialization, $\boldsymbol{x}_i \sim N(0, I_d)$ and $\boldsymbol{w}_k(0) \sim N(0, d^\epsilon I_d)$. Thus for fixed $\boldsymbol{x}_i$, by Gaussian anti-concentration we have $\Pr|\boldsymbol{x}_i^\top \boldsymbol{w}_k| < \log d \leq O(1/d^{1/2+\epsilon_1})$ for some $\epsilon_1 > 0$. In addition, note that $\|\boldsymbol{w}_k(t) - \boldsymbol{w}_k(0)\|_2 = O(d^{-1/2})$ for all $k$, and therefore for $i, j, k$ such that

$|\boldsymbol{x}_i^\top \boldsymbol{w}_k(0)| > O(\log d)$ and $|\boldsymbol{x}_j^\top \boldsymbol{w}_k(0)| > O(\log d)$, we know that $|\phi'(\boldsymbol{x}_i^\top \boldsymbol{w}_k(t))\phi'(\boldsymbol{x}_j^\top \boldsymbol{w}_k(t)) - \phi'(\boldsymbol{x}_i^\top \boldsymbol{w}_k(0))\phi'(\boldsymbol{x}_j^\top \boldsymbol{w}_k(0))| = O(d^{-2})$.

Given fixed $\boldsymbol{x}_i$, define $y_k = \mathbf{1}\{|\boldsymbol{x}_i^\top \boldsymbol{w}_k| < \log d\}$ as the indicator variable that the $k$-th neuron does not saturate. We know that $\mathbb{E}[y_k] = O(1/d^{1/2+\epsilon_1})$, and $\mathrm{Var}[y_k] = \mathbb{E}[y_k^2] - \mathbb{E}[y_k]^2 = O(1/d^{1/2+\epsilon_1})$. By Bernstein's inequality

$$\Pr\left| \frac{1}{h} \sum_{k=1}^h y_k - \mathbb{E}[y_k] \right| > \varepsilon \leq 2 \exp\left( -\frac{h\varepsilon^2}{2\sigma^2 + 2\varepsilon/3} \right). \tag{165}$$

Setting $\varepsilon = \sqrt{\frac{c \log h}{h^{1+\epsilon_2}}}$, we know that with probability at least $1 - h^{-c}$,

$$\frac{1}{h} \sum_{k=1}^h y_k \leq \varepsilon + \mathbb{E}[y_k] = O\left( \frac{\mathrm{poly} \log h}{h^{1/2+\epsilon_3}} \right). \tag{166}$$

Therefore, given $\boldsymbol{x}_i$ and $\boldsymbol{x}_j$, for large enough $c_1$ with probability at least $1 - h^{-3}$ we have

$$\left| \sum_{k=1}^h \phi'(\boldsymbol{w}_k(t)^\top \boldsymbol{x}_i)\phi'(\boldsymbol{w}_k(t)^\top \boldsymbol{x}_j) - \sum_{k=1}^h \phi'(\boldsymbol{w}_k(0)^\top \boldsymbol{x}_i)\phi'(\boldsymbol{w}_k(0)^\top \boldsymbol{x}_j) \right| = O(h^{1/2-\epsilon_4}), \quad (167)$$

in which we utilized the boundedness of $\phi'$. Taking union bound over $d^2$ elements in the random feature matrix yields

$$\begin{aligned}
& \|K(t) - K(0)\|_2 \\
=& \left\| X^\top X \circ \frac{1}{h} \left[ \phi'(X^\top W(t))\phi'(W(t)^\top X) - \phi'(X^\top W(0))\phi'(W(0)^\top X) \right] \right\| \\
\leq& \frac{1}{h} \left\| X^\top X \right\|_2 \max \left\{ \left| \phi'(X^\top W(t))\phi'(W(t)^\top X) - \phi'(X^\top W(0))\phi'(W(0)^\top X) \right|_{ij} \right\} \\
\leq& \frac{1}{h} O(d)O(d^{1/2-\epsilon'}) = O(d^{1/2-\epsilon'}).
\end{aligned} \tag{168}$$

Using the exact same argument, one can derive that $\|\boldsymbol{u}_{NN}(\hat{\boldsymbol{x}}) - \boldsymbol{u}_{NTK}(\hat{\boldsymbol{x}})\|_2 = O(d^{1/2-\epsilon'})$, the proof of which we omit.

$\square$

# E    ADDITIONAL RESULTS

## E.1    RISK OF ReLU NETWORK UNDER SYMMETRIC DATA

If the dataset is symmetric, that is

**(A5) Symmetric Data:** $\forall i \in [1, n], \exists! j \in [1, n]$ s.t. $\boldsymbol{x}_i + \boldsymbol{x}_j = 0$,

then population risk of the gradient flow solution can be given explicitly for certain nonlinearities:

**Proposition 20.** *Given (A1-3)(A5), if the nonlinearity satisfies $\phi'(\boldsymbol{x}) + \phi'(-\boldsymbol{x}) = C$ for constant C, then as $n, d, h \to \infty$*

$$R_{(\gamma_1 < 0.5)}(\hat{f}) \to \frac{2\gamma_1}{1 - 2\gamma_1} \sigma^2; \quad R_{(\gamma_1 \geq 0.5)}(\hat{f}) = \left( 1 - \frac{1}{2\gamma_1} \right) r^2 + \frac{1}{2\gamma_1 - 1} \sigma^2. \tag{169}$$

Note that the requirement on the nonlinearity holds for ReLU and SoftPlus. This expression is again independent to $\gamma_2$ and aligns with the experimental results in Figure 9 (we only plot the bias component for verification). In addition, the bias is upper-bounded by the null risk for all $\gamma_1$. We remark that the symmetry assumption does not hold for i.i.d. samples from symmetric distributions, and Figure 9 demonstrates that the additional condition alters the risk.

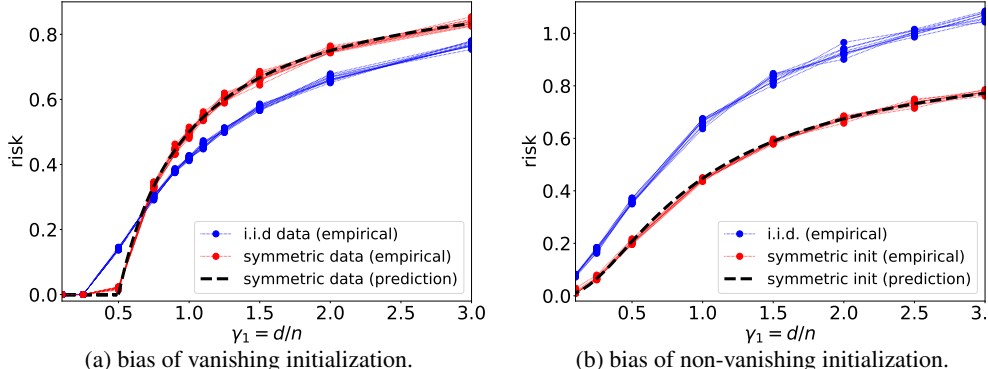

Figure 9: Bias of two-layer ReLU networks with optimized first layer under Gaussian data and linear teacher. Individual dotted lines correspond to different $\gamma_2$ (from 0.2 to 2) which is independent to the risk. (a) Vanishing initialization. The bias under symmetric data is predicted by Proposition 20. (b) Non-vanishing initialization. The red and blue lines represent models optimized from i.i.d. and symmetric initialization, respectively. The bias for symmetric initialization is predicted by Theorem 8.

**Proof.** Without loss of generality assume $X = [X_0, -X_0]$. Then by (100) we have

$$\frac{\partial \boldsymbol{w}_+}{\partial t} = \frac{1}{2n_0} \sum_{i=1}^{2n_0} \left[ \left( y_i - h_0\phi(\boldsymbol{w}_+^\top \boldsymbol{x}_i) + h_0\phi(\boldsymbol{w}_-^\top \boldsymbol{x}_i) \right) \phi'(\boldsymbol{w}_+^\top \boldsymbol{x}_i) \boldsymbol{x}_i \right], \tag{170}$$

and the flow for $\boldsymbol{w}_-$ follows from symmetry. In this case one can show that from exact zero initialization, for nonlinearity satisfying $\phi(x) - \phi(-x) = x$, such as ReLU and SoftPlus,

$$\frac{\partial(\boldsymbol{w}_+)}{\partial t} + \frac{\partial(\boldsymbol{w}_-)}{\partial t} = \frac{1}{2n_0} \sum_{i=1}^{2n_0} \left[ \left( y_i - h_0\phi(\boldsymbol{w}_+^\top \boldsymbol{x}_i) + h_0\phi(\boldsymbol{w}_-^\top \boldsymbol{x}_i) \right) (\phi'(\boldsymbol{w}_+^\top \boldsymbol{x}_i) - \phi'(\boldsymbol{w}_-^\top \boldsymbol{x}_i)) \boldsymbol{x}_i \right] = 0. \tag{171}$$

And therefore the gradient flow of $\boldsymbol{w}_+$ is

$$\begin{aligned}
\frac{\partial \boldsymbol{w}_+}{\partial t} &= \frac{1}{2n_0} \sum_{i=1}^{2n_0} \left[ \left( y_i - h_0\phi(\boldsymbol{w}_+^\top \boldsymbol{x}_i) + h_0\phi(-\boldsymbol{w}_+^\top \boldsymbol{x}_i) \right) \phi'(\boldsymbol{w}_+^\top \boldsymbol{x}_i) \boldsymbol{x}_i \right] \\
&= \frac{1}{2n_0} \sum_{i=1}^{n_0} \left[ \left( y_i - h\phi(\boldsymbol{w}_+^\top \boldsymbol{x}_i) + h_0\phi(-\boldsymbol{w}_+^\top \boldsymbol{x}_i) \right) (\phi'(\boldsymbol{w}_+^\top \boldsymbol{x}_i) + \phi'(-\boldsymbol{w}_+^\top \boldsymbol{x}_i)) \boldsymbol{x}_i \right] \\
&= \frac{1}{2n_0} \sum_{i=1}^{n_0} \left[ \left( y_i - h_0\boldsymbol{w}_+^\top \boldsymbol{x}_i \right) \boldsymbol{x}_i \right] = \frac{1}{2n_0} X_0 \boldsymbol{y}_0 - \frac{1}{2n_0} h_0 X_0 X_0^\top \boldsymbol{w}_+. \tag{172}
\end{aligned}$$

The flow of $\boldsymbol{w}_-$ follows from symmetry. Solving for the stationary points (i.e. gradient becomes zero), it the clear that

$$\boldsymbol{w}_+^{(t=\infty)} = -\boldsymbol{w}_-^{(t=\infty)} = \begin{cases} \dfrac{1}{h_0}(XX^\top)^{-1}X\boldsymbol{y}, & \gamma_1 < 0.5, \\[2mm] \dfrac{1}{h_0}X(X^\top X)^{-1}\boldsymbol{y}, & \gamma_1 > 0.5. \end{cases} \tag{173}$$

And hence the asymptotic risk is

$$R_{(\gamma_1 < 0.5)} \to \frac{2\gamma_1}{1 - 2\gamma_1}\sigma^2; \quad R_{(\gamma_1 \geq 0.5)} = \left(1 - \frac{1}{2\gamma_1}\right)r^2 + \frac{1}{2\gamma_1 - 1}\sigma^2. \tag{174}$$

The same conclusion holds for vanishing initialization if we assume that the trajectory stays close to that of exact zero initialization. Note that although the prediction aligns well with the experimental results, the argument in Theorem 7 does not directly apply due to the undefined derivative of ReLU at the origin, and thus this result is not rigorously justified. $\qquad\square$

# F    EXPERIMENT SETUP

**Optimizing the Second Layer.**    We compute the minimum-norm solution by directly solving the pseudo-inverse. We set $n = 1000$ and vary $\gamma_1, \gamma_2$ from 0.1 to 3. The linear teacher model $F(\boldsymbol{x}) = \boldsymbol{x}^\top \boldsymbol{\beta}$ is fixed as $\boldsymbol{\beta} = -\mathbf{1}_d/\sqrt{d}$. For each $(\gamma_1, \gamma_2)$ we average across 50 random draws of data.

**Optimizing the First Layer.**    For both initializations, we use gradient descent with small step size ($\eta = 0.1$) and train the model for minimally 25000 steps and till $\|\nabla_W f(X, W)\|_F^2 < 10^{-6}$. We fix $n = 320$ and vary $\gamma_1, \gamma_2$ from 0.1 to 3 with the same linear teacher model $\boldsymbol{\beta} = -\mathbf{1}_d/\sqrt{d}$. The risk is averaged across 20 models trained from different initializations.

