# OpenReview forum: "Generalization of Two-layer Neural Networks: An Asymptotic Viewpoint"
_ICLR.cc/2020/Conference — Accept (Spotlight)_

### Official Review · AnonReviewer2 · 2019-10-14
**Official Blind Review #2**

**Rating:** 8

**Review:**

The authors study the generalization error of two-layer neural nets, where an asymptotic point of view is taken. Their main results can be summarized as follows.
1. If only the second layer is optimized, they observe the double-descent phenomenon.
2. However, if only the first layer is optimized, the double-descent is not observed.
This shows that recent results for certain linear models (e.g. Song, Montanari 2019) do not directly transfer to neural networks. As the authors point out, however, if a different scaling is used in the asymptotics, double descent might still be observed.

I see the following strengths of the paper.
-This is a very well-written paper with a clear message.
-The result is important and gives new insights into the generalization properties of neural networks.

In my view, this is an interesting contribution, which should be accepted.

---------

Thank you for your response. I will leave the rating unchanged.

**Experience Assessment:**

I do not know much about this area.

**Review Assessment: Checking Correctness Of Derivations And Theory:**

I did not assess the derivations or theory.

**Review Assessment: Checking Correctness Of Experiments:**

I did not assess the experiments.

**Review Assessment: Thoroughness In Paper Reading:**

I read the paper at least twice and used my best judgement in assessing the paper.

---

> ### Author Response · Authors · 2019-11-15
> **Reply to Reviewer 2**
>
> Thank you for the comments and suggestions. We agree that characterizing the generalization properties of neural network under different scalings is an important future direction.
>
> We have updated the manuscript with a few minor modifications: 1) Figure on the population risk of sigmoid network (first layer optimized) in addition to SoftPlus; 2) additional remarks on the population risk of network in the kernel regime in Section 5.2; 3) corrected typos.

---

### Official Review · AnonReviewer3 · 2019-10-21
**Official Blind Review #3**

**Rating:** 6

**Review:**

This paper provides exact bounds on the risk when training a two-layer neural network in an asymptotic regime. Namely, the paper considers training under the square-loss objective, a two-layer neural network with $h$ hidden units on inputs of dimension $d$ and training on $n$ samples. The asymptotic regime is considered by making all of $d$, $h$, $n$ go to $\infty$, in a way that the ratio $d/n$ approaches $\gamma_1$ and the ratio $h/n$ approaches $\gamma_2$.

This paper considers the following scenarios of training described below, where the data is generated from a linear model on Gaussian inputs and with a zero-mean noise. The emphasis of the results is on understanding when a "double descent" type phenomenon occurs ("Double descent" is a recently coined phenomenon in literature where the risk, as a function of the "complexity of the model", initially has a classical U-shape behavior, but eventually decreases again once the complexity of the model exceeds the number of training points.)

1. Training only the second layer: The risk is first decomposed into a bias and a variance term. An exact bound on the variance term of the risk is obtained. While the exact nature of the bound is rather complex to parse, the takeaway is that a double descent phenomenon is observed in terms of $\gamma_2$, namely, the risk blows up when $h \approx n$, but decreases as $h$ is increased beyond $n$.

2. Training only the first layer: Two different regimes are considered here, depending on the scale of initialization, called "vanishing" and "non-vanishing" initializations. In both regimes, the risk is independent of $\gamma_2$, that is, the risk does not depend on number of hidden units (although the risk bounds are different and there is an additional assumption in the case of non-vanishing initialization to ensure that the initialized network computes the zero function). In other words, a "double descent" phenomenon is not observed in this setting.

Recommendation:
I recommend "weak acceptance". The paper extends prior works that obtain asymptotic risk bounds on linear models to the setting of two-layer neural networks (where only one layer is trained).  However, I am unable to assess the technical novelty of this work as it seems to heavily rely on prior work which in turn use techniques from random matrix theory.

Technical Comments:
- I felt that while it is valuable to have exact bounds on the risk, the form of the bounds are quite complex and hard to parse (especially in Thm 4, case of training only the second layer). Moreover, these bounds are just in the case where the teacher model is linear and while it is claimed that this could be relaxed to a more general class of functions, the specific bounds might change drastically. So any insights on the nature of these bounds will be valuable, especially with some comments on how these bounds change if the teacher model is itself realized as a 2-layer neural network.
- The parameter count of a 2-layer network with $h$ hidden units and input dimension $d$ is $O(dh)$. So perhaps it makes sense to study an asymptotic regime where $dh/n$ approaches $\gamma$, instead of both d and h growing linearly in n. While this issue is hinted at in the discussion section, I don't understand the statement "the mechanism that provably gives rise to double descent from previous works Hastie et al. (2019); Belkin et al. (2019) might not translate to optimizing two-layer neural networks."
- Another future direction that could be included in discussions is the setting where both layers are trained simultaneously.

**Experience Assessment:**

I have read many papers in this area.

**Review Assessment: Checking Correctness Of Derivations And Theory:**

I did not assess the derivations or theory.

**Review Assessment: Checking Correctness Of Experiments:**

I assessed the sensibility of the experiments.

**Review Assessment: Thoroughness In Paper Reading:**

I read the paper at least twice and used my best judgement in assessing the paper.

---

> ### Author Response · Authors · 2019-11-15
> **Reply to Reviewer 3**
>
> Thank you for the comments and suggestions. The technical comments are addressed below:
>
> Extending result to other target functions:
> We agree that the problem might be significantly more difficult for different target functions, and would like to make the following remarks:
> 1. Note that in our bias-variance decomposition, only the bias term depends on the target function. In other words, our result on the variance (including Theorem 4) would still be valid for other targets, such as two-layer neural network. One caveat is that for general target function, the output needs to be properly scaled since our current analysis in Section 5 relies on linearizing the network.
> 2. When the target function is a multiple-neuron neural network, deriving the bias term can be challenging. However, we note that under the same setup, the bias may be obtained when the teacher is a slightly more general single-index model, i.e. $y=\psi(\beta^\top x)$ with Lipschitz link function $\psi$, equivalent to a single-neuron network. For instance, the bias under vanishing initialization is the same as that of least squares regression on the input, which can be solved under isotropic prior on $\beta$ via decomposing the activation function similar to Appendix C.5.
>
> Parameter count:
> To clarify our statement in the discussion section, our current result requires $n,d,h$ to grow at the same rate, and thus $n = O(dh)$ is beyond the regime we consider. This is also true for previous works on double-descent in random feature model [Hastie et al. (2019)][Mei and Montanari (2019)].  When $h \ll n$, it is not clear if the same analysis still applies (for instance approximating the network with a kernel model), and thus the instability of the inverse may not be the complete explanation of double-descent (if it appears). Characterizing the generalization in this regime would be an interesting direction.
>
> Training both layers:
> Thank you for the suggestion; we have included training both layers simultaneously as a future direction. We would like to briefly mention that under certain model parameterization and initialization, gradient flow on both layers may reduce to one of the three models we analyzed (see [Williams et al. (2019)]). More generally, our current result may be extended to cases where the dynamics of training both layers can be linearized (for instance initialization in the "kernel regime"), for which the learned model can be written down in closed-form.

---

### Official Review · AnonReviewer1 · 2019-10-23
**Official Blind Review #1**

**Rating:** 8

**Review:**

Overview: This work is an interesting work to understand the generalization capabilities of a two layered neural network in a high dimensional setting (samples, features and neurons tend to infinity). It studies the conditions under which the "double descent phenomenon" may be observed.

Summary: The work shows that in two layered neural networks with non-linearity
1) the double descent phenomenon of the bias-variance decomposition may be observed when the second layer weights are optimized assuming that the first layer weights are constant.
2) the bias-variance decomposition does not exhibit double descent when optimizing only the first layer with both vanishing and non-vanishing initialization of weights.
3) For vanishing initalization of weights for the first layer with non-linear activation , the gradient flow solution is asymptotically close to a two layered linear network. It is independent of overparametrization. However, the condition for this is smooth activation and the result does not hold for ReLU activation.
4) For non-vanishing initilization of the weights for the first layer with non-linear activation, the gradient flow solution is well approximated by a kernel model. However, the risk is independent of overparametrization.

I believe this is an interesting work that needs to be accepted.

**Experience Assessment:**

I have read many papers in this area.

**Review Assessment: Checking Correctness Of Derivations And Theory:**

I assessed the sensibility of the derivations and theory.

**Review Assessment: Checking Correctness Of Experiments:**

I did not assess the experiments.

**Review Assessment: Thoroughness In Paper Reading:**

I read the paper at least twice and used my best judgement in assessing the paper.

---

> ### Author Response · Authors · 2019-11-15
> **Reply to Reviewer 1**
>
> Thank you for the comments and suggestions. As you pointed out, our current result in Section 5 does not apply to non-smooth activations -- understanding the generalization of ReLU networks would be interesting future work.
>
> We have updated the manuscript with a few minor modifications: 1) Figure on the population risk of sigmoid network (first layer optimized) in addition to SoftPlus; 2) additional remarks on the population risk of network in the kernel regime in Section 5.2; 3) corrected typos.

---

### Decision · Program_Chairs · 2019-12-19

**Decision:**

Accept (Spotlight)

**Comment:**

This paper focuses on studying the double descent phenomenon in a one layer neural network training in an asymptotic regime where various dimensions go to infinity together with fixed ratios. The authors provide precise asymptotic characterization of the risk and use it to study various phenomena. In particular they characterize the role of various scales of the initialization and their effects. The reviewers all agree that this is an interesting paper with nice contributions. I concur with this assessment.  I think this is a solid paper with very precise and concise theory. I recommend acceptance.